# Modelling extreme water levels using intertidal topography and bathymetry derived from multispectral satellite images

Wagner Luiz Langer Costa[1], Karin Roisin Bryan[1], Giovanni Coco[2]

[1]School of Science, University of Waikato, Hamilton, Aotearoa New Zealand.
[2]School of Environment, University of Auckland, Auckland, Aotearoa New Zealand.

*Correspondence to: Wagner Luiz Langer Costa (wc119@students.waikato.ac.nz)*

**Abstract.** Topographic and bathymetric data are essential for accurate predictions of flooding in estuaries because water depth and elevation data are fundamental components of the shallow-water hydrodynamic equations used in models for storm surges and tides. Where LiDAR or in-situ acoustic surveys are unavailable, recent efforts have centred on using satellite images to derive bathymetry (SDB) and topography (SDT). This work is aimed at (1) determining the accuracy of SDT and (2) assessing the suitability of the SDT and SDB for surge/tidal modelling of estuaries. The SDT was created by extracting the waterline as it tracks over the topography with changing tides. The method was applied to four different estuaries in Aotearoa New Zealand: Whitianga, Maketū, Ōhiwa and Tauranga Harbours. Results show that the waterline method provides similar topography to the LiDAR with a root-mean-squared error equal to 0.2 m, and it is slightly improved when two correction methods are applied to the topography derivations: the removal of statistical bias (0.02 m improvement) and hydrodynamic modelling correction of waterline elevation (0.01 m improvement). The use of SDT in numerical simulations of surge levels was assessed for Tauranga Harbour in eight different simulation scenarios. Each scenario explored different ways of incorporating the SDT to replace the topographic data collected using non-satellite survey methods. In addition, one of these scenarios combined SDT (for intertidal zones) and SDB (for subtidal bathymetry), so only satellite information is used in surge modelling. The latter SDB is derived using the well-known ratio-log method. For Tauranga Harbour, using SDT and SDB in hydrodynamic models does not result in significant differences in predicting high water levels when compared with the scenario modelled using surveyed bathymetry.

## 1 Introduction

Coastal flooding has become increasingly concerning because of growing storm intensity (Emanuel, 2005; Webster et al., 2005; Sobel et al., 2016) and sea-level rise, which will potentially increase the risk exposure of coastal communities (Nicholls and Cazenave, 2010; Oppenheimer et al., 2019). In practice, predicting flooding depends on understanding the contribution from the astronomical tide, storm surge, wave run-up, changes in the sea level and, in some cases, the fluvial discharge and vertical land motion. In coastal zones, these processes can interfere with each other, for example, in tide-surge interactions (Spicer et al., 2019; Wankang et al., 2019). In the case of estuaries, bathymetric and topographic data are essential for coastal risk

assessment (Parodi et al., 2020) because they influence the accuracy of water level predictions (Cea and French, 2012; Pedrozo-Acuña et al., 2012; Falcão et al., 2013; Mohammadian et al., 2022). Water depth is a fundamental component in the shallow-water hydrodynamic equations used in extreme water level modelling. Together with the estuary's geometry and length — which can cause shoaling and choking — and bed-shear stress — which reduces energy due to its effect on friction — bathymetry and topography control the amplitude and phase (timing) of the propagating tide. The estuary's morphology is also fundamental for studying the tidal response to sea-level rise ( Du et al., 2018; Khojasteh et al., 2020, 2021).

The methods used to measure bathymetry and topography in coastal zones have evolved rapidly. In estuaries, there are permanently inundated areas and intertidal zones, which are flooded and exposed to the tide. Here we define the terms bathymetry and topography to reflect permanently-inundated and intertidal areas, respectively. Currently, there are four types of systems for measuring these: ship-based systems (e.g., single-beam and multibeam echosounders); non-imaging active remote sensing (e.g. LiDAR); imaging active remote sensing (e.g. synthetic aperture radar — SAR); and, imaging passive remote sensing (e.g. optical systems) (Jawak et al., 2015; Salameh et al., 2019; Ashphaq et al., 2021). Traditionally, the most commonly-used systems are echosounders and LiDAR. Both produce highly accurate data; however, several factors constrain their application, such as cost, labour, inaccessibility of remote areas, and environmental conditions (e.g., low tide navigational restrictions). Consequently, according to IHO (2020), approximately 70% of the world's coastal areas have bathymetric surveys that need updating or are insufficiently detailed (e.g., are of large scale 1:100).

Space-borne remote sensing techniques overcome the limitations of traditional techniques and can provide topographic and bathymetric data for a wide range of environments, including areas that are more difficult to measure, such as remote shallow coastal waters (Lyzenga, 1985; Ehses and Rooney, 2015; Caballero and Stumpf, 2019) and extensive intertidal areas (Bishop-Taylor et al., 2019; Fitton et al., 2021). Several methods are used to derive bathymetric data — hereafter called satellite-derived bathymetry (SDB) — in shallow waters (i.e., between 0–15 m depth) using imaging passive remote sensing of reflectance (Ashphaq et al., 2021). Most methods are developed around the process of light attenuation through the water column and can be categorised into two approaches. Empirical methods — which use direct observations of water depth in the study area to calibrate the reflectance-to-depth relationship (e.g., Stumpf et al., 2003; Caballero and Stumpf, 2019) — and physics-based inversion algorithms — which use physical processes/models to solve for water depth (e.g., radiative transfer models) without the need for in situ calibration data (e.g., Lee et al., 1998, 1999; Kerr and Purkis, 2018).

The present manuscript focuses on empirical methods to obtain the SDB, including the ratio-log method proposed by Stumpf et al. (2003). The main limitations of the Stumpf method are the requirement of in situ bathymetric data for calibration and its sensitivity to environmental conditions that can change bottom and water reflectance — e.g., high water turbidity and variation in the benthic substrates — both of which often occur in enclosed seas, bays and estuaries (Morris et al., 2021). Some studies have proposed techniques to tackle these empirical issues (e.g., Geyman and Maloof, 2019; Caballero and Stumpf, 2020). For

example, Geyman and Maloof (2019) implemented the cluster-based regression algorithm to deal with different bottom
substrates, first segmenting the satellite image into zones of spectral homogeneity and then calibrating the log-linear colour-
to-depth relationship separately for each class. Caballero and Stumpf (2020) adjusted reflectance ratios to reduce the effects
of water turbidity and calculated the maximum chlorophyll index prior to analysis to identify pixels containing floating and
submerged vegetation, allowing them to remove these pixels before further implementation of the ratio-log formula.

In intertidal regions, remote sensing can also be used to obtain satellite-derived topography (SDT) — and the waterline method
is the most commonly applied. The method was first applied to SAR images (Mason and Davenport, 1996) and recently also
to multispectral space-borne images (Khan et al., 2019; Salameh et al., 2020; Fitton et al., 2021). The technique functions by
detecting the edge between the flooded and exposed intertidal zone in multiple images (i.e., the waterline) and assigning a
height to each waterline by using the local tidal level at the time of image acquisition. The tidal level can be acquired by a
numerical tide model (e.g., Khan et al., 2019; Kang et al., 2020; Salameh et al., 2020) or from a local tide gauge (Mason and
Davenport, 1996; Salameh et al., 2020). The resulting collection of waterlines is interpolated over the intertidal domain,
generating a digital elevation model (DEM). The approach assumes that estuary morphology does not change between images
and has a gentle slope. The main disadvantages of this method are the dependence of accuracy on the number of images used
in the processing and the reduced performance when applied to sites with complex morphology, i.e., variable terrain slopes
within the intertidal zone (Liu et al., 2013; Salameh et al., 2019, 2020). Other methods used to derive topography in intertidal
zones are the interferometric SAR (Li and Goldstein, 1990), satellite radar altimetry (Salameh et al., 2018) and near-infrared
logistic approach (Bué et al., 2020).

As remote-sensing techniques have developed, cloud computation and storage systems such as Google Earth Engine (Gorelick
et al., 2017) have also advanced considerably. Consequently, scientists now have an enhanced capacity to quickly manage
large geographical datasets, allowing global-scale studies in coastal science to evolve rapidly (e.g., Murray et al., 2019; Vos
et al., 2019; Bishop-Taylor et al., 2019). For instance, databases now exist on the distribution of and changes to global tidal
flats (Murray et al., 2019), as well as a global estimate of coastline position (Almeida et al., 2021; Vos et al., 2019). Satellite-
derived bathymetry (SDB) and topography (SDT) techniques are now routinely applied over extensive areas (e.g., Traganos
et al., 2018; Bishop-Taylor et al., 2019; Fitton et al., 2021). Despite the vast and growing application of SDB and SDT methods
to coastal science and engineering (Turner et al., 2021), it is not yet clear whether the accuracy of the resulting
bathymetry/topography is suitable for modelling extreme water levels in coastal areas (e.g., estuaries and bays). Only limited
studies exist on SDB, SDT and numerical modelling —  generally aimed at using the model to assign the waterline height
(Khan et al., 2019; Salameh et al., 2020; Fitton et al., 2021). For instance, Mason et al. (2010) used SDT to calibrate a
morphodynamical model.

The present study aims to evaluate whether SDT and SDB can replace surveyed data as a boundary condition in hydrodynamic modelling — focusing on predicting high water levels (surges and extreme high tides) in estuaries with complex morphology. The study has three specific objectives:

100

1. To determine whether satellite imagery can be used to extract accurate SDT;
2. To investigate the main source of errors in the satellite-derived techniques; and,
3. To assess the use of SDT and SDB for hydrodynamic modelling of estuaries as an alternative to data derived from traditional methods.

105

This manuscript is divided into two main parts, as illustrated by the two grey panels in Figure 1: (a) the SDT and SDB framework and (b) the hydrodynamic modelling assessment. The chart flow follows the numbered sections within the text, with the two small left-side panels contributing to Parts (a) and (b). The methods section begins with a description of the study sites and database (blue box in Figure 1). Following this, the optical waterline (black box) and ratio-log (light-yellow box) methods for generating SDT (waterline and ratio-log) and SDB (ratio-log) are described. The hydrodynamic modelling description is given in Sect. 2.4 (left side box in Figure 1). Following, the two correction methods are explained: the dynamical correction (green box, Figure 1) and the statistical correction (red box, Figure 1). The main workflow for the modelling assessment of SDT/SDB is illustrated in Part (b) (Figure 1) and consists of running simulation scenarios using different combinations of topo-bathymetry datasets and hydrodynamic forcing conditions. Results for Parts a and b are shown in Sect. 3, including the waterline-derived and the ratio-log-derived intertidal elevation, the proposed correction techniques, and the modelling assessment. In Sect. 4, the main findings are discussed: the advantages and limitations of our proposed SDT and SDB framework and correction approaches; and the hydrodynamic modelling assessment. The conclusion is in Sect. 5 (not shown in Figure 1).

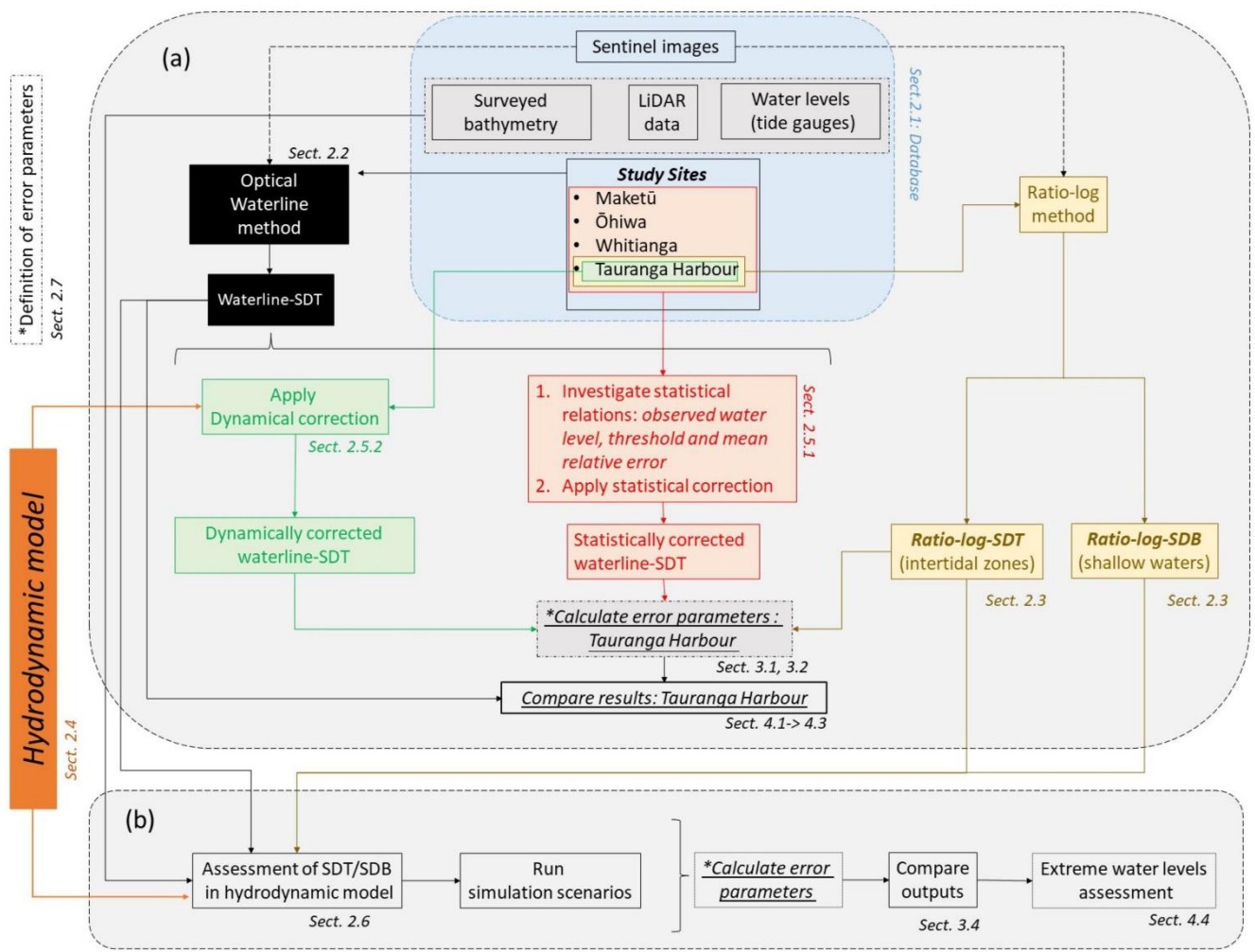

**Figure 1: A flow chart showing the main structure of the manuscript. Panel (a) shows the steps taken to derive the SDT/SDB and how the statistical relationships and source of errors were investigated. Panel (b) summarises the framework to test the utility of SDT/SDB in modelling high water levels.**

## 2 Methods

### 2.1 Study site and database

The study areas are four estuaries on the east coast of Aotearoa New Zealand's North Island. Three are in the Bay of Plenty region: Tauranga, Ōhiwa and Maketū Harbours and one in the Coromandel: Whitianga Harbour, Figure 2 (a). The sites consist of barrier-enclosed sandy estuaries, which are common in Aotearoa New Zealand (Hume et al., 2007, 2016) and all have micro-tidal regimes — the spring tidal range varies between 1.4 m to 1.9 m — and spring tides combined with severe storm surges drive the extreme sea levels (Rueda et al., 2019; Stephens et al., 2020). In Aotearoa New Zealand, the storm surges usually add ≤ 0.5 m to the water level; however, larger storm surges can occur occasionally (Stephens et al., 2020). The extensive

intertidal zones and vegetation (e.g., seagrass and mangrove) that are present in the majority of the estuaries in Aotearoa New Zealand can attenuate tides (Tay et al., 2013) and storm surges (Montgomery et al., 2019). The water level inside the estuaries is not considered to be substantially affected by waves (i.e., wave set-up) because all of them are enclosed coastal lagoons with restricted entrances. All four estuaries have large intertidal areas covering 58% to 84% of the estuaries' total area (Hume et al.,

2007, 2016); see Table 1. The extent of the tidal flats is evident in Tauranga Harbour by comparing low and high tide satellite images, Figure 2 (b) and (c), respectively. Mangrove forests can be observed in all the estuaries, and seagrass banks are visible in Maketū, Ōhiwa and Tauranga Harbours. Detailed images of the intertidal zones in Tauranga Harbour, showing seagrass banks and mangroves, can be seen in Figure S3.

Imagery, tidal levels and topography data (e.g., LiDAR) were acquired to implement and validate the SDT techniques. For the Bay of Plenty region, historical tide levels were downloaded from the Bay of Plenty Council data portal (https://envdata.boprc.govt.nz/); the topography data consisted of the LiDAR survey, with a spatial resolution of $1 \times 1$ m, available on the Land Information New Zealand (LINZ) data portal (https://data.linz.govt.nz/). For Whitianga, water level time series and elevation data (LiDAR) were acquired through the Thames-Coromandel District Council's website

(http://www.tcdc.govt.nz/). The LiDAR data have an accuracy of $\pm$ 0.2 m in the vertical and $\pm$ 0.6 m in the horizontal with 95% confidence for the Bay of Plenty. All LiDAR data were converted to the local vertical datum (i.e., Moturiki 1953), which is 0.13 m below mean sea level (MSL), using the GEOID elevation grids available in the LINZ data portal.

Satellite images from European Space Agency (ESA) Copernicus Sentinel were accessed through Google Earth Engine

(Gorelick et al., 2017) and consisted of spacecraft Sentinel 2A and B, product type level-2A. The Sentinel-2 products are composed of elementary tiles, which are $100 \times 100$ km$^2$ ortho-images in the UTM/WGS84 projection, with a revisit frequency of 5 days in the Aotearoa New Zealand region. The level-2A product type provides bottom-of-atmosphere (BOA) images, which are already corrected for the effects of the top-atmosphere, terrain and cirrus cloud using the Sen2Cor processing tool (ESA). Each image has the spectral resolution of 12 bands with spatial resolution differing between 10, 20 and 60 m depending

on the band. The green (band 3, 560 nm), blue (band 2, 490 nm) and near-infrared (band 8, 842 nm) bands were used for this analysis, all of them with 10 m spatial resolution.

In summary, a complete set of LiDAR, tidal gauge observations and a satellite image was obtained for each estuary. For example, the Tauranga Harbour dataset is shown in Figure 2, including the location of the tide gauges (Ōmokoroa, Hairini,

Ōruamatua, and Moturiki) and the intertidal exposure during low tide (Figure 2b) and high tide (Figure 2c). In Figure 2 (d), the corresponding water level records for the acquisition period of the satellite images are shown. (Moturiki time series is not shown).

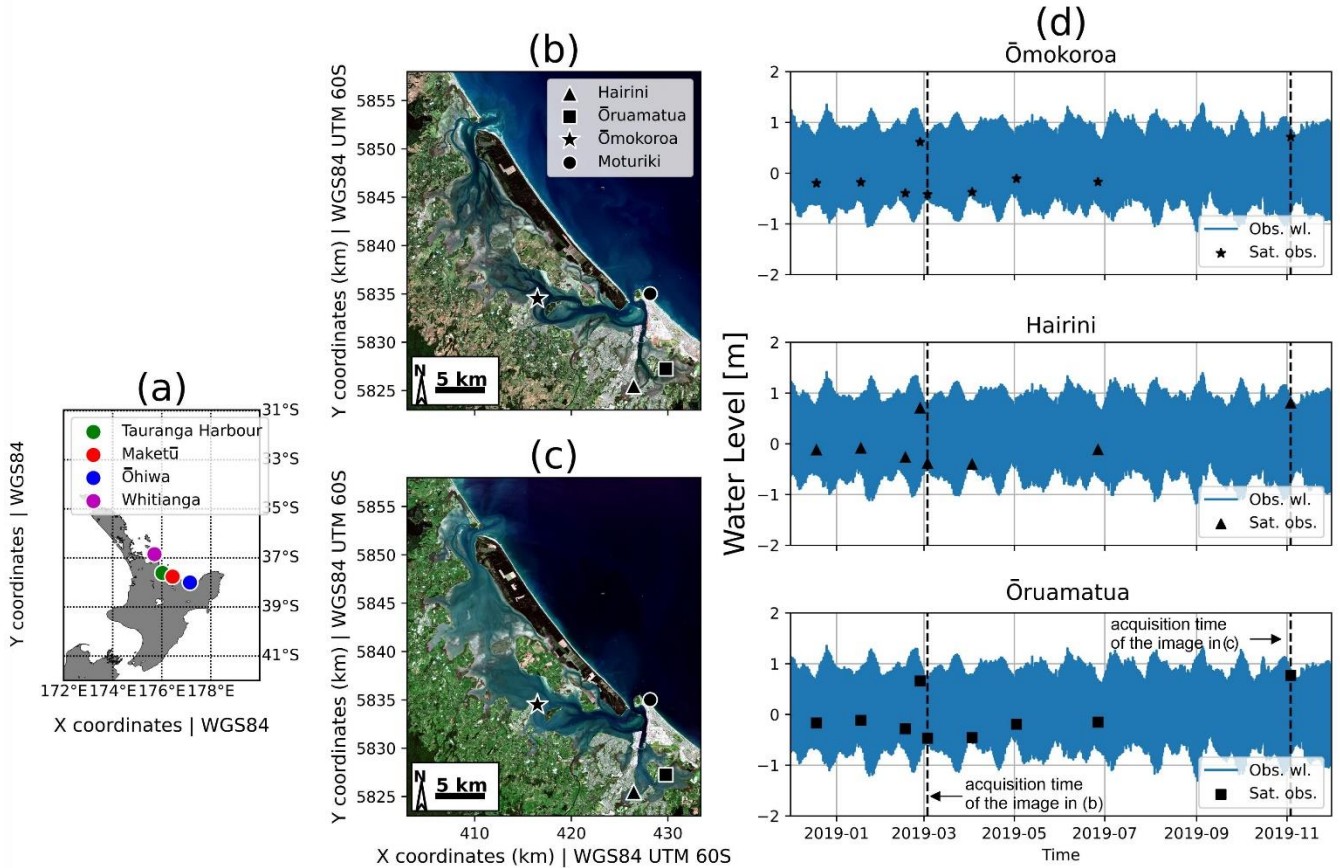

**Figure 2: The four estuaries where the SDT method was tested (a). Tauranga Harbour and tide gauge locations during low tide (b) and high tide (c) with the background image from ESA Sentinel 2A. Water level time series from the three of local tide gauges shown in panels b and c during the period over which satellite images were acquired (d). The water levels associated with images shown in panels b and c are marked with a vertical dashed black line in (d) (Vertical Datum: MSL).**

## 2.2 Satellite-derived topography: the waterline method.

The framework to generate the SDT in intertidal zones using the waterline method (hereafter called waterline-SDT) was composed of three stages, as illustrated in Figure 3: Stage 1 was to query an image collection; Stage 2 was to identify the intertidal zone; and, Stage 3 was to determine the waterline position and height. First, an image collection was acquired for each estuary through the Google Earth Engine application (Gorelick et al., 2017) using the Google Colaboratory environment. Each image collection has images from the satellite Sentinel 2A and B, product type level-2A, covering the estuary domain, in which fewer than 5% of the pixels are covered by clouds. A small number of images with low cloud coverage were included because of the restricted number of available images; however, any irregularities from the small areas of clouds and their shadows were removed manually in post-processing quality control. The number of images corresponding to the collection and environmental properties for each estuary (e.g., coverage of intertidal zone in the estuary; spring tidal range) is shown in Table 1.

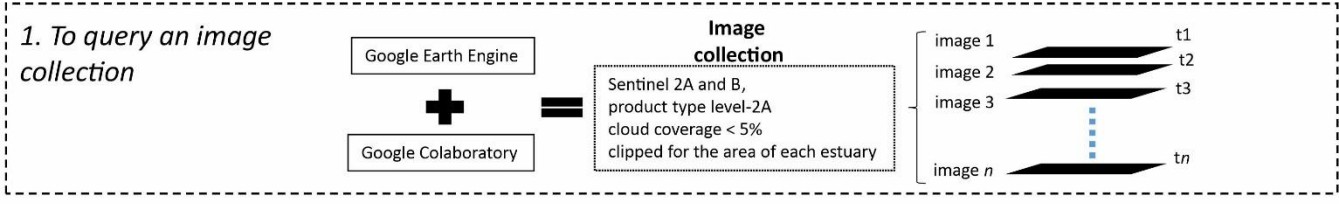

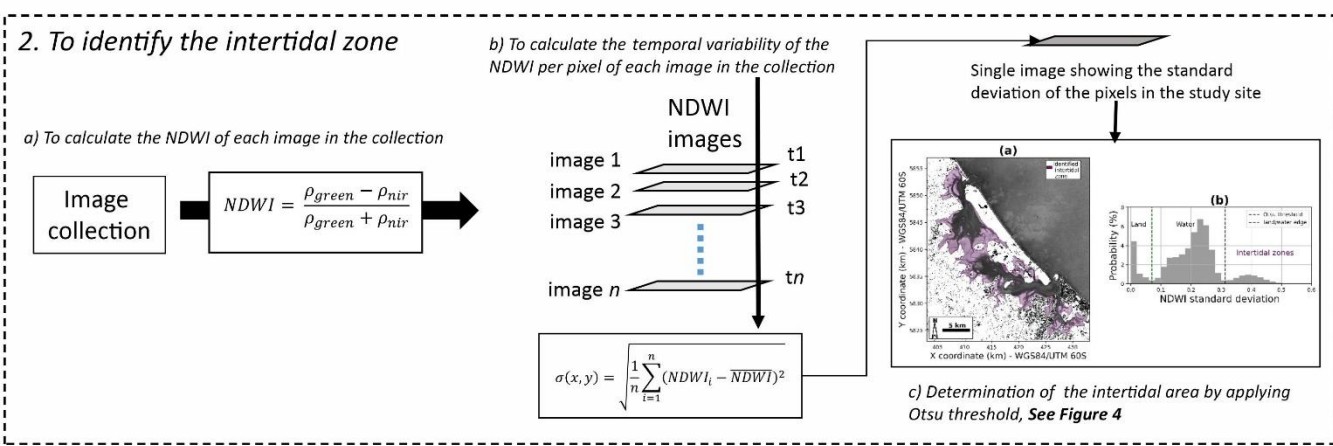

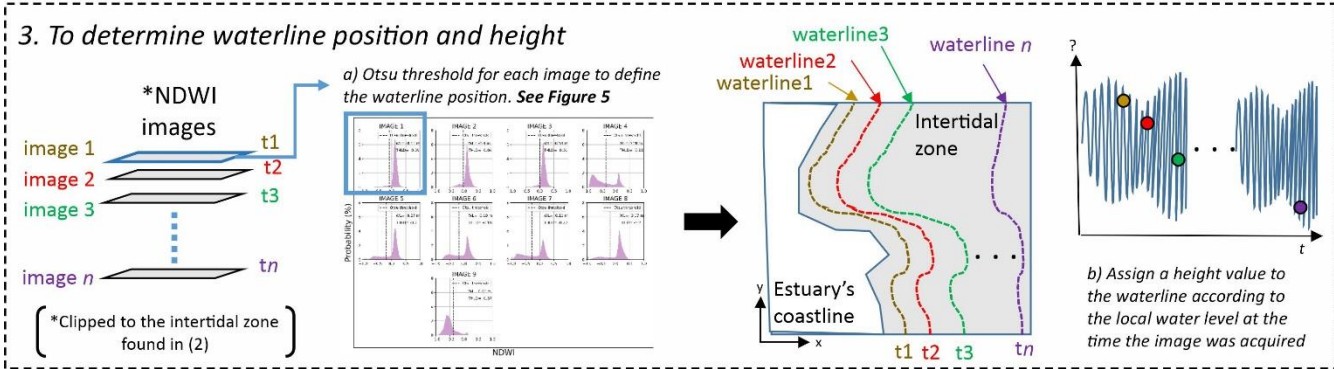

**Figure 3: The framework for the application of the waterline method to derive topographic data in intertidal zones. First (1) an image collection was acquired. Second (2), the intertidal zone was identified by calculating the temporal of NDWI. Note that NDWI is the index used to detect the existence of water from satellite reflectance (see text). Third (3), the waterline position and height were determined. This was done by identifying the boundary between wet and dry cells within the intertidal zone (i.e., waterline) and assigning a height value for the waterline obtained from the local tide gauge observation at the time of the image acquisition.**

**Table 1 Number of images in the image collection for each estuary.**

| Estuary | Nº of images in the collection | Total intertidal area (Hume et al., 2016) | Surface area (Hume et al., 2016) | Spring tidal range |
|---|---|---|---|---|
| | | | | |

| | | | | |
|---|---|---|---|---|
| *Tauranga Harbour* | 9 | 77% | ~200 km$^2$ | 1.75 m |
| *Ōhiwa* | 6 | 84% | ~27 km$^2$ | 1.9 m |
| *Maketū* | 12 | 58% | ~ 2.6 km$^2$ | 1.4 m |
| *Whitianga* | 8 | 72% | ~ 15.5 km$^2$ | 1.7 m |

In the second stage (Figure 3), the intertidal zone was identified. The aim was to eliminate pixels that are not in the intertidal

area — thus avoiding needless image processing. For that, the approach based on Bué et al. (2020) was used, in which the intertidal extent was determined by calculating the temporal variability of the Normalized Difference Water Index (NDWI) (McFeeters, 1996) at each pixel over the entire image collection, using Equation 1 and 2:

$$\sigma(x,y) = \sqrt{\frac{1}{n}\sum_{i=1}^{n}(NDWI_i - \overline{NDWI})^2};$$ (1)

where

$$NDWI = \frac{\rho_{green} - \rho_{nir}}{\rho_{green} + \rho_{nir}};$$ (2)

and where *x* and *y* are the pixel coordinates, and *n* is the number of images in the collection. *NDWI* is determined as a normalised difference between $\rho_{green}$ and $\rho_{nir}$, which are the reflectance of the green and near-infrared bands of Sentinel-2 images, respectively. As a result, one single greyscale image was generated representing the *NDWI* temporal standard deviation ($\sigma$), Figure 4 (a) because of the consistent change between exposed (low tide) and inundated (high tide) conditions, the highest standard deviation values were assumed to occur in the intertidal zones. Thus, the pixels representing the intertidal zone were

the ones with an $\sigma$ greater than the threshold value. For instance, Figure 4 (b) shows that for Tauranga Harbour, the threshold is > 0.32. The threshold was set using the Otsu (1979) approach, where its value depends on the probability distribution of $\sigma$, as illustrated in Figure 4 (b). The Otsu method identifies the optimum threshold between two data classes in the image distribution that maximises the value of the within-class variance. The advantage of using an adaptive threshold is that it can be objectively tailored to each image collection and estuary.

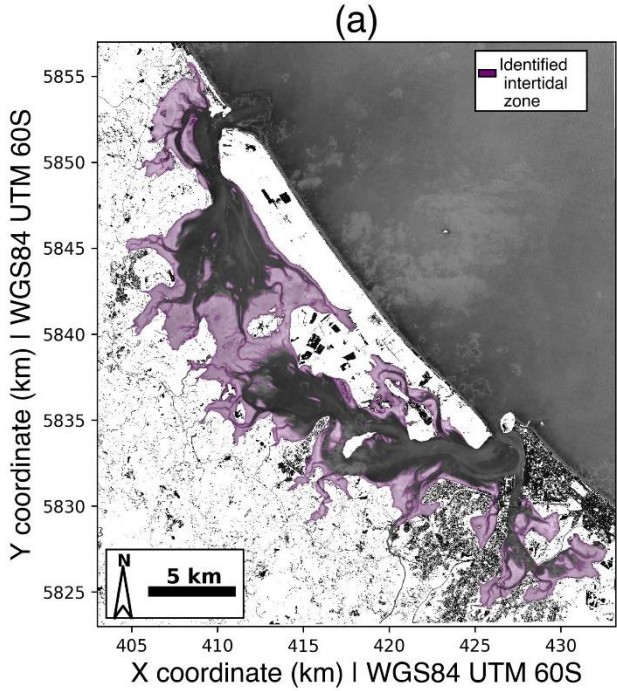

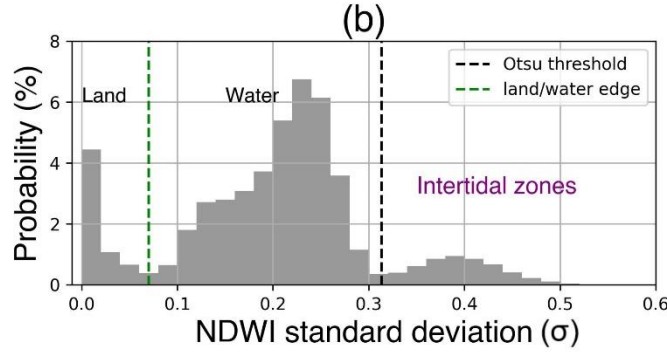

**Figure** 4: **(a) Intertidal areas identified using the temporal variability of *NDWI* (σ) of the Tauranga Harbour image collection. Determination of the Otsu threshold for the identification of the intertidal zone (b).**

In the third stage (Figure 3), for each image in the collection, the corresponding *NDWI* maps were clipped into the intertidal zone (which was defined using the whole image collection in stage two). From the intertidal *NDWI* maps, the waterline position
in that image was extracted by applying the algorithm "Finding_Contours" from the scikit.measure (Van Der Walt et al., 2014) Python library. This contour extraction method searches for a given value (i.e., threshold) in a two-dimensional array of pixels, using the 'marching squares' algorithm (Lorensen and Cline, 1987) to identify contour boundaries precisely by linearly interpolating between adjacent pixel values; therefore, the method is able to define waterline with a subpixel resolution. The Otsu method was used to determine the threshold that should be applied to each image. Figure 5 illustrates the different
thresholds applied to each image in the Tauranga collection according to the corresponding distribution of *NDWI*. Once the waterline position was identified for a given image, a height value was assigned to the waterline by finding the corresponding observed tide level at the local tide gauge (Ōmokoroa for the Tauranga Harbour case study, Figure 2d). After all images in the collection were processed, a collection of waterlines with different height values was created (see Figure 3, Stage 3b), which was gridded to create an SDT (i.e., waterline-SDT). The accuracy of the waterline-SDT was assessed against the LiDAR data
by comparing the LIDAR at each point along the waterline and by comparing the corresponding digital elevation model (DEM). The module DELFT-QUICKIN was used to create DEMs for each estuary. The triangular interpolation method was applied in a grid with a spatial resolution of 10 m.

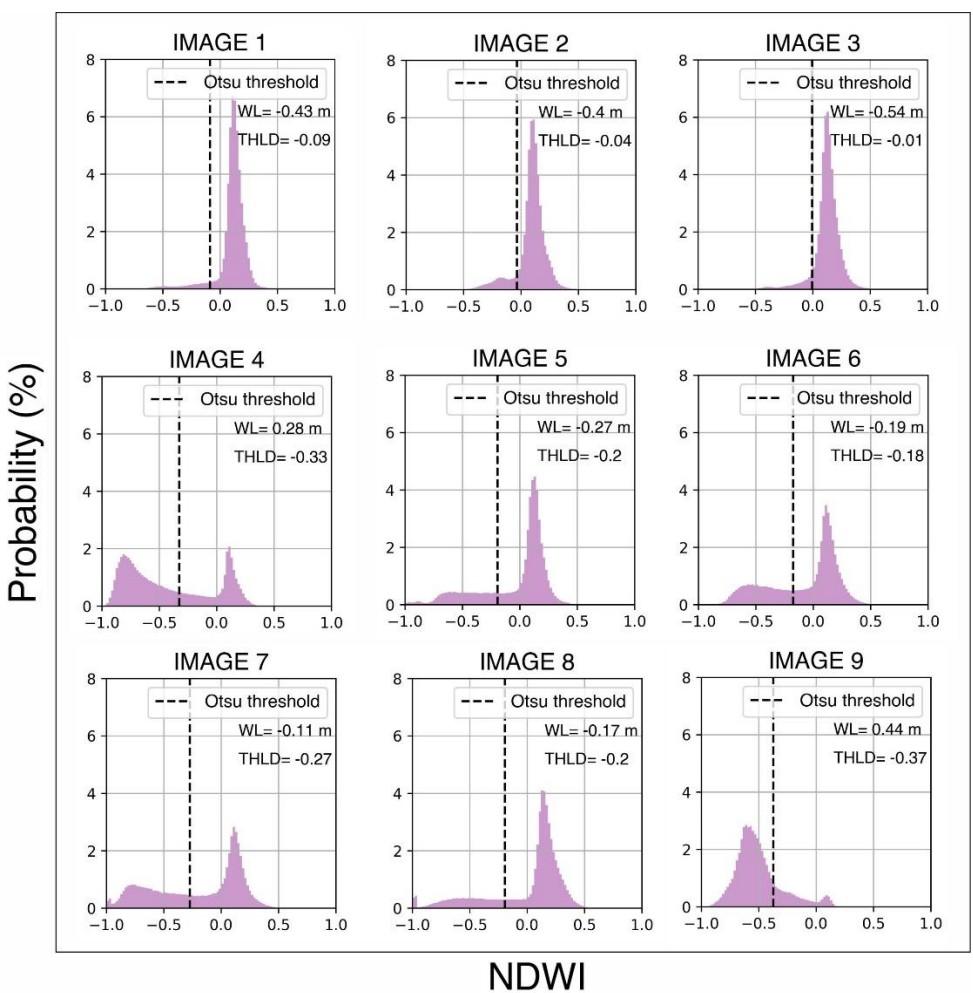

**Figure 5: Otsu threshold (THLD) applied to identify the waterline position for each image in the Tauranga Harbour collection. The observed water level from the Ōmokoroa tide gauge at the moment of the image acquisition (i.e., waterline height relative to mean sea level) is also shown in each panel (marked WL).**

### 2.3 Satellite-derived bathymetry: the ratio-log method.

Additionally, the ratio-log method (Stumpf et al., 2003) was applied in Tauranga Harbour separately for intertidal zones (ratio-log-SDT) and shallow water (ratio-log-SDB). The ratio-log and waterline methods were assessed in two different ways: first, to compare their approximation of topographic data in intertidal zones (Figure 1a) and second, to compare their use in hydrodynamic models (Figure 1b). The ratio-log method is not limited by the number of satellite images available covering the estuarine area and can be applied to extract the SDT from a single image. In its original application, the ratio-log empirical approach was used to derive shallow water bathymetry (SDB). However, because of the relatively low turbidity of intertidal water in Tauranga Harbour, the method may also be suitable for deriving topography (SDT) within intertidal zones. To compare, the ratio-log method was applied to an image acquired at high tide, where the intertidal zone was completely flooded.

The numerical assessment was built on a pilot study by Costa et al. (2021), where the method was trialled in a small region within the Tauranga Harbour. Detailed information about the application of the ratio-log method and the results for tidal flats and shallow water are provided in Supplement A, Figure S1 and Figure S2. Because the ratio-log method is based on an empirical fit, additional bathymetric data were needed to implement the ratio-log method for shallow water. For this, multiple-

source bathymetric data were used, which is detailed in Sect. 2.4.

### 2.4 Hydrodynamic modelling: the baseline model

A baseline hydrodynamic model was set up for Tauranga Harbour. The modelling study was only undertaken in this estuary (rather than all four sites) because a calibrated and validated DELFT3D-FLOW model was already available (Stewart, 2021). The baseline model was first used to account for the tidal propagation within the estuary as a way of improving the waterline

SDT (Sect. 2.5.2) and, second, was used as a base-case against which to assess the use of SDT and SDB as the bathymetry/topography for modelling extreme water levels in estuaries (Sect. 2.6). The grid domain and interpolated bathymetry are shown in Figure S4 (Supplement B), and they cover the central to the southern part of the Harbour with a 20 × 20 m resolution grid. The north and south boundaries were set as open boundaries (free Neuman), and the model was forced with water level along the seaward boundary with the astronomical components of the tide. For the latter, harmonic

astronomical tidal analysis was undertaken on the Moturiki Island tide gauge using U_tide (Codiga, 2011). The topographic and bathymetric data used in the baseline hydrodynamic model were assembled using a combination of data from multiple sources: Multibeam survey (Port of Tauranga, 2017), LiDAR (2008 from AAMHATCH and 2016 from LINZ) and LINZ hydrological charts NZ 5411, 2016. These data were all converted to mean sea level (MSL) vertical reference.

The model was validated to ensure the bed roughness parameters were appropriate by simulating an equinoctial tidal period from 01/03/2019 to 31/03/2019. The details of the model setup, calibration and validation are presented in Supplement B. The vertical datum in the simulation was the MSL, and the time step was 0.5 min; the advection scheme for calculating the flooded and dried cells is cyclic, using the water level averaged on the grid cells. The model was calibrated against three tide gauge observation points (Ōmokoroa, Hairini and Ōruamatua). For details on the model calibration and validation, see Supplement

B.

### 2.5 The SDT correction approaches

### 2.5.1 Correcting SDT using the bias between LiDAR data and SDT: the statistical correction

The first method to correct the waterline-SDT trialled was to remove the statistical bias—potentially caused by conditions that can interfere with the pixel reflectance and, as a consequence, the waterline position at different tide levels within the tidal

flats. Conditions that can interfere with detection include complex intertidal zone morphology, water turbidity, variation of the benthic substrates (sand, seagrass), and groundwater seepage. Specifically, groundwater seepage leaves a film of moisture on

the exposed intertidal detectable in images (Huisman et al., 2011). Because the studied estuaries have similar characteristics — i.e., morphology complexity (extensive intertidal zones and channelisation), tidal range, white sand, and presence of seagrass — a statistical correction was developed for all four estuaries on the basis that the detected waterline is consistently further seaward or landward than the actual waterline across all sites. For the statistical correction, a linear equation was fitted to the relationship between the value of the Otsu threshold — used to position the waterline within the intertidal zone (see Sect. 2.2) — and the bias between the waterline-SDT and the LiDAR data in all the estuaries (i.e., Tauranga, Maketū, Ōhiwa, and Whitianga Harbours).

### 2.5.2 Correcting SDT using the hydrodynamic model: the dynamical correction

The second method to correct the waterline SDT was using the hydrodynamic model (Sect. 2.4); this correction approach is hereafter called the dynamical correction. The water level is not homogeneous throughout a large estuary at one instant in time because of the time it takes for the tide to propagate around the estuary and the potential tidal wave deformations induced by the estuarine morphology, such as shoaling, reflection and dampening. Thus, for every image processed by the waterline method, the detected waterline height might vary spatially, and assigning a waterline height using information from one tide gauge (as in Sect. 2.2) may not be appropriate. The dynamical correction uses the hydrodynamic model to assign a spatially-varying height to each waterline.

The dynamical correction was implemented as follows. First, the model bathymetry was replaced in the intertidal zones with the waterline-SDT (with the original bathymetry retained in the shallow water areas), and a new depth file was created. Using this new depth file, nine independent simulation cases were performed corresponding to the acquisition times of the nine images in the Tauranga Harbour image collection. Each case had a simulation period starting ten days prior to the date and time that the satellite image was acquired to allow for model spin-up. The spatially-varying water level model output was extracted along each of the corresponding waterlines (one waterline is detected in each image, Sect.2.2). The waterline height was assigned by interpolating the position of each waterline onto the gridded model output.

### 2.6 Assessing extreme water level simulations with SDB and SDT

To properly assess the use of satellite-derived topo-bathymetric data in hydrodynamic modelling, scenarios were designed with different combinations of SDB, SDT, and hydrodynamic forcing conditions. Table 2 shows the specifications of the eight simulation scenarios: the source of elevation data, the physical forcing, and the simulation period. Note that the "surveyed topo-bathymetry" data refers to the multiple-source data described in Sect. 2.4.

The four first scenarios (i.e., S1–S4) test different combinations of topo-bathymetric data to be used in hydrodynamic modelling. Here, the model was forced with the astronomical constituents extracted from the Moturiki tide gauge record, as described in Sect. 2.4, varying only the topo-bathymetric data. For instance, our baseline model (S1) represents the condition

when the modeller depends only on the in situ measured elevation. In the S2 and S3 scenarios, the intertidal zone bathymetry was replaced with the SDT generated by using the waterline (waterline-SDT) and the ratio-log (ratio-log-SDT) methods, respectively. The S4 scenario was developed to assess the use of only SDB and SDT in the entire model domain (so only satellite-derived data). Thus, the waterline-SDT was used for the tidal flat and the ratio-log-SDB for the shallow areas within the harbour.

The simulation scenarios S5–S8 were designed to investigate the potential of SDT and SDB as a replacement for the surveyed topo-bathymetry in modelling more extreme water level events, where the interactions between storm surge and astronomical tides are important. For that, two extreme events were selected based on the water level observations at the Moturiki tide gauge in the period ranging from the years 2002 to 2018, see Figure S10. Trends of sea-level rise were first filtered from the water level observations by applying a 1-year-running average, which resulted in a filtered time series hereafter called total water level. Both extreme events represented the highest water levels for the period and consisted of peaks of a total water level of ~1.4 m. However, the contribution of the storm surge is different for each event. The definition of storm surge used here is the difference between astronomical tide and the detrended water level for a given time series. The first event occurred on 16/04/2003 and consisted of a storm surge = 0.55 m, which represents ~ 40% of the observed total water level. The second event occurred on 05/01/2018 and consisted of a storm surge = 0.32 m, which represents ~ 22% of the observed total water level. Ultimately, these two events were modelled using only surveyed topo-bathymetry (S5 and S7) and only satellite-derived data (S6 and S8).

To assess the simulations, the tide levels extracted from each scenario were compared to the astronomic tide at three points for S1–S4 (Ōmokoroa, Hairini, and Ōruamatua), the total water level at two points for S5–S8 (Hairini, and Ōruamatua), and the tide and total water level output maps from each simulated scenario. Note that the Ōmokoroa tide gauge was not used to assess S5–S8 because its records do not cover the period of the first storm event.

**Table 2 - Simulation scenarios to assess the use of SDT and SDB in hydrodynamic modelling.**

| Scenarios | Source intertidal zone | Source shallow waters | Forcing | Contribution of storm surge at the peak of water level | Simulation start time | Simulation end time |
|---|---|---|---|---|---|---|
| S1 | surveyed topo-bathymetry | surveyed topo-bathymetry | Astronomical constituents | - | 01/03/2019 | 31/03/2019 |
| S2 | waterline-SDT | surveyed topo-bathymetry | " | - | " | " |
| S3 | ratio-log-SDT | surveyed topo-bathymetry | " | - | " | " |
| S4 | waterline-SDT | Ratio-log-SDB | " | - | " | " |

| | | | | | | |
|---|---|---|---|---|---|---|
| S5 | surveyed topo-bathymetry | surveyed topo-bathymetry | Total water level | 22% | 02/01/2018 | 10/01/2018 |
| S6 | waterline-SDT | Ratio-log-SDB | " | " | " | " |
| S7 | surveyed topo-bathymetry | surveyed topo-bathymetry | Total water level | 40% | 10/04/2003 | 19/04/2003 |
| S8 | waterline-SDT | Ratio-log-SDB | " | " | " | " |

## 2.7 Assessment of framework performance

The accuracy of the SDB, SDT, the hydrodynamic model, and the dynamical and statistical corrections was assessed by calculating the following error metrics: root mean square error (RMSE), maximum absolute error (MAXE), relative error (RE), coefficient correlation ($R^2$), and bias (BIAS) (Eq. 3–7 respectively). In the corresponding equations, $h_{est}$ is the estimated value (e.g., SDT, SDB, hydrodynamic model output), and $h_{obs}$ is the observed value (e.g., LiDAR data, tide gauge measurements).

In the case of SDB and SDT evaluation, the RE can be either negative or positive. For the results derived from the ratio-log method (SDT and SDB), the RE reflects only the vertical difference between the estimate and the in-situ data (i.e., LiDAR data for SDT and surveyed bathymetry for SDB). For the waterline-SDT, the RE reflects both vertical and horizontal accuracies. For instance, Figure 6 shows a schematic topo-bathymetric profile illustrating the error calculation for the waterline-SDT. Although the error is evaluated as a height difference, it can originate from either horizontal ($\delta x$) or vertical ($\delta z$) inaccuracies. Thus, negative (positive) RE means that the estimate is shallower (deeper) or located further landward (seaward) than the LiDAR data.

$$RMSE = \sqrt{\sum_{i=1}^{n} \frac{(h_{est} - h_{obs})^2}{n}} ; \tag{3}$$

$$MAXE = \max_{i=1\ldots n} |h_{est}i - h_{obs}i| ; \tag{4}$$

$$RE = h_{obs} - h_{est} ; \tag{5}$$

$$R^2 = \frac{\sum_{i=1}^{n}(h_{est}i - \overline{h_{obs}})^2}{\sum_{i=1}^{n}(h_{obs_i} - h_{est_i})^2 + (h_{est_i} - \overline{h_{obs}})^2} ; \tag{6}$$

$$BIAS = \overline{h_{obs}} - \overline{h_{est}} \quad ; \tag{7}$$

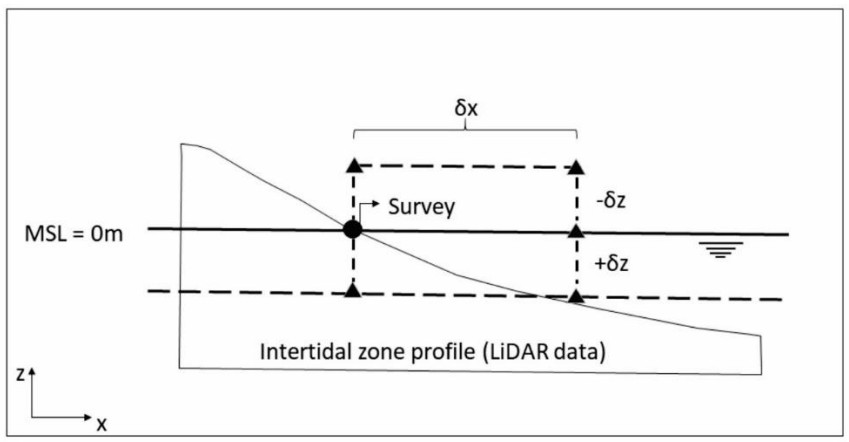

**Figure 6: Schematic showing the error calculation. The circle shows the actual location of the water line, and triangles show the location of the remotely sensed shoreline. There are two ways that an error can be caused. The waterline can be detected landward or seaward of its actual location (δx), or the waterline is assigned an elevation that is too high or too low (δz).**

## 3 Results

### 3.1 The waterline satellite-derived topography (waterline-SDT)

The waterline-SDT accuracy, compared to the LiDAR data, for all the studied estuaries is shown in Table 3; the average RMSE across all estuaries was 0.33 m, and the average MAXE was 1.74 m. The technique's worst performance was in the Maketū Estuary (RMSE =0.41 m and MAXE= 2.38 m), Figure S6. Ōhiwa and Whitianga Estuary have similar performance, Figures S5 and S7. Tauranga Harbour was associated with the best estimates with RMSE = 0.20 m. Note that the error parameters calculated for the corresponding DEMs are lower, especially in terms of MAXE. The detail of the images that were acquired and the corresponding water level for each estuary is shown in Supplement C.

**Table 3   Waterline-SDT errors for every studied estuary. DEM is the digital elevation model obtained by interpolating the corresponding waterline-SDT in the intertidal zone with a spatial resolution of 20m and triangulation method. The elevation range in the LiDAR data within the intertidal zone is also shown. Vertical Datum: MSL.**

|  | SDT | | DEM | | LiDAR |
| --- | --- | --- | --- | --- | --- |
| *Estuary* | *RMSE (m)* | *MAXE (m)* | *RMSE (m)* | *MAXE (m)* | *elevation range (m)* |
| *Maketū* | 0.41 | 2.38 | 0.47 | 2.19 | -0.63 \| +1.90 |
| *Ōhiwa* | 0.35 | 2.00 | 0.34 | 1.61 | -0.98 \| +2.98 |
| *Tauranga Harbour* | 0.20 | 1.60 | 0.23 | 1.14 | -1.11 \| +1.44 |
| *Whitianga* | 0.35 | 1.00 | 0.28 | 1.17 | -1.12 \| +1.92 |
| *Average* | 0.33 | 1.74 | 0.33 | 1.53 | |

Although the SDT accuracy differed depending on the estuary, the BIAS increased at high and low tide for all estuaries, and the lowest errors occurred at mid-tide. Figure 7 shows the linear correlation found between the bias (BIAS), the waterline height (Z), and the Otsu adaptive threshold (THLD), which also demonstrates the aforementioned dependence on the tide level. For instance, the THLD and the BIAS have a goodness of fit of $R^2 = 0.58$ (Figure 7b); the THLD and the Z have a goodness of fit of $R^2 = 0.83$ (Figure 7a), and BIAS and Z have a goodness of fit of $R^2 = 0.68$ (Figure 7c). However, whether a waterline is extracted on the flooding or the ebbing tide cycles does not affect the waterline-SDT accuracy. Equation 8 (also shown in Figure 7c) was used as the basis of the statistical correction (where the statistical bias was removed from the waterline-SDT for Tauranga Harbour (Sect. 2.4.1)).

$$\text{BIAS} = -0.49 * Z - 0.088 \qquad (8)$$

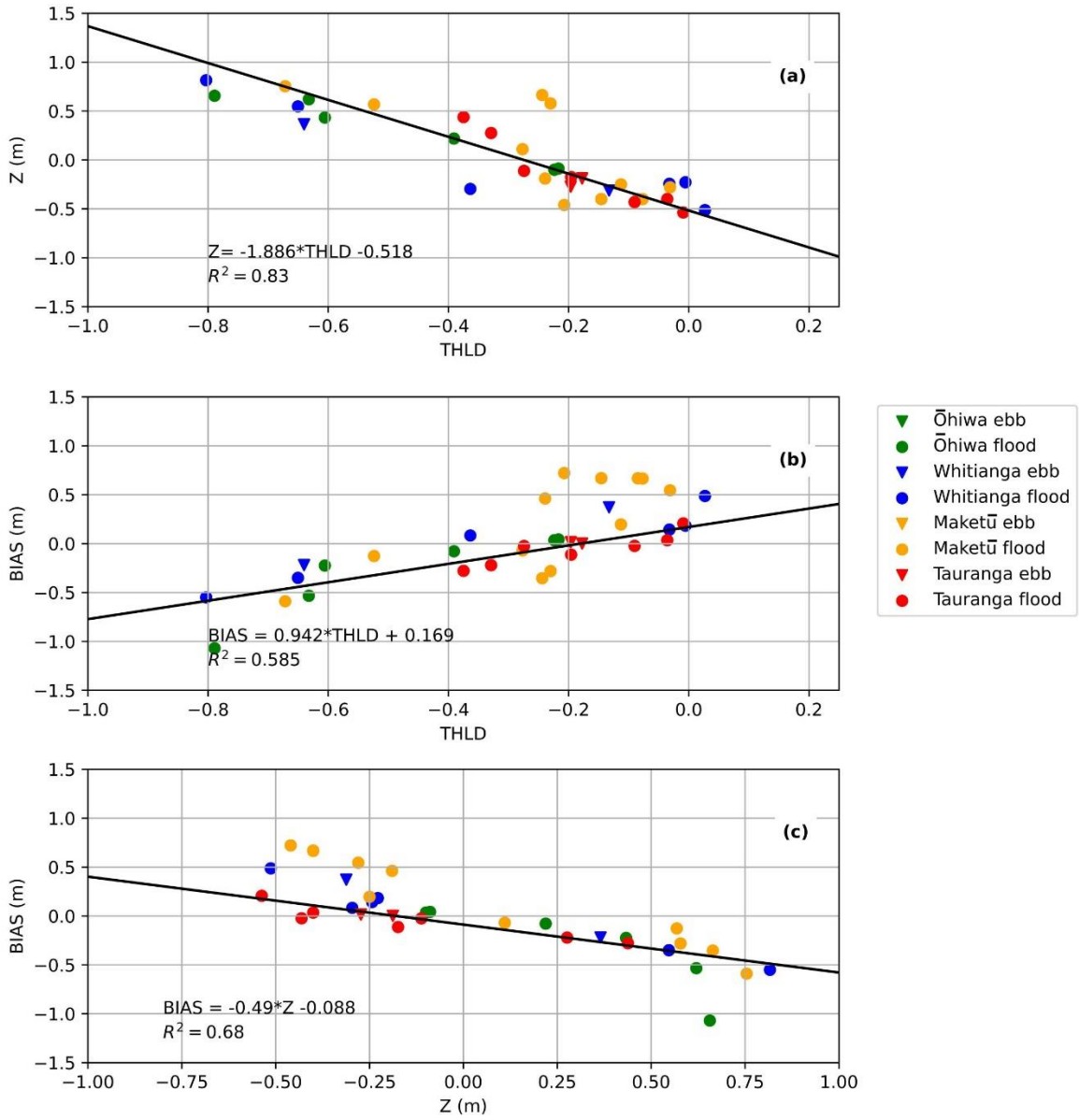

**Figure 7: Statistical relationships at all estuaries (Ōhiwa, Whitianga, Tauranga, Maketū): (a) water level Otsu threshold (THLD) and observed water level (Z); (b) THLD and the SDB mean error (bias); (c) Z and the SDB mean relative error (bias). The relationship shown in (c) was used in the statistical correction (see Sect. 3.3).**

## 3.3 The statistical and dynamical corrections

The waterline-SDT (Sect. 2.2), the statistically (Sect. 2.4.1 and 3.1), and the dynamically (Sect. 2.4.2) corrected waterline-SDT showed an overall RMSE equal to 0.20, 0.18 and 0.19 m, respectively (Figure 8). The statistical correction was effective where the SDT was strongly biased (e.g., Figure 8, images 4 and 9). However, for the cases where the uncorrected SDT showed

good results (Figure 8, images 1–3), the statistical correction causes bathymetry estimates to be less accurate by increasing the corresponding bias. The dynamical correction was more effective when the waterline was extracted from images collected at mid to high tides Figure 8; images 4–9) compared to at low tides (Figure 8; images 1–3), improving the RMSE values by 0.05 m on average. However, the estimates during low tides are less accurate (a reduction of 0.10 m on average).

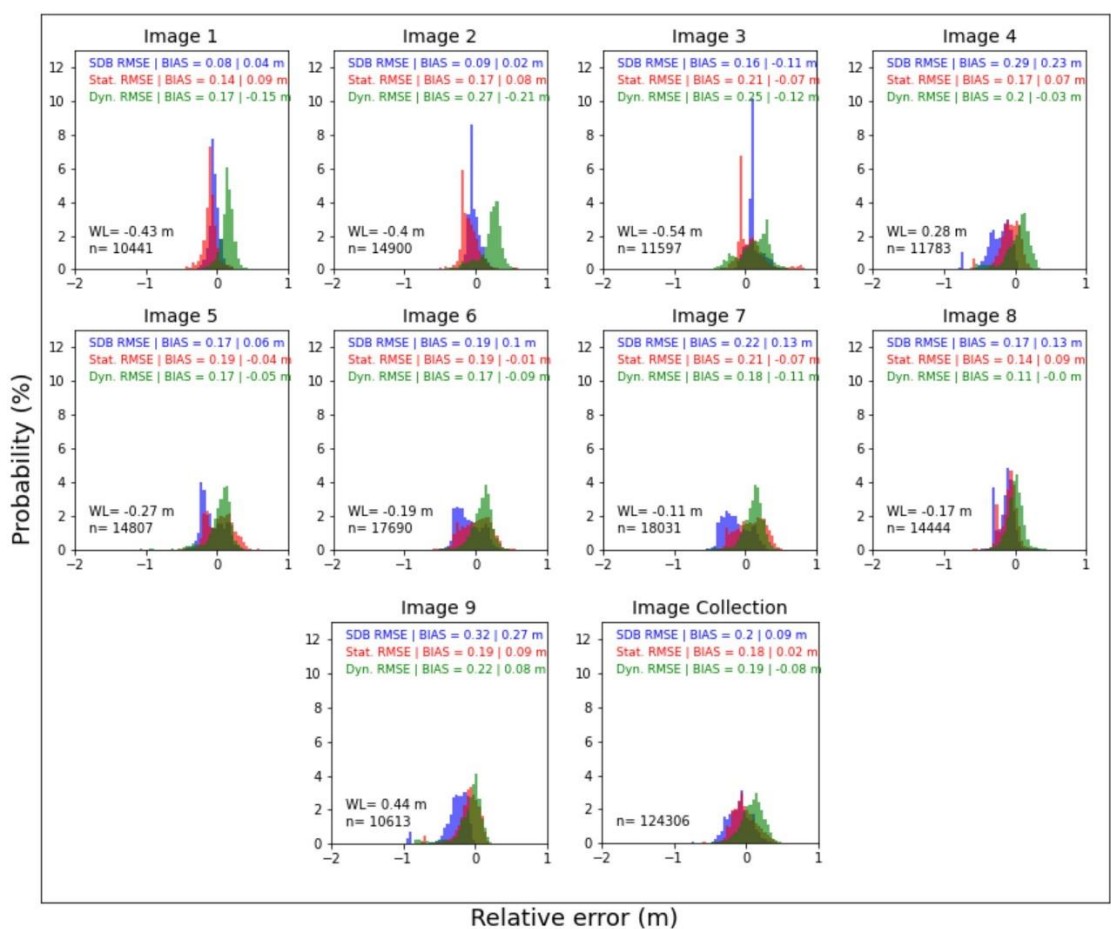


**Figure 8: Histograms of the waterline-derived SDT relative error (RE) for each image in the collection for Tauranga Harbour: waterline-SDT (blue), statistically corrected waterline-SDT (red) and dynamically corrected waterline-SDT (green). RMSE, BIAS, waterline height (WL), and number of waterline samples (n) are shown.**

### 3.4 Prediction of extreme water level using the SDB

The simulation scenarios showed that the combined use of SDB and SDT can be used to obtain water level predictions of similar accuracy to those predicted using only surveyed bathymetry (although the log-ratio method requires some in situ calibration data). To simplify the interpretation of the results, Figure 9 illustrates the average error parameters calculated when comparing the model output with the record of the three tide gauges. A detailed assessment of each of the gauges is provided

in Supplement D. In S4, the waterline-SDT for intertidal zones combined with the ratio-log-SDB for shallow waters can predict
the astronomical tide more accurately (RMSE ~ 0.07 m). In the S1 results, the model uses surveyed bathymetry (S1) with
poorer performance (RMSE ~ 0.09 m). S4 also performs better than S1 when assessed with the maximum absolute error
(MAXE), ~ 0.25 m and ~ 0.31 m, respectively. In addition, Figure S9 (Supplement D) shows that at the location of the
Ōruamatua tide gauge, the predictions were improved in the S4 scenario (RMSE = 0.05 m) compared to S1 (RMSE = 0.13 m).
Regarding the scenarios where SDTs replace only the intertidal topography (S2 and S3), the waterline-SDT (S2) provided
superior performance (RMSE ~ 0.07 m). The model that uses ratio-log-SDT (S3) showed poorer performance (RMSE ~ 0.09
m).

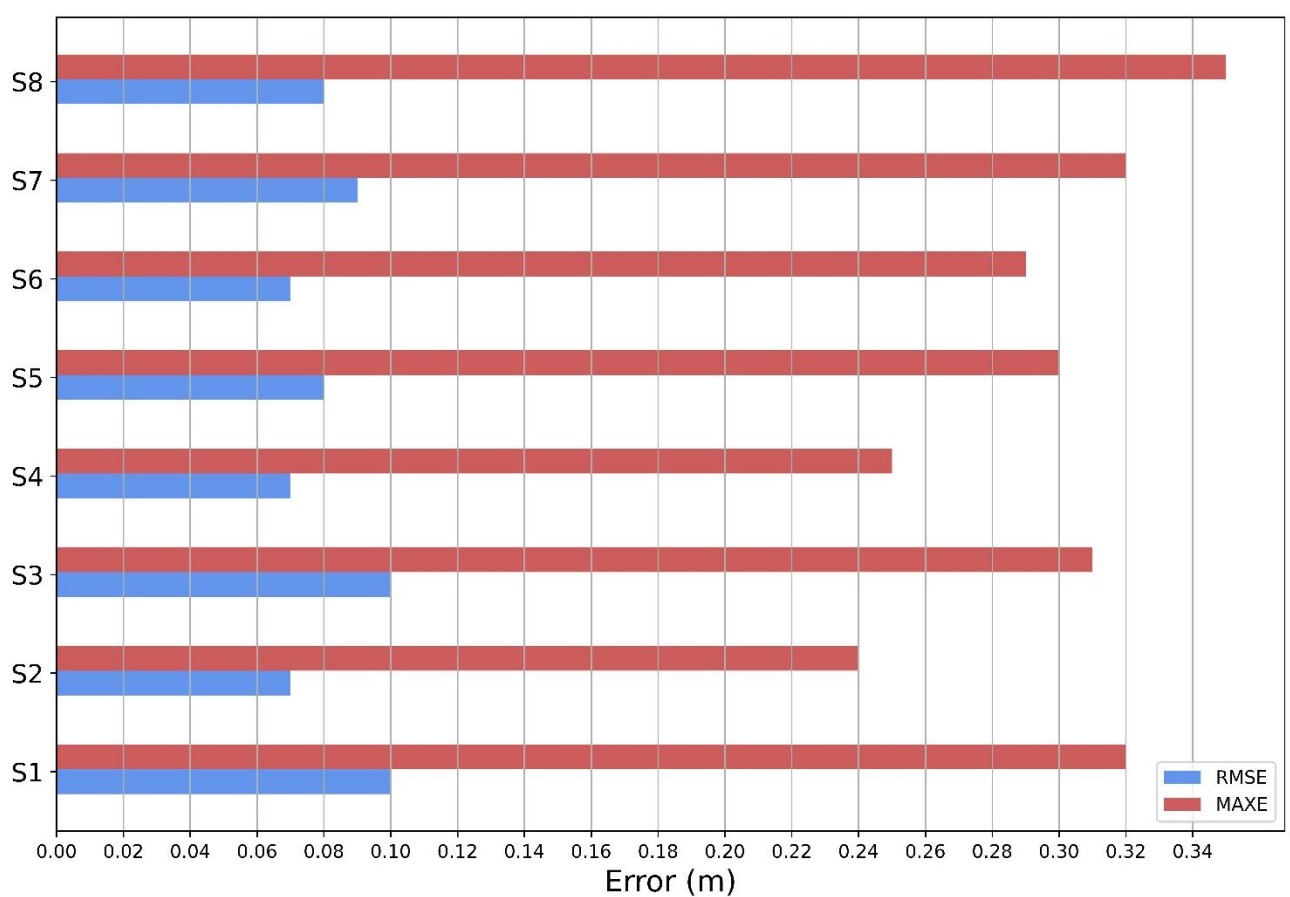

**Figure 9: The average parameter errors calculated considering the results at the three tide gauge locations (Ōmokoroa, Hairini, Ōruamatua) for each simulation scenario (S1, S2, S3, and S4) — RMSE (blue bar), MAXE (red bar).**

Similar accuracies were shown in the simulation tests using two distinct extreme events (scenarios S5–S8); see Figure S11.
The averaged error parameters for the scenarios using satellite-derived data (i.e., S6 and S8) are close to the ones for scenarios
using only surveyed topo-bathymetry (i.e., S5 and S7), as shown in Figure 9. The RMSE of these scenarios range from 0.07
m to 0.09 m, and the MAXE from 0.29 m to 0.35 m. At the time of the most extreme total water level, the scenarios using

surveyed topo-bathymetry and satellite-derived data showed agreement on their accuracy, with some differences depending

on which extreme event and the location analysed (i.e., Hairini and Ōruamatua). For the first event, scenarios S5 and S6 underestimated the total water level at Harini by -0.05 m and -0.07 m and at Ōrumatua by -0.33 m and -0.23 m, respectively (Figure S. For the second event, the scenarios S7 and S8 overestimated the total water level by 0.27 m and 0.21 m at Hairini, and by 0.07 m and 0.12 m at Ōruamatua, respectively. Note that the assessment here is not aimed at assessing whether the model accurately predicted the total water level but rather whether the satellite-derived topo-bathymetry prediction accuracy

is similar to that of the surveyed topo-bathymetry.

The good agreement between scenarios using surveyed topo-bathymetry and satellite-derived data is also shown over the entire model domain. For example, Figure 10 shows the difference at the maximum (a) and minimum (b) astronomical tide/ total water level at each grid cell over the entire simulation between the scenarios S1 x S4 (I), S5 x S6 (II), and S7 x S8 (III). The

differences are > -0.10 m and < +0.10 m at the maximum astronomical tide/ total water level (a). Figure 10 (a. II and III) shows that these differences intensify across the estuary when storm surge is considered in the model forcing. The north-western (i.e., above Ōmokoroa) and the south-eastern (i.e., Hairini and Ōruamatua) estuarine regions are the most affected, showing slightly stronger red colours for S5 x S6 (a.II) and S7 x S8 (a.III) than in S1 x S4 (a.I) comparisons. Major differences (≤ -0.10 m and ≥+0.10 m) in the astronomical tide/ total water level predictions occur in the estuary's inner channels at the minimum (low

tide) total water level at each grid cell (b).

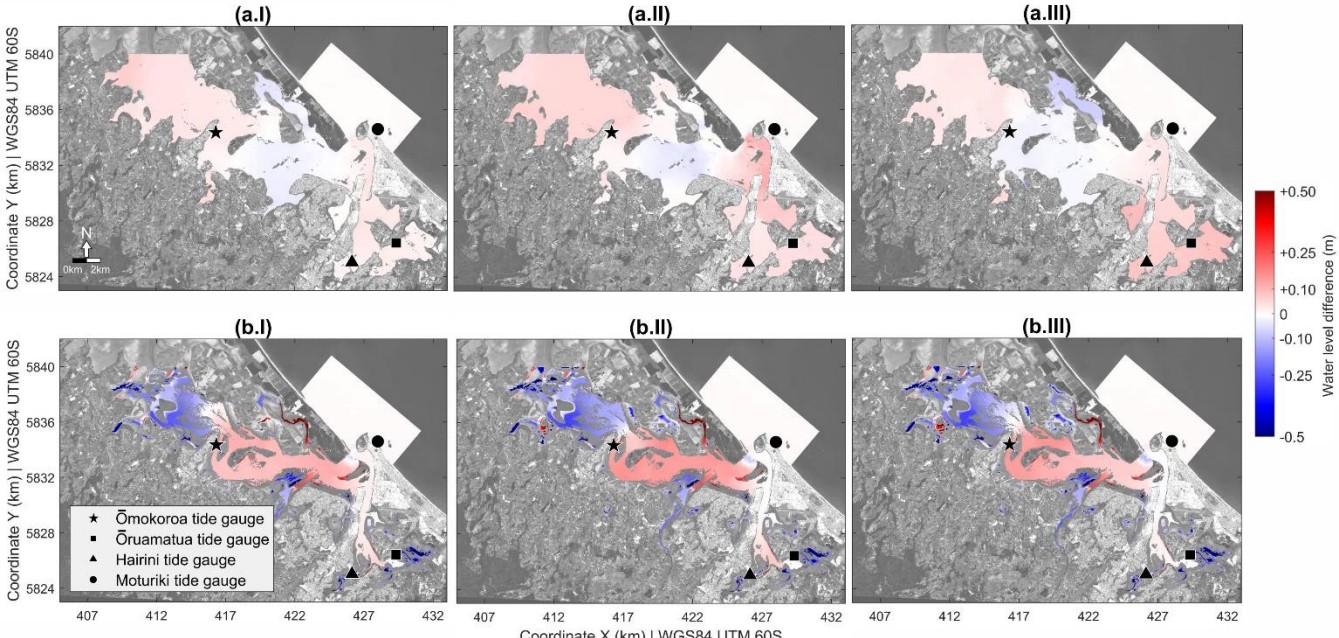

**Figure 10: Spatial difference between the hydrodynamic model output using surveyed topo-bathymetry (S1, S5, S7) and the waterline-derived plus ratio-log derived SDB (S4, S6, S8). The differences in the maximum (a) and minimum (b) astronomical tide/ total water level per grid cell over the simulation period are compared for S1 x S4 (I), S5 x S6 (II) and S7 x S8 (III). Red (blue)**

**colours represent positive (negative) differences, which means that the resulting water level from scenarios using only topo-bathymetric surveyed data (i.e., S1, S5, S7) is higher (lower) than the resulting water levels from the scenarios using only satellite-derived topo-bathymetric data. Background image: ESA Sentinel 2A. Date and time of the image acquisition: 18/12/2018 10:15 h.**

## 4. Discussion

### 4.1 The proposed waterline method for deriving topography from space-borne images and its limitations.

Topography estimated using the waterline method compares well to LiDAR data in our study sites — considering that the topography in the intertidal zone ranges between -1.12 m to +2.98 m relative to MSL (see Table 3) and the vertical and horizontal accuracy of the LiDAR data are 0.20 m and 0.60 m, respectively. Although it is hard to directly compare studies conducted in different coastal areas, our results show a similar bias to other studies performed in similar estuarine environments. For example, Salameh et al. (2020) applied the waterline method to Arcachon Bay in France, with the estimated DEM accuracy

of RMSE = 0.27 m; Bué et al. (2020) generated SDTs for Azinheira Estuary (Portugal) based on logistic regression with an RMSE = 0.6 m.

Despite the encouraging results of waterline-SDT in Aotearoa New Zealand's estuaries, the method is sensitive to the correct positioning and height-assigning of the waterline. Environmental conditions such as the complex morphology, varied bed

substrates, and groundwater seepage could reduce the accuracy of the waterline position. Also, the location of the tide gauge used to assign the waterline height is important. For instance, Maketū Estuary is a small estuary with complex morphology, and the tide gauge of Moturiki is located approximately 27 km from the estuarine entrance, which likely explains the low accuracy of SDT for that location. Furthermore, using just one tide gauge to assign the waterline height can add vertical error to the estimates because it does not account for the tide deformation and propagation in such a complex environment. Maketū

is currently undergoing staged engineering works to remove former flood protection, which could have caused changes to the bathymetry between images and after the LiDAR survey was undertaken.

Complex morphology affects the estimates differently over different parts of the topographic profile — waterlines closer to the MSL (water level ~0 m) are more accurate, and waterlines closer to the peak of the high and low tides are less accurate

(see Sect. 3.3). Our results corroborate those of Liu et al. (2013), who quantitatively analysed the waterline method in Dongsha Sandbank, China (an exposed coastal area). In their study, the authors have found that the main error source in the waterline method is linearly correlated to the slope and area of the intertidal zone. These can be directly linked to the tidal range (micro, meso and macro regimes). Assuming the same intertidal slope, the area of the tidal flats would increase with the tidal range, which would, in turn, require more images to adequately represent the topographic profile. Furthermore, having enough images

to characterise the morphology of the study site is a commonly limiting factor in the waterline method, as highlighted by previous studies (e.g., Liu et al., 2013; Salameh et al., 2019). Our results are also clearly affected by the number of images in

our collection. For example, gaps where no topographic data could be derived, can be seen between waterlines, shown in Fig. 7 (Sect. 3.2). Although the Sentinel-2 images are acquired every five days, they are often not useable due to cloud coverage.

The bed substrate can directly affect the waterline positioning, especially in Aotearoa New Zealand's estuaries, where clear water is common. For example, in Tauranga Harbour, the seagrass banks (Ha et al., 2020) and the groundwater seepage (shown to cause an error in water line detection in Huisman et al., 2011) can abruptly change the reflectance of the pixels around the waterline, especially in the centre part of the estuary, where the seagrass banks occur (Figure S3). The adaptive Otsu method (Nobuyuki Otsu, 1979) was used to detect the edges between water and intertidal zones. The method showed good performance

in determining the waterline location in estuaries, corroborating studies on lakes, rivers, water reservoirs (Donchyts et al., 2016), and coastlines (Vos et al., 2019). Other edge detection techniques were also tested in the present study,  such as by calculating the mean or the median of NDWI distribution following approaches in previous studies (Sagar et al., 2017; Bishop-Taylor et al., 2019). However, these did not perform as well (not shown). The Otsu method defines a threshold by detecting the value that maximises the within-class variance between two classes of a grey-scale distribution, which has two limitations:

first, the inability to correctly detect waterlines in images with complex conditions (i.e., where the water is clear, and the bed substrates reflectance can be seen in the satellite images); and, second, the Otsu threshold method will detect waterlines even when all intertidal pixels are flooded (at high tide) or exposed (at low tide), adding bias in the extremes of the topographical profiles.

There are currently several methods for edge detection that have been implemented in waterline-SDT that can potentially overcome the issues highlighted above. One practical solution would be the manual identification of the waterline; however, it is subjective and labour-intensive when applied over a large area and in multiple study sites. Another way would be to apply image segmentation techniques, for instance, K-means clustering techniques applied to edge detection (Salameh et al., 2020). Alternatively, the identification of sea-grass banks could be used to remove areas where the waterline is poorly detected prior

to analysis. Simple algorithms could be used for this, such as Ha et al. (2020), who identified seagrass using ensemble-based machine learning algorithms. Caballero and Stumpf (2020) identified algae and seagrass by using an empirical formula to calculate the maximum chlorophyll index, which uses three different optical bands to explain the radiance peak at the red-edge band.

**4.2 The proposed correction methods for waterline-SDTs.**

Our proposed correction methods (i.e., statistical and dynamic) for the SDT only resulted in a 1–2 cm improvement across the case-study estuary. However, our insights into why and where the correction resulted in improvements provide the basis for further work (e.g., when more imagery becomes available to test error sources more thoroughly). The statistical relationship between the error and the waterline height, the elevation on the tidal flat (LiDAR) and the waterline detection threshold in all four studied estuaries allowed us to set a semi-independent framework to correct the vertical level in the waterline-derived

SDT. For example, the relation between THLD and MRE could be learned in similar estuaries and subsequently used to correct to an entirely different study area with similar intertidal zone properties, such as estuaries with similar sediment colour, water turbidity, spring tidal range and intertidal area coverage.

The dynamical correction should give more realistic waterline heights because it accounts for the tide propagation within the
estuary. However, the approach did not significantly improve the SDT as expected. There are four possible reasons for the limited improvement. First, inaccuracies in horizontal waterline position may be more important than inaccuracies in the waterline height. This can be demonstrated by examining the location of waterlines on LIDAR-derived profiles. Figure 11 (m1) shows three different waterline positions (red, blue, and green lines) and three different LIDAR-derived profiles that intersect these waterlines (p1, p2, and p3) in Tauranga Harbour. In panels p1, p2, and p3, the three profiles are shown in detail.
The horizontal location of the three uncorrected waterline-SDTs is represented on the profiles by the coloured circles (red, green, blue) with their corresponding tide-gauge-derived heights (solid line). The corresponding dynamically-corrected waterline heights are also plotted, represented by the dashed lines. Note that in the dynamical correction, just the waterline height is corrected, and the observed waterline position remains unchanged. Because the numerical model is expected to give more realistic water levels — that account for the tidal propagation within the estuary— the correct position of the waterline
should be where the dynamically corrected waterline height (dashed lines) intersects the topographic profile (continuous black line) in the Figure11 (panels p1, p2, and p3). Thus, all the waterlines in p1 would need to be further seaward than they are to intersect with the profile at the correct location. In p2, waterline 3 (blue circle) should be slightly landwards, and waterline 1 (green circle) and 2 (red circle), further seaward. In p3, all corrected waterlines should be further seaward. However, when the

waterline is well positioned, waterline heights are similar to the LiDAR data; for instance, in p2 for all waterlines (all dashed

lines).

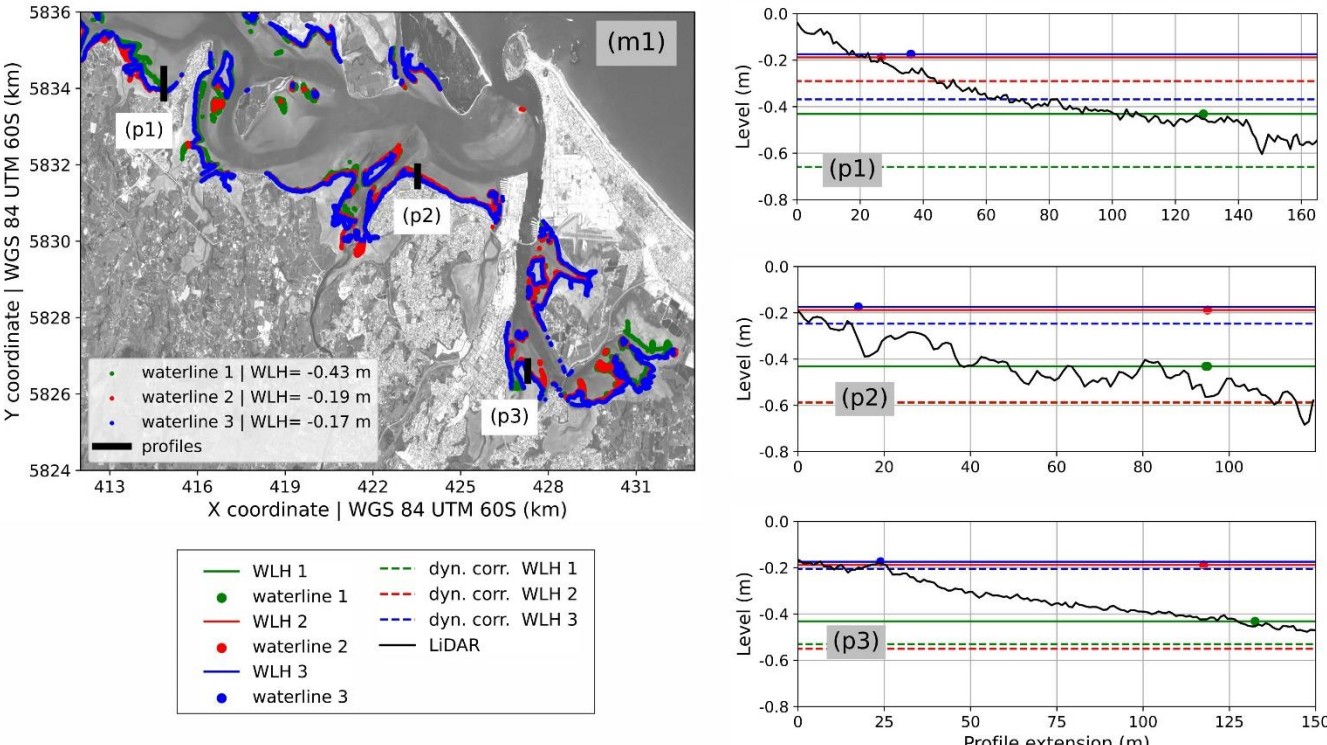

**Figure 11: Analysis of the dynamical correction in three different profiles (p1, p2, p3). [m1] shows the location of the LIDAR-derived profiles in Tauranga Harbour. The panels (p1), (p2), and (p3) show the location of the three waterlines along the three profiles (black**
**line). Each panel highlights the waterline height (WLH) extracted from the tide gauges and the horizontal position of three different waterlines (green (waterline 1), red (waterline 2), and blue (waterline 3)). The corresponding dynamically corrected waterline height (dyn. corr. WLH) is also shown (dashed lines). Considering that the hydrodynamical model should give more accurate water levels, the correct waterline positioning should be where the dyn. corr. WLH intersects the LiDAR profile. Background image: ESA Sentinel 2A. Date and time of the background image acquisition: 18/12/2018 10:15 h.**


The second factor that can influence the performance of the dynamical correction is the hydrodynamic model accuracy —

especially at low tide, as can be seen in Figure 10. The spatial resolution of the numerical grid (20 m) can limit the model's

ability to correctly solve the flooding and drying within grid cells around narrow channels, potentially adding horizontal bias

to the waterline heights. Furthermore, the tidal wavelength is hundreds of kilometres, which means that the water level should

not be affected significantly by smaller-scale bathymetric features. Moreover, the fourth limiting factor is the LiDAR data

horizontal and vertical accuracy, which limits the potential correction. The use of hydrodynamic modelling to assign the

waterline heights is a common practice when generating SDTs (e.g., Liu et al., 2013; Khan et al., 2019; Salameh et al., 2020;

Fitton et al., 2021). However, in most cases, the studies cover extensive areas with exposed coastlines or sand banks, in which

regional tide models are used where there are not enough tide gauges to provide tide levels (Liu et al., 2013; Khan et al., 2019; Fitton et al., 2021). In just a few of these studies, enclosed estuaries were studied by setting up local-scale hydrodynamic modelling (e.g., Liu et al., 2013; Salameh et al., 2020). In Arcachon Bay (France), Salameh et al. (2020) compared the waterline-SDTs that were generated by assigning waterline heights according to single-location tide gauges, single-location model point output, and grid hydrodynamic model output. Similar to our results, they found that the waterline heights assigned by using grid model output did not improve the SDTs compared with using a single-location tide gauge within the estuary. They explained the unexpected result as due to the slope of the tidal flat and the model's inability to provide accurate sea level heights over the intertidal area. Similarly, Liu et al. (2013) used a regional tide model for the South Yellow Sea (China) to assign waterline heights to a local-scale study in Dongsha Sandbank (an exposed tidal flat), which limited the vertical accuracy of the SDT up to 30 cm (corresponding to the model's accuracy).

**4.3 Comparison between the waterline method and ratio-log for intertidal zones.**

The results show that, for Tauranga Harbour, the waterline and the ratio-log techniques performed similarly for the task of deriving topography over intertidal zones using satellite images. Thus, for estuaries with low water column turbidity, pre-existing surveyed topo-bathymetric data, and low numbers of available satellite images covering its area — as is the case of Tauranga Harbour — the ratio-log method could potentially replace the waterline method for deriving elevation data for intertidal zones. Although the waterline method shows better performance when considering the RMSE — either evaluated on a point-to-point basis (0.20 m) or evaluated using the DEM (0.23 m), see Table 3 — than the ratio-log method (0.25 m). Evaluating RMSE using the DEM provides more information for comparison. Figure 12 shows the density SDT points and distribution of the relative vertical error (RE) for Tauranga Harbour's waterline-SDT and ratio-log-SDT for intertidal zones, where the colour represents positive (red) or negative (blue) errors. Positive (negative) errors indicate that SDT estimates are deeper (shallower) or further landward (seaward) than the LiDAR data (see Sect. 2.7). The waterline-SDT (Figure 12: a1, b1, c1, d1) provides estimates that are generally shallower or further seaward than the LiDAR — as the negative RE indicates — with the worst estimates in the tidal flat's upper region (bluer colour dots). The positive RE values (redder colour dots) are concentrated in the estuary's wide flat centre region (Figure 12 b1) and indicate that the estimates are deeper or further landward than the LiDAR data. As discussed in Sect. 4.1, the waterline method is mainly limited by the number of images required to properly define the morphology of the study site. In the case of Tauranga Harbour, as a consequence of the high complexity of its morphology, the SDT provided by the present framework could be substantially improved with more images, making the waterline method even more accurate than the ratio-log method.

The ratio-log-SDT (Figure 12: a2, b2, c2, d2) allows the water depth to be assessed on a pixel-by-pixel basis, with a resolution of 10 m in the case of Sentinel Copernicus data. However, the application of this method has several limitations. In our application, the intertidal topography was flattened; the positive and negative errors are in the upper and lower parts of the tidal flats, respectively. In the middle of the topographic profile, the estimates are more accurate (whiter colour dots). The low

data variability (pixels reflectance) probably causes the lower accuracy of the ratio-log-SDT in comparison to the waterline-SDT (the ratio-log method depends on calibration data covering the range of conditions). The ratio-log (green/blue band ratio) poorly explains the depth (LiDAR data), which leads to a low correlation coefficient ($R^2=0.12$). This assumption is confirmed by the higher correlation coefficient ($R^2=0.24$) obtained when the same method is applied to the shallow waters within Tauranga Harbour. In addition, the presence of seagrass in the intertidal zones and shallow water (Figure S3) can potentially worsen the ratio-log-SDT and SDB by affecting the reflectance (Geyman and Maloof, 2019; Caballero and Stumpf, 2020).

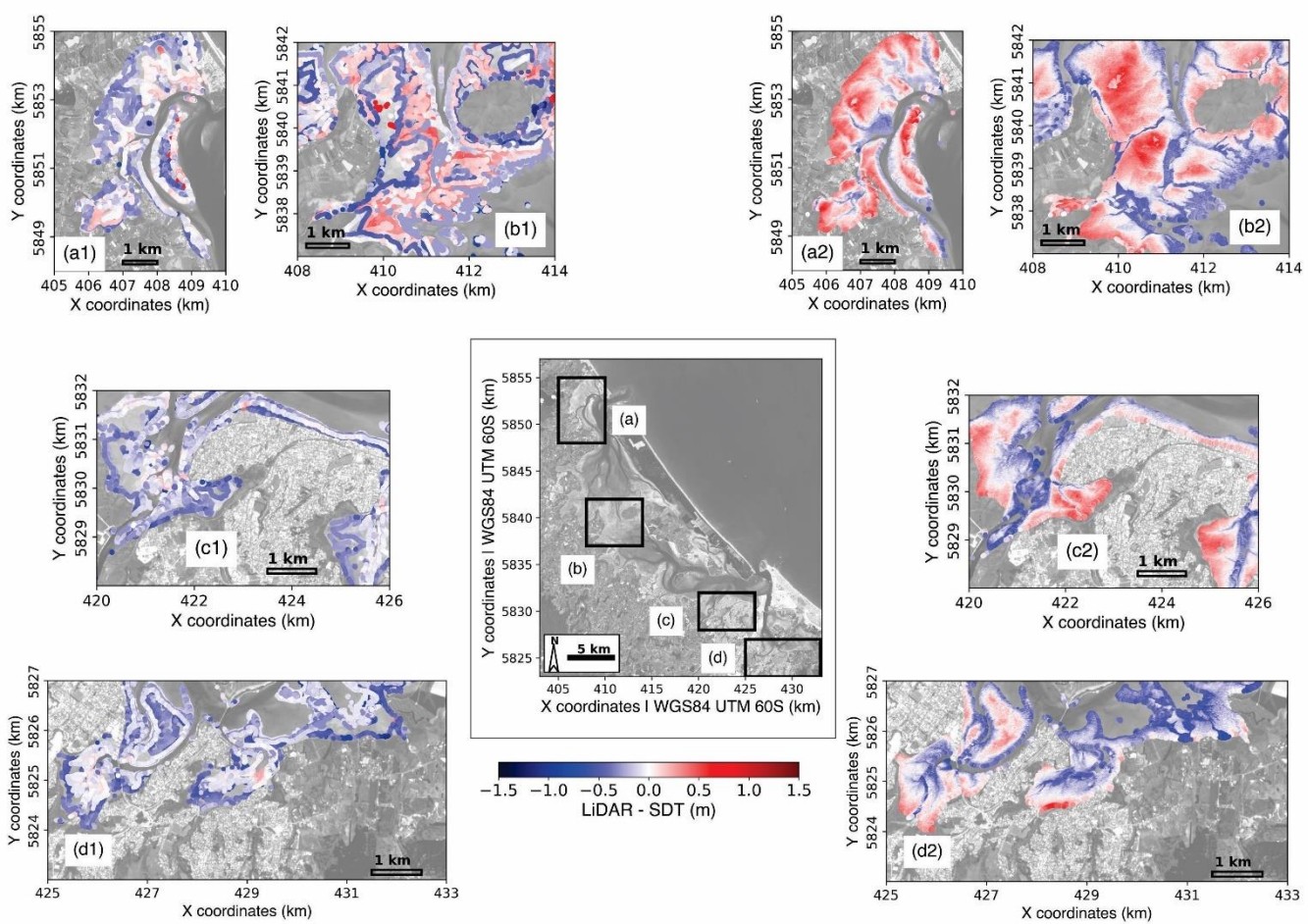

**Figure 12: Estimated SDT and corresponding relative vertical error (LiDAR-SDT) for the intertidal zone in Tauranga Harbour using waterline-derived (a1, b1, c1, and d1) and ratio-log (a2, b2, c2, and d2) methods. The root-mean-squared error for the waterline method is 0.20 m, and for the ratio-log method is 0.25 m (not shown in the figure). However, the waterline method results in less density of estimates (due to imagery constraints), while the ratio-log method results in a pixel-by-pixel estimate density. Background image: ESA Sentinel 2A. Date and time of the background image acquisition: 18/12/2018 10:15 h.**

Many estuaries have turbid water, which would reduce the quality of SDBs and SDTs derived from both ratio-log and waterline methods. The ratio-log derivations would be affected by the interference of the suspended material on the light absorption rate through the water column — which could be improved by using methods that adjust the ratio-log method for turbid water (e.g.,

Caballero and Stumpf, 2020). The waterline derivations would be affected by intertidal zone identification. The NDWI of the pixels in shallow water with a high concentration of suspended materials could have similar values to those in the intertidal zone. Consequently, determining the intertidal areas would be less accurate in environments with high concentrations of suspended material.

In addition, image pre- and post-processing are other factors that may improve the accuracy of the SDT and SDB of waterline and ratio-log methods. In the present manuscript, available Sentinel images were used, which are already pre-processed by using Sen2Cor, which creates surface-reflectance images (see Sect. 2.1). However, several pre-processing tools are available (Pereira-Sandoval et al., 2019). Some of these are designed specifically for use in coastal areas — where water is often turbid, containing a high concentration of suspended sediments and other materials. For instance, the ACOLITE tool (Vanhellemont and Ruddick, 2018) has been widely applied in estuaries (e.g., Bué et al., 2020; Salameh et al., 2020; Fitton et al., 2021). In the case of exposed coastal areas or where local wind waves can increase the rugosity of the water surface, filters to eliminate sun glint can be applied (Hedley et al., 2005).

## 4.4 Hydrodynamic modelling assessment

The bathymetric and topographic data quality is fundamental for reliable hydrodynamic modelling. Despite the limited accuracy of the SDT and SDB (see Sect. 4.1–4.3), our results show that hydrodynamic models using satellite-derived elevation can predict water level with similar accuracy to models using only surveyed data (see Sect.3.4, Figures 9 and 10). Thus, water level modelling may not be sensitive to small uncertainties in the bathymetric data but rather to the larger scale characteristics of the estuary, such as the width of entrances and overall geometry. Bathymetric uncertainties can arise from the interpolation technique used to create the DEMs (Circus et al., 2000; Kang et al., 2017, 2020; Salameh et al., 2020) — e.g., spline, kriging, inverse distance weighting, nearest neighbour, triangulation. However, previous studies have found that uncertainties in the elevation data lead to minor differences in the water level predictions (Cea and French, 2012; Falcão et al., 2013). For instance, Cea and French (2012) showed that water level predictions do not significantly change with vertical uncertainties of up to 1 m in the bathymetry. Similarly, Falcão et al. (2013) have shown that the DEMs created with the same interpolation technique (i.e., kriging) but with a different spatial resolution (i.e., 5 and 50m) did not significantly affect the water level prediction. Corroborating our results, Falcão et al. (2013) also found that the worst predictions are for grid cells where the water level is at a minimum when comparing these two scenarios. The stream current magnitude and direction predictions are affected the most by the uncertainties in the bathymetric and topographic data (Cea and French, 2012; Falcão et al., 2013).

Despite the uncertainties in the estimates, SDT and SDB can generate a fair approximation of estuary relief, which can be helpful in long-term predictions for coastal management applications. In idealistic numerical studies, the extension and slope of the intertidal zone, the estuary's length, and the width of the mouth are the main factors causing changes in the tidal range within harbours ( Du et al., 2018; Khojasteh et al., 2020; Khojasteh et al., 2021). For instance, Du et al. (2018) show that the

length of an estuary and intertidal zone slope strongly influences the tidal range. However, the entrance restriction drives the estuarine response to sea-level rise (Khojasteh et al., 2020); the smaller the cross-sectional area of the estuary mouth, the smaller would be the tidal range within the estuary. Moreover, SDT and SDB could be used for data assimilation in numerical modelling, as in Mason et al. (2010), who used the SDT to calibrate a morphodynamic model. Ultimately, the SDTs and SDBs can decrease the uncertainties of flood-risk management in the present and future scenarios of sea-level rise where studies are limited due to a lack of elevation data in remote areas such as small islands in developing states (Parodi et al., 2020) and coastal lagoons in developing countries (Pedrozo-Acuña et al., 2012).

Although the differences in the resulting water level between the SDT, SDB, and surveyed bathymetry simulation scenarios show that satellite techniques compare well, our simulations were only conducted in one estuary, albeit a large and relatively complex estuary — where the astronomical spring tides are the main driver for estuarine flooding. Therefore, studies are required in sites with different physical conditions would be useful to validate the use of SDT and SDB more broadly. For example, estuaries where the storm surge is the main driver for flooding; or/and exposed estuaries where the wave forces can increase the water level (i.e., wave set-up); e.g., Bertin et al. (2019). Many such effects can be strongly dependent on the topography, for example, the wind effect on the storm surge (wind surface stress) and generation of local waves (e.g., Smith et al., 2001; Bertin et al., 2015) and tide-surge interactions (Spicer et al., 2019; Wankang et al., 2019; Zheng et al., 2020).

## 5. Conclusions

A waterline technique for deriving topography from multispectral satellite images was developed, and its use in hydrodynamic modelling was assessed. The simple pre-processing required for the satellite images combined with the use of cloud computing and storage make the present framework highly applicable to regional-scale studies. Our main findings show that the accuracy of the waterline SDT is similar or even superior to other techniques applied in previous studies to comparable sites and similar to the vertical error in the LiDAR dataset used to assess accuracy. In addition, optical empiric methods such as the ratio-log could potentially replace the waterline-SDT in case of imagery constraints and when applied to an estuary with low water turbidity. The main source of error for the waterline-SDT is whether the number of satellite images is adequate to cover the intertidal zone and the accuracy of the waterline positioning. Statistical and dynamical corrections were trailed but provided limited improvements. The hydrodynamic modelling assessment was encouraging and showed that SDT and SDB techniques have the potential for use in predictions for extreme water levels (such as those associated with spring tides and severe storm surges). Scenarios using different applications of the SDT and SDB did not show major high water level differences over most of the numerical domains for Tauranga Harbour. The use of SDT and SDB for hydrodynamic modelling in estuaries can make flooding assessment for remote coastal areas feasible and provides a pathway around the need for expensive surveys for economically depressed vulnerable areas.

**Code availability**

The codes used in this work are available as Python notebooks at https://github.com/CostaAndCoasts/Intertidal-zones-satellite-derived-bathymetry.

**Credit authorship contribution statement**

Wagner L.L. Costa: methodology, data analysis, writing – original draft, visualisation. Karin R. Bryan: conceptualisation, supervision, writing – review & editing, resources, funding acquisition. Giovanni Coco: Writing – review & editing, supervision.

**Competing interests**

The authors declare that they have no conflict of interest.

**Acknowledgement**

The authors would like to thank Dr Ben Stewart for numerical modelling assistance. This work was supported by the National Science Challenge: Resilience Challenge "Coasts" programme, GNS-RNC040. Data were supplied by Land Information New Zealand (LINZ), Bay of Plenty Regional Council, and Waikato Regional Council.

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
