# Peer review of "Modelling extreme water levels using intertidal topography and bathymetry derived from multispectral satellite images"

_Natural Hazards and Earth System Sciences, 2021_

## Author Comment (AC1)

**Responses to Review Anonymous Referee 1**

Dear Reviewer anonymous referee 1,

Thank you for your observations regarding our preprint. Your suggestions helped to improve our manuscript. See below our responses marked in blue for each of your questions (marked in black).

The manuscript presents an interesting approach to deriving intertidal bathymetry from the waterline method through multispectral images, covering four (4) estuarine study areas on the east coast of Aotearoa New Zealand's North Island (Tauranga, Ohiwa, Maketu and Whitianga harbour). It represents a current thematic area, and it can be particularly useful to be applied in remote or inaccessible areas or where the bathymetric or cartographic data is very outdated. The main objectives of the study are to determine if multispectral images can be used to extract accurate intertidal bathymetric area and to assess the use of the SDB for hydrodynamic modelling of estuarine.

Good English level however the manuscript is not well-structured, quite confusing and the reader easily misses the main guidelines and the aim of the study. In section 1 (Introduction) is very difficult to establish a connection between the different ideas and paragraphs. A deeper revision of the state of the art is needed to bring the reader into the SDB theme and waterline method. The flow chart in chapter 2 is useful but does not really explain the methods used by the authors. Furthermore, the Methods Chapter establish that the main method was divided into 2 steps (1-SDB estimation and 2-Hydrodynamic modelling assessment) and that step 1 is also two methods for removing the bias, but a clear explanation of the methodology is not present in this section of the manuscript. A very short discussion and a shorter conclusions section are shown, where no clear main findings can be found. Modelling Storm surge is only referenced in the title of the manuscript.

The manuscript shows that a lot of work has been made, however, a big gap throughout the presented structure is noticed and the methodology used is not well described, creating a lot of misunderstanding between the methods applied and the different steps described by the authors. I, therefore, do not recommend the publication of this manuscript as it was presented. A major revision of the structure and methodology form is recommended.

Based on your revue and those provided by the other reviewers, we understand that we need to modify the structure of the paper because the current format is confusing. We have undertaken a deeper revision of the state of the art on SDB (adding new references and text to the introduction section). We have added a much clearer aim to the introduction, and worked on linking the methods to the aim in a much more clear and logical order. In the methods section, we added further explanations about the different methods implemented to remove the bias (i.e., statistical and dynamical methods). In the discussion and conclusion sections, we built further on the context provided in the new references added in the introduction part. Please note that we did not model the storm surge, but we analyzed the maximum astronomical tide in all simulation scenarios and compared the outputs between scenarios using only surveyed bathymetry, only SDB, and mixed surveyed bathymetry combined with SDB. In terms of coastal flooding, the maximum water level is the main parameter studied and in most places in the world, the water level is dominated by the tide. In summary, we are happy to modify the paper structure as you and other reviewers recommend.

My main critics are the following:

1. In the Introduction section the theme is not quite explained, and only part of the aim of the study is presented in the last sentence of the last paragraph. In this section is expected that the authors explain the reasons that have motivated this study, as well as what will be presented in the different sections of the entire manuscript.

We will add a motivation paragraph linking the importance of SDB to enable hydrodynamic modelling of water level variations in shallow intertidal estuaries, and how and why it can be difficult to obtain information on bathymetry using traditional surveying methods. We will also add sentences providing an overview of the material presented in each of the following sections of the manuscript.

2. The different figures do not follow a consistent presentation. The geographic coordinates in some cases are presented as latitude/longitude with no reference datum associated (Fig.2(a)); others as X/Y coordinates (km) WGS84/UTM60S (Figs. 2 (b), (c), 3 (a)) and even other examples as X/Y coordinates NZGD2000 (km).

All figures will be changed so that they use the same SI units.

3. The same Figures, presented in different sections, have different SI unit references, like Fig.2 (b) and (c) are expressed in X/Z coordinate (km) and Figures S1 and S2 in X/Z coordinate (m) – show a lack of consistency.

All figures will be updated to use the same SI unit references.

7.   The areas A, B, C and D depicted in Figure7 (central figure) are not quite perceptible, and the small figures (a1, a2, b1, b2, c1, c2, d1 and d2) do not have geographic coordinates associated, neither the scale factor.

We will modify the colour scheme and sizes to make the Figure more readable. We will make the ABCD areas easily visible. We will also add the coordinates and a scale factor.

5. The profile lines drawn in Figure 9 (m1) are barely noticed. Maybe the authors could choose a different colour palette.

The colour and the thickness of the profile will be modified to make it more legiable.

6. In the text, the figures are not correctly cited, like Fig 2A (line 85); Fig 2B and 2C (line 88). In the Figures, the panels are mentioned with small letters (a, b and c), as well as in the figure capture.

We will change all the labelling to the format of the journal (always using small case letters).

7. The data access information at the reference links (lines 94-97) is missing.

We will add the links.

8. I do not understand how the intertidal area is identified, the method is not well explained. Is used the tidal level at the time of the acquisition of each image? Or is used an average tidal range (tidal amplitude?) for all the images. Is also not clear the tidal level for each image, as depicted in Figure 2(d). All images are used to generate the intertidal area presented in Figure 3(a)?

The intertidal area is identified by calculating the standard deviation of NDWI over the whole collection of images, at each pixel (this is described in section

2.2, using equation 1). Because the water level changes substantially through time (because the tide completely drains and inundates these areas), these areas are easily identified, in a collection of images, by the high standard deviation of NDWI. We use a threshold to find the areas with high standard deviation and define the 'intertidal' area as being the area of high NDWI standard deviation. We will make this more clear in the text. We did not use the level of the tide to demarcate the intertidal region, because this would require apriori knowledge of the elevation of the intertidal. (The tide level is not used to generate Figure 3, only the satellite reflectance). We have added the level of the tide to Figure 2, and also marked the times that these two images were collected on the timeseries shown in Panel D.

Once the intertidal region of interest is demarcated, then the x,y coordinates waterline is extracted separately from each image in the correction, using the NDWI to determine the location between wet and dry pixels. The threshold between wet and dry pixels is determined using the Otsu threshold. In the final step, the x-y coordinates are associated with a water level height (z), which we assume to be equivalent to the level of the tide when that image was collected. The x,y,z coordinates from all the images are then collected into one dataset, which is then gridded to make a bathymetric map. We will make this much more clear.

9. The threshold value used, and all the contour extraction method (lines 153-159) are quite confusing. And which values of threshold and water level were used for the other study areas, regarding that Figure 4 presents the water level and threshold values for each image. A table with this information, for all the different study areas, as supplementary information could be very useful.

We will make this more clear (see response to comment #8). A table will be added containing the information in the supplementary material.

10. What do you mean with the Stumpf-ratio method was applied for deeper areas (lines 164-165). The Stumpf ratio method (Stumpf et al., 2003) is not quite good for all different benthic areas and for very deeper areas. What was the maximum depth value which the authors have used this method?

The waterline method can only be applied in the intertidal zone. When we refer to deeper areas, we refer to all areas within the estuary that are not intertidal zones or land. We are aware of the Stumpf-ratio limitations. One of our main

aims is to determine whether, despite the limitations of both waterline and Stumpf-ratio methods, we can predict reasonably the water level (using numerical model). We will rewrite the aims to make this more clear, and to review the limitations of the Stumpf method.

11. It was not explained by the authors all the pre-processing steps applied to the multispectral images, such as sun glint correction (for example Hedley, J.D.; Harborne, A.R.; Mumby, P.J. Simple and robust removal of sun glint for mapping shallow-water benthos. Int. J. Environ. 2005, 113, 2107–2112). If this step was considered, it should be enunciated in the manuscript. The authors described that Level 2 image was used, with BOA values corrected for the effects of the top-atmosphere (lines 103-104), but it was not explained why they used these images rather Level 1 with atmospheric correction.

We used the level 2 images because they were already corrected for the effects of the top of the atmosphere, and we did not believe it necessary to undertake our own correction. We did not apply a sun glint correction because the Otsu algorithm could detect the waterline well without this correction.

12. I can not understand if the evaluation of the model performance in section 2.4 is one of the results of this study. And if they are, why not present them in the results section? Lines 201-215 have a challenging interpretation.

We can move this to the results. We will make the explanations on these lines more clear.

13. Why an" Average" line in table 3. Does not make sense.

The "average" refers to the average of the error metrics over all the estuaries for the corresponding parameter (MAE, RMSE, R2).

14. Lines 234-239 should be included in the Discussion section, not here, where the results are presented.

We will move the referred lines to the discussion section.

15. In the ESA Sentinel 2A images used as background in several figures are missing the data acquisition time and the water-level information (Figures 2, 7, S1, S2, S3, S5, S6, S7).

We will add the information required to the figures.

16. The authors cannot quantify as good or strongly correlated/related the R2 achievements (lines 242-245). Why R2=0.70 should be considered as strongly related? Once more the authors are discussing the presented results in the Results section, and it is a recurrent procedure throughout this section. Perhaps if the authors had previously described in section 1 the contents of each section, the reader could understand better the manuscript. The structure of each section is quite confusing.

We will describe the contents of each section in the introduction part as required. We will also make sure that each section is started with a clear topic sentence.

17. Lines 274-278: R2 values assumption/classification (low/higher). And R2 is referred to as the coefficient of determination (line 276) and a coefficient of correlation (line 278) in the same paragraph. Is not coherent.

We will change it.

18. The authors do not explain why the results and the application of the methodology were only presented for one study area (Tauranga Harbour). They are free to do it, even for editorial figures or pages restrictions, however, this fact should be mentioned and explained in the manuscript and the main results for each area should be resumed (table format perhaps) in the supplementary material section.

The results were presented for just one study site because of limitation number of pages/figures, and we have just Tauranga Harbour with a numerical model already validated. The results of the SDB estimates for each estuary are presented in the main manuscript (table 3) and in the supplement material (Figures S5, S6, and S7). We will change the text to note this explicitly.

19. What is the spatial grid resolution value (line 298)? Is 20 m as assumed in line 336?

The grid resolution we applied here was 20 m. We have added this to the text.

20. Lines 313-316 are quite confusing. A better explanation is needed.

We will change the sentence to: "Otsu threshold works well to bimodal distributions (when dry and wet pixels are presented in the image), however,

the algorithm is limited when the distribution is not bimodal (when most of the pixels are dry or wet)."

21. The prediction of water level using the SDB is presented in section 3.4 for the 3 tide gauges (Omokoroa, Hairini and Oruamatua) (lines 323-331). The average error parameter presented is for each tide gauge and Figure 10 shows the average between all the tide gauges. Was this methodology that was used? I am confused.

We meant the average values between the three tide gauges records and not the average values for each tide gauge. We will reformulate the sentence to make it clearer.

22. Can I assume that, for lower tide values images, the presented methodology can not be used? Or only for the Stumpf ratio method application (SDB)? Lines 338-341. The Stumpf ratio method can not be directly applied to intertidal areas, exactly due to the image reflectance of the dry pixels (low water level images).

The model results for lower tide simulations showed the worst results. Thus, yes, it is not recommended to use SDB if the focus of the work is the processes occurring around the low tide, however for purposes of coastal flooding studies, usually, the highest values are of interest. The STUMPF-SDB is calculated using an image acquired at high tide. It is shown in the Methods (section 2) and in the supplementary material (Figure S2). We will make this more clear in the general discussion.

23. What represents the rectangle-shaped figure in Figure 11? Survey bathymetry data or LIDAR data?

The bars represent the comparison between the in-situ observed water level at each of the 3 sites where there are water level sensors, and the hydrodynamic model output at those same sites, where the hydrodynamic model is run using 4 different bathymetries (S1, S2, S3, S4, are described in the text). We will change the figure caption to make it more clear.

---

## Author Comment (AC2)

**Responses to Review Anonymous Referee 2**

Dear Reviewer anonymous referee 2,

Thank you for your observations regarding our preprint. Your suggestions helped to improve our manuscript. See below our responses marked in blue for each of your main concerns (marked in black).

1- The manuscript presents a novel methodology to deriving intertidal bathymetry for four estuaries in New Zealand (Tauranga, Ohiwa, Maketu and Whitianga harbour) characterized by a complex morphology. I find this thematic interesting, as it allows to update and improve the boundary conditions of regional numerical models. However, I think the structure and writing of the manuscript require further work to reflect all the work done. The manuscript needs a better use of English, a restructuring of the chapters and above all to emphasize the purpose of the work as well as the authors' motivation and innovations. Therefore, I do not recommend the publication of this manuscript as submitted. This review is critical, nonetheless the authors have the potential to have a great manuscript and I would like to encourage them in their progress.

We appreciate that you found our manuscript interesting, despite the problems regarding its structure. The inadequate structure is a problem highlighted by all the 3 reviewers. We intend to modify it accordingly. For instance, we plan to describe the content of each chapter at the end of the introduction section and set the paper up with more clearly presented aims.

2 - I mainly concern of the reasoning and the reading flow, which is quite confusing and the reader can easily miss the guidelines of the study. Section 1 does not clearly show the developments achieved by the scientific community, the relevance of the chosen methodology and, above all, the authors' motivations.

In the introduction section, we showed the main techniques used in the manuscript, and their advantages and disadvantages. However, we recognize that we could improve the introduction and will re-review of the literature about SDBs and add newly-available literature. We will also build the introduction more clearly toward our aims.

3 - Section 2 is very long and presents too many technical concepts, and even results, that is hard to follow how they were implemented. The study area

should be expanded with a description of the main processes describing water level dynamic, since the work's title mentions storm surge modelling.

We will move model validation and put it into the results. We understand that some terms used in section 2.5 can lead to confusion; thus, we propose the re-writing of section 2 to make it clearer. Regarding the expansion of the section to improve the description of the water level dynamics, we think this would be an excellent addition to the discussion.

4- Results and figures in Section 3 present a lack of consistency of SI units, authors should homogenize them. I think criteria presented in Fig. 3 and 4 are unclear and need a deeper discussion. Errors should be accompanied by their percentage for better interpretation. Unfortunately, the color map chosen for Fig. 7 and 9 is not good for presenting such significant results. The conclusions are extremely short and not summarize the reasoning of the work.

We will use consistent SI units in all figures. We understand that the explanation of the identification of the intertidal zone and the waterline position is still unclear; we propose to add more information to the paragraphs in relation to Figures 3 and 4. We will provide a more clear presentation on the range of the depths within the intertidal zone so that the error presentation is much clear. The colour maps of Figures 7 and 9 will be changed. We will add more detail to the discussion and conclusion; in particular, we will add more on the hydrodynamic modelling results and new references about SDB techniques.

---

## Author Comment (AC3)

**Responses to Review Anonymous Referee 3**

Dear Reviewer anonymous referee 3,

Thank you for your observations regarding our preprint. Your suggestions helped to improve our manuscript. See below our responses marked in blue for each of your main concerns (marked in black).

1. I have read the paper entitled "Modelling tides and storm surge using intertidal bathymetry derived from the waterline method applied to multispectral satellite images" by Costa et al. The study aims to determine whether satellite imagery can be used to extract accurate intertidal bathymetric data; and assess the use of the SDB for hydrodynamic modelling of estuaries. The paper is interesting, and one can see that quite a lot of effort, based on the complex methodology, was put into it. However, the manuscript is quite confusing making it difficult to understand. Part of the problem I had was regarding the use of the term SDB to represent extracted shorelines when the term is coined to deriving bathymetry. The study seems to have its merits but needs a complete re-structure. I found results difficult to understand.

We are pleased that the reviewer finds our manuscript interesting. We are aware (based on your review and others) that the structure of the paper is not ideal, and the structure makes the paper confusing to read. We intend to re-structure the paper according to the reviewers' comments. Regarding the methods implemented, there is a misunderstanding. The identification of the waterline position is done as part of the waterline method for estimating bathymetric data in the intertidal zone. We will make this more clear.

2. Because it presents lots of technical concepts, I'd divide this paper in two manuscripts and focus on convincing the reader that waterline extraction can be useful to derive intertidal DEM in NZ and leave the SDB and modelling approach for another opportunity. A short discussion is also presented in this submitted version for such complex topic. Therefore, a major revision or complete new submission is recommended

We understand that the novelty of our work is the part that discusses the assessment of hydrodynamic modelling. Thus, removing this part from our manuscript would affect the paper's novelty. The SDB techniques shown here

are well discussed in the literature and our paper is not aiming to improve any of them in a significant way.

3. It is not clear to me why one would embark on a shoreline extraction method to create an intertidal model, when one could use SDB (Stumpf and others) to obtain bathymetry, especially where white water/waves are absent.

The Stumpf method has the disadvantage of not properly working in deep areas or when the turbidity of the water is great. The waterline technique has been proven to be efficient in intertidal zones, performing better than the Stumpf method in several studies, when compared. Also the Stumpf method requires in situ depth measurements, but the waterline method does not.

4. I found the introduction a quite confusing as it mixed two different uses of satellite. One to derive bathymetry and the other to derive shoreline positions. This is carried out from the Abstract to Introduction to the other parts of the text, and therefore I suggest a complete rewrite of these sections.

Shoreline positions are used as part of the method for estimating bathymetric data through satellite images. We will make this more clear, as we can see that readers do not expect it to be used in this way.

Some specific points below:

5. L9 - I'd suggest to modify text to make better use of the acronym – Satellite-derived bathymetry (SDB) which obviously differ from "use of satellite images to estimate bathymetry"

We changed "use of satellite images to estimate bathymetry" to "satellite-derived bathymetry".

6. L11 four instead of 4. Same in L16

Done.

7. L18 The use of the satellite derived bathymetry in hydrodynamic models does not result in significant differences in terms of water levels, when

compared with the scenario modelled using surveyed bathymetry. This seems a big claim to me considering that the method was only used in microtidal settings and NDWI performance in macrotidal places can be more complicate due to the larger wet areas. Sea grass bed areas appears also to be an issue.

We agree. We will modify the text to "In our case, The use of the satellite-derived bathymetry in hydrodynamic models does not result in significant differences in terms of water levels.."

8. LN23- what about meteorological tides? They seem quite important for predicting floods

We agree that these are important. However, we decided to focus on the astronomical tides as many places around the world have extreme astronomical tides (spring and equinoctial tides) as the main driver for coastal flooding. In addition, storm surges in New Zealand are quite low (average of 0.5 m). We will add a paragraph in section 1 or in section 2, to better describe the physical processes involved in the flooding events and the particularities of New Zealand locations.

9. LN24 no hyphen in sea level

Ok, it will be corrected.

10. LN25- to my knowledge atmospheric pressure is the one driving storm surges along the coast, fluvial discharge definitely adds to it, but it is the difference in pressure that elevates the water level

We agree and have changed the sentence.

11. Ln34 the references following this sentence should focus on SDB and not shoreline "To overcome these issues, efforts have centred on using spaceborn remote sensing (RS) techniques (Bishop-Taylor et al., 2019; Bué et al., 2020; Caballero and Stumpf, 2019)," should be replaced

We have not made ourselves clear. These references are focused on SDB. The Waterline extraction is only part of the method, as described in paragraph 4, lines 53 to 64; and also in section 2.2. We will rewrite this section to make it more clear.

12. LN41 Again I don't think the Bishop et al. ref is appropriate here, as their article addresses shoreline and not bathymetry

Our mapping method uses the shoreline, and so we do think that the reference is relevant, but we will change the introduction to make it much more clear why the shoreline literature is relevant.

13. LN 42 rewrite "use a radiometric approach, which uses the property that different wavelengths are attenuated to varying degrees in the water column".

We will re-write this as "use a radiometric approach, which is based on the property that light attenuation occurs in the water column in different degrees according to the wavelengths of the visible spectrum".

14. LN55 detecting the land-water boundary has nothing to do with deriving bathymetry with satellites. The shoreline is the interface not the morphology of the seafloor. Bishop et al., didn't derive bathymetry. They derived intertidal DEM, linking terrestrial and bathy datasets

We clearly have made this very confusing. We are using the changing shoreline over the intertidal region to map the morphology of the region, and use this to create an intertidal DEM, which we then refer to as bathymetric data. We will make sure that our terminology is clearly and consistently defined at the beginning of the paper.

15. LN57-59 only here I start to get a feeling of why you are talking about SDB, but even after that I think that you are creating a DEM of a few mts-depending on the local amplitude- instead of lets say shallow water bathymetry

We agree, and we will make sure the next version is clear.

16. LN71 bathymetry. You are talking about creating a DEM based on shoreline positions. Some of these positions will be above tide datum. Does that make bathymetry or topography?

Interesting. We are considering it as bathymetry because is below the maximum possible tide/water level. But this is something we should clarify in

the manuscript or adopt the term Intertidal topography instead of intertidal bathymetry.

17. LN77 2 main steps (Fig. 1).

Ok. It will be done.

18. LN78 two instead of 2

Ok. It will be done.

19. LN88 the intertidal zone is easily distinguished by the colour of sand accentuating reflectance in the near infrared band – This sentence seems out of context or needs some further clarification as Fig 2 is not a false colour image.

Ok. It will be re-written.

20. LN89 Associated with tidal flats, mangrove forest can be observed in all the studied estuaries. Where can I observe mangrove and seagrass banks? Modify text

Ok. We will give indications (e.g., southern part of the estuary with mangrove and the northern part with seagrass).

21. Ln92 I get confused here." For the implementation of SDB techniques, only tidal levels and imagery are needed. We used additional in situ bathymetric data to validate the SDB." Do you need bathy or not for implementing SDB?

You need additional bathymetry to validate the method, as explained in section 2.3. We will make this more clear.

22. LN114 t seems to me that some of the derived shoreline elevations cannot be considered bathymetry and this is part why I don't like the use of the term for this shoreline extractions. Some of the elevations will be above MSL. Can you still call it bathymetry? Shouldn't be topography then?

That is a good question. We will do some research on terminology and make sure we are consistent; in any case, we will make it clear that we are estimating the elevations in the intertidal zone to avoid misunderstanding.

23. LN140 you are using NDWI to define the intertidal area. Please explain know the 9 images for Tauranga or the 7 for Ohiwa are capturing the full extent of the intertidal area. Where they acquired during the lowest-highest tidal range?

You can see the full extension of the range captured by looking at Figure 7 for Tauranga harbour and in the supplementary material for the other locations. Indeed, the number of images limits the method, this aspect is discussed in section 4. A greater number of images would make for smaller errors.

24. LN156 Again- colloquialism

We will remove this.

25. LN157 (Fig. 4)

Change will be made.

26. LN158 once the waterline for a given image is identified, a height value is assigned to it accordingly to the corresponding tide level observed at the closest tide gauge (Omokoroa for the Tauranga Harbour case study, Fig. 2D). I see 4 gauges in the estuary. What's the rationale for choosing Omokoa? Oraumatua seems way closer to let's say Rangataua Bay! How do you account for the tidal lag? The level at the entrance is different than at the head.

We choose to use just Omokoroa tide observations to show the difference that could occur when the tidal lag is not accounted for, particularly when the tide gauge is situated a long way from the region of interest. In section 2.4 and 3.3 the use of what we called "dynamical correction" is applied to correct the effects of tide lag (where a local tide is used).

27. Ln202 hyphen mean-sea level

Change will be made.

28. LN259 Tauranga Harbour's waterline-derived SDB (primary SDB)- Sometimes I get really lost- What's primary SDB and how it differs from the other SDBs. Please explain

Primary SDB is the SDB without the statistical or dynamical corrections (explained in section 2.4). We will go through the entire paper and make sure that our explanations and definitions are much more clear.

29. Fig 8 font size too small

Change will be made.

30. Fig 9 Why are the coordinates in this map in NZGD? This figure needs to be improved. The colour scheme does not allow differentiation btw gauges and lines. It has 2 contradicting legends showing water lines as points and lines. Where are the LiDAR and the dynamic waterlines? Are they only shown in profile? I bit confusing to understand

The figures coordinates will be homogenized through all manuscript. The colour scheme will be changed. The points and triangles are the waterline position in profile for primary SDB and the dynamical SDB, respectively. The continuous and dashed lines represent the water level in the local tide gauge and in the dynamic model output. Lidar data are represented in black, only in profile. We will make sure everything is more clearly defined.

31. LN323 The simulation scenarios showed that it is possible to obtain similar, or even enhanced water level predictions, by using the SDB rather than the surveyed bathymetry – I'm a bit lost here. My understanding is that we need bathymetry to do SDB! At least I had to use a few lines in the past.

We meant that by using the estimates from SDB, as the bathymetry in numerical modelling, we can obtain reasonable predictions of water-level/tide. We will make this more clear.

32. LN 383 Bathymetric data are fundamental for solving the hydrodynamic equations in shallow water – This seems obvious, isn't?

Indeed. But is it too obvious for someone that is not familiar with numerical models? In any case, we can remove this sentence and the paragraph will start with "Hydrodynamic models and flooding risk assessments in coasts and estuaries are highly sensitive to depth values…".

33. LN6that it? A 1.5 pg long discussion, for such a complex paper?

We will expand the discussion. (This was also noted by the other reviewers)

---

## Author Response (AR1)

**Responses to Review Anonymous Referee 1**

Dear Reviewer anonymous referee 1,

Thank you for your observations regarding our preprint. Your suggestions helped to improve our manuscript. See below our responses marked in blue for each of your questions (marked in black).

The manuscript presents an interesting approach to deriving intertidal bathymetry from the waterline method through multispectral images, covering four (4) estuarine study areas on the east coast of Aotearoa New Zealand's North Island (Tauranga, Ohiwa, Maketu and Whitianga harbour). It represents a current thematic area, and it can be particularly useful to be applied in remote or inaccessible areas or where the bathymetric or cartographic data is very outdated. The main objectives of the study are to determine if multispectral images can be used to extract accurate intertidal bathymetric area and to assess the use of the SDB for hydrodynamic modelling of estuarine.

Good English level however the manuscript is not well-structured, quite confusing and the reader easily misses the main guidelines and the aim of the study. In section 1 (Introduction) is very difficult to establish a connection between the different ideas and paragraphs. A deeper revision of the state of the art is needed to bring the reader into the SDB theme and waterline method. The flow chart in chapter 2 is useful but does not really explain the methods used by the authors. Furthermore, the Methods Chapter establish that the main method was divided into 2 steps (1-SDB estimation and 2-Hydrodynamic modelling assessment) and that step 1 is also two methods for removing the bias, but a clear explanation of the methodology is not present in this section of the manuscript. A very short discussion and a shorter conclusions section are shown, where no clear main findings can be found. Modelling Storm surge is only referenced in the title of the manuscript.

The manuscript shows that a lot of work has been made, however, a big gap throughout the presented structure is noticed and the methodology used is not well described, creating a lot of misunderstanding between the methods applied and the different steps described by the authors. I, therefore, do not recommend the publication of this manuscript as it was presented. A major revision of the structure and methodology form is recommended.

Based on your revue and those provided by the other reviewers, we understand that we need to modify the structure of the paper because the current format is confusing. The new version has a deeper revision of the state of the art on SDB (adding new references and text to the introduction section). We have added a much clearer aim to the introduction, and worked on linking the methods to the aim in a much more clear and logical order. In the methods section, we added further explanations about the different methods implemented to remove the bias (i.e., statistical and dynamical methods). In the discussion and conclusion sections, we built further on the context provided in the new references added in the introduction part. Please note that we did not model an extreme storm surge, but we analyzed the maximum water level in all simulation scenarios and compared the outputs between scenarios using only surveyed bathymetry, only SDT, and mixed surveyed bathymetry combined with SDT. In terms of coastal flooding, the maximum water level is the main parameter studied in many places in the world, the water level is dominated by the tide. The discussion section has now 7 pages (previously 2 pages), and it is divided into several subsections as described in the text (4 subsections). Several new references were used (please see some of them in the end of the document).

My main critics are the following:

1. In the Introduction section the theme is not quite explained, and only part of the aim of the study is presented in the last sentence of the last paragraph. In this section is expected that the authors explain the reasons that have motivated this study, as well as what will be presented in the different sections of the entire manuscript.

We restructured the entire introduction section. We first said why elevation data are important in a coastal of flooding assessment (lines 24–46), stating that because of the limitation in the traditional methods, remote sensing techniques have been widely applied for topography and bathymetry data acquisition. Thus, we shown the main techniques regarding the satellite derived bathymetry (SDB) (lines 48–68) and topography (SDT) (lines 70–82). In the following lines (84–95), we said how cloud computation and satellite derived techniques have been allowed great progress in coastal science but there are only a few studies where numerical modelling and satellite derived

elevation have been used in combination (which is the main motivation of our work). To finish the section, we clearly state our main and specific goals (lines 97–103) and the content of each section (lines 105–114).

2. The different figures do not follow a consistent presentation. The geographic coordinates in some cases are presented as latitude/longitude with no reference datum associated (Fig.2(a)); others as X/Y coordinates (km) WGS84/UTM60S (Figs. 2 (b), (c), 3 (a)) and even other examples as X/Y coordinates NZGD2000 (km).

All figures were changed to the same unit and datum (WGS84/UTM 60S). Only Figure2(a) could not be put in UTM because the package I'm using in python does not support it.

3. The same Figures, presented in different sections, have different SI unit references, like Fig.2 (b) and (c) are expressed in X/Z coordinate (km) and Figures S1 and S2 in X/Z coordinate (m) – show a lack of consistency.

All figures will be updated to use the same SI unit references (km).

7. The areas A, B, C and D depicted in Figure7 (central figure) are not quite perceptible, and the small figures (a1, a2, b1, b2, c1, c2, d1 and d2) do not have geographic coordinates associated, neither the scale factor.

We modified the colour scheme and sizes to make the Figure more readable. The ABCD areas are easily visible now. We also added the coordinates and a scale factor.

5. The profile lines drawn in Figure 9 (m1) are barely noticed. Maybe the authors could choose a different colour palette.

The colour and the thickness of the profile were modified to make it more legible.

6. In the text, the figures are not correctly cited, like Fig 2A (line 85); Fig 2B and 2C (line 88). In the Figures, the panels are mentioned with small letters (a, b and c), as well as in the figure capture.

We changed all the labelling to the format of the journal (always using small case letters).

7. The data access information at the reference links (lines 94-97) is missing.

We added the links.

8. I do not understand how the intertidal area is identified, the method is not well explained. Is used the tidal level at the time of the acquisition of each image? Or is used an average tidal range (tidal amplitude?) for all the images. Is also not clear the tidal level for each image, as depicted in Figure 2(d). All images are used to generate the intertidal area presented in Figure 3(a)?

The intertidal area is identified by calculating the standard deviation of NDWI over the whole collection of images, at each pixel (this is now better described in Section 2.2, using equation 1). Because the water level changes substantially through time (because the tide completely drains and inundates these areas), these areas are easily identified, in a collection of images, by the high standard deviation of NDWI. We use a threshold to find the areas with high standard deviation and define the 'intertidal' area as being the area of high NDWI standard deviation. We made this clearer now, by adding a chart flow of our waterline method (Figure 3) and reformulating some sentences in Sect. 2.2.

We also added information about the tide level, date, and time of the image acquisition for all images processed to generate the SDT in each estuary (Supplement C).

9. The threshold value used, and all the contour extraction method (lines 153-159) are quite confusing. And which values of threshold and water level were used for the other study areas, regarding that Figure 4 presents the water level and threshold values for each image. A table with this information, for all the different study areas, as supplementary information could be very useful.

We add extra infomation and a detailed chart flow of the waterline method (see response to comment #8).

10. What do you mean with the Stumpf-ratio method was applied for deeper areas (lines 164-165). The Stumpf ratio method (Stumpf et al., 2003) is not quite good for all different benthic areas and for very deeper areas. What was the maximum depth value which the authors have used this method?

We applied the Stumpf-ratio (now called ratio-log) for shallow waters (the areas within the estuary that are not intertidal zones or land) and intertidal zones. This is well described now by the chart flow in Figure 1. The motivations for using the ratio-log-SDB are explained in Sect. 2.3. More specifically in lines (219–221):

"… Originally, the ratio-log empirical approach has been used to derive shallow water bathymetry. However, because of the relative low turbidity of intertidal water in Tauranga Harbour, we hypothesized that the method could be suitable also for deriving topography on intertidal zones…."

We are aware of the method's limitations, and these are discussed in Sect. 4.3. Details on the implementation and result of the ratio-log SDT and SDB are shown in Supplement A.

11. It was not explained by the authors all the pre-processing steps applied to the multispectral images, such as sun glint correction (for example Hedley, J.D.; Harborne, A.R.; Mumby, P.J. Simple and robust removal of sun glint for mapping shallow-water benthos. Int. J. Environ. 2005, 113, 2107–2112). If this step was considered, it should be enunciated in the manuscript. The authors described that Level 2 image was used, with BOA values corrected for the effects of the top-atmosphere (lines 103-104), but it was not explained why they used these images rather Level 1 with atmospheric correction.

We used the level 2 images because they were already corrected for the effects of the top of the atmosphere, and we did not believe it necessary to undertake our own correction when the contrast between dry and wet cells was so clear. We did not apply a sun glint correction because the Otsu algorithm could detect the waterline well without this correction. We wrote an entire paragraph discussing the pre-processing of satellite images in Sect. 4.3, lines (519–526).

12. I can not understand if the evaluation of the model performance in section 2.4 is one of the results of this study. And if they are, why not present them in the results section? Lines 201-215 have a challenging interpretation.

We changed the structure of how we were describing the hydrodynamic model in the manuscript. We now explained why we are using the model (Sect. 2.4.) and said that we calibrated and validated it. But all results of the calibration and validation are shown in the supplement B.

13. Why an" Average" line in table 3. Does not make sense.

The "average" refers to the average of the error metrics over all the estuaries for the corresponding parameter (MAE, RMSE, R2). This has been made more clear.

14. Lines 234-239 should be included in the Discussion section, not here, where the results are presented.

We have deleted these lines in the revised manuscript. The discussion section was completely reformulated.

15. In the ESA Sentinel 2A images used as background in several figures are missing the data acquisition time and the water-level information (Figures 2, 7, S1, S2, S3, S5, S6, S7).

We added the information in all captions of the figures and also in Supplement C.

16. The authors cannot quantify as good or strongly correlated/related the R2 achievements (lines 242-245). Why R2=0.70 should be considered as strongly related? Once more the authors are discussing the presented results in the Results section, and it is a recurrent procedure throughout this section. Perhaps if the authors had previously described in section 1 the contents of each section, the reader could understand better the manuscript. The structure of each section is quite confusing.

Changes done.

We described the contents of each section in the introduction part (last paragraph) as required, and also added a chart-flow illustrating every step of the manuscript. Also, we moved the discussion related to the Figures 7 and 9 (now Figures 12 and 11) to the discussion section (Sect. 4.3 and Sect. 4.2, respectively)

17. Lines 274-278: R2 values assumption/classification (low/higher). And R2 is referred to as the coefficient of determination (line 276) and a coefficient of correlation (line 278) in the same paragraph. Is not coherent.

We changed this and all comparisons between the ratio-log and waterline method have been moved to the discussion part (Sect. 4.3), line 504.

18. The authors do not explain why the results and the application of the methodology were only presented for one study area (Tauranga Harbour). They are free to do it, even for editorial figures or pages restrictions, however, this fact should be mentioned and explained in the manuscript and the main results for each area should be resumed (table format perhaps) in the supplementary material section.

The results were presented for just one study site because of limitation number of pages/figures, and Tauranga Harbour is the only harbor for which we have a numerical model already validated. We have added a sentence to specifically say this "We only applied the modelling study to this estuary because it has already a previous model calibrated and validated for the bed roughness (following (Stewart, 2021))." In the first paragraph of Section 2.4 (lines 230–231) . The results of the SDB estimates for each estuary are presented in the main manuscript, Sect. 3.1 (table 3, pg. 15) and in the supplement material (Figures S6, S7, and S8).

19. What is the spatial grid resolution value (line 298)? Is 20 m as assumed in line 336?

The grid resolution we applied here was 20 m. We have added this to the text (lines 234–235), Sect. 2.4.

20. Lines 313-316 are quite confusing. A better explanation is needed.

This entire section was moved to the discussion (Sect. 4.2) and rewritten to make it clearer.

21. The prediction of water level using the SDB is presented in section 3.4 for the 3 tide gauges (Omokoroa, Hairini and Oruamatua) (lines 323-331). The average error parameter presented is for each tide gauge and Figure 10 shows the average between all the tide gauges. Was this methodology that was used? I am confused.

We meant the average values between the three tide gauges records and not the average values for each tide gauge. We reformulated the sentence to make it clearer: "Figure 10 illustrates the average error parameters calculated when comparing the model output with the record of the three tide gauges. For a detailed assessment of each one of the gauges, please consult Supplement D" (lines 355–356).

22. Can I assume that, for lower tide values images, the presented methodology can not be used? Or only for the Stumpf ratio method application (SDB)? Lines 338-341. The Stumpf ratio method can not be directly applied to intertidal areas, exactly due to the image reflectance of the dry pixels (low water level images).

The model results for lower tide simulations showed the worst results. Thus, yes, it is not recommended to use SDB if the focus of the work is the processes occurring around the low tide, however for purposes of coastal flooding studies, usually, the highest values are of interest (as we show). The ratio-log-SDB is calculated using an image acquired at high tide. This is made more clear in the Methods (section 2.3).

23. What represents the rectangle-shaped figure in Figure 11? Survey bathymetry data or LIDAR data?

The bars represent the comparison between the in-situ observed water level at each of the 3 sites where there are water level sensors, and the hydrodynamic model output at those same sites, where the hydrodynamic model is run using 4 different bathymetries (S1, S2, S3, S4, are described in the text). We edited the figure and now all the labels and legends are more clear (now Figure 10).

NEW ADDED REFERENCES

Almeida, L. P., Efraim de Oliveira, I., Lyra, R., Scaranto Dazzi, R. L., Martins, V. G., and Henrique da Fontoura Klein, A.: Coastal Analyst System from Space Imagery Engine (CASSIE): Shoreline management module, Environ. Model. Softw., 140, 105033, https://doi.org/10.1016/j.envsoft.2021.105033, 2021.

Ashphaq, M., Srivastava, P. K., and Mitra, D.: Review of near-shore satellite-derived bathymetry : Classification and account of five decades of coastal bathymetry research, J. Ocean Eng. Sci., 6, 340–359, https://doi.org/10.1016/j.joes.2021.02.006, 2021.

Caballero, I. and Stumpf, R. P.: Towards routine mapping of shallow bathymetry in environments with variable turbidity: Contribution of sentinel-2A/B satellites mission, Remote Sens., 12, https://doi.org/10.3390/rs12030451, 2020.

Falcão, A. P., Mazzolari, A., Gonçalves, A. B., Araújo, M. A. V. C., and Trigo-Teixeira, A.: Influence of elevation modelling on hydrodynamic simulations of a tidally- dominated estuary, J. Hydrol., 497, 152–164, https://doi.org/10.1016/j.jhydrol.2013.05.045, 2013.

Fitton, J. M., Rennie, A. F., Hansom, J. D., and Muir, F. M. E.: Remotely sensed mapping of the intertidal zone: A Sentinel-2 and Google Earth Engine methodology, Remote Sens. Appl. Soc. Environ., 22, 100499, https://doi.org/10.1016/j.rsase.2021.100499, 2021.

Geyman, E. C. and Maloof, A. C.: A Simple Method for Extracting Water Depth From Multispectral Satellite Imagery in Regions of Variable Bottom Type, Earth Sp. Sci., 6, 527–537, https://doi.org/10.1029/2018EA000539, 2019.

IHO, I. H. O.: ORGANISATION HYDROGRAPHIQUE INTERNATIONALE ORGANIZACION HIDROGRAFICA INTERNACIONAL IHO / OHI Publication C-55, 2020.

Khojasteh, D., Hottinger, S., Felder, S., DeCesare, G., Heimhuber, V., Hanslow, D. J., andGlamore, W.: Estuarine tidal response to sea level rise: The significance of entrance restriction, Estuar. Coast. Shelf Sci., 244, 106941, https://doi.org/10.1016/j.ecss.2020.106941, 2020.

Khojasteh, D., Glamore, W., Heimhuber, V., andFelder, S.: Sea level rise impacts on estuarine dynamics: A review, Sci. Total Environ., 780, 146470, https://doi.org/10.1016/j.scitotenv.2021.146470, 2021.

Lee, Z., Carder, K. L., Mobley, C. D., Steward, R. G., andPatch, J. S.: Hyperspectral remote sensing for shallow waters I A semianalytical model, Appl. Opt., 37, 6329, https://doi.org/10.1364/ao.37.006329, 1998.

Liu, Y., Li, M., Zhou, M., Yang, K., andMao, L.: Quantitative analysis of the waterline method for topographical mapping of tidal flats: A case study in the dongsha sandbank, china, Remote Sens., 5, 6138–6158, https://doi.org/10.3390/rs5116138, 2013.

Salameh, E., Frappart, F., Marieu, V., Spodar, A., Parisot, J. P., Hanquiez, V., Turki, I., andLaignel, B.: Monitoring sea level and topography of coastal lagoons using satellite radar altimetry: The example of the Arcachon Bay in the Bay of Biscay, Remote Sens., 10, 1–22, https://doi.org/10.3390/rs10020297, 2018.

Salameh, E., Frappart, F., Almar, R., Baptista, P., Heygster, G., Lubac, B., Raucoules, D., Almeida, L. P., Bergsma, E. W. J., Capo, S., DeMichele, M. D., Idier, D., Li, Z., Marieu, V., Poupardin, A., Silva, P. A., Turki, I., andLaignel, B.: Monitoring Beach Topography and Nearshore Bathymetry Using Spaceborne Remote Sensing: A Review, Remote Sens., 11, https://doi.org/10.3390/rs11192212, 2019.

Salameh, E., Frappart, F., Turki, I., and Laignel, B.: Intertidal topography mapping using the waterline method from Sentinel-1 & -2 images: The examples of Arcachon and Veys Bays in France, ISPRS J. Photogramm. Remote Sens., 163, 98–120, https://doi.org/10.1016/j.isprsjprs.2020.03.003, 2020.

Turner, I. L., Harley, M. D., Almar, R., and Bergsma, E. W. J.: Satellite optical imagery in Coastal Engineering, Coast. Eng., 167, 103919, https://doi.org/10.1016/j.coastaleng.2021.103919, 2021.

**Responses to Review Anonymous Referee 2**

Dear Reviewer anonymous referee 2,

Thank you for your observations regarding our preprint. Your suggestions helped to improve our manuscript. See below our responses marked in blue for each of your main concerns (marked in black).

1- The manuscript presents a novel methodology for deriving intertidal bathymetry for four estuaries in New Zealand (Tauranga, Ohiwa, Maketu and Whitianga harbour) characterized by a complex morphology. I find this thematic interesting, as it allows to update and improve the boundary conditions of regional numerical models. However, I think the structure and writing of the manuscript require further work to reflect all the work done. The manuscript needs a better use of English, a restructuring of the chapters and above all to emphasize the purpose of the work as well as the authors' motivation and innovations. Therefore, I do not recommend the publication of this manuscript as submitted. This review is critical, nonetheless the authors have the potential to have a great manuscript and I would like to encourage them in their progress.

We appreciate that you found our manuscript interesting, despite the problems regarding its structure. The inadequate structure is a problem highlighted by all the 3 reviewers.

We restructured the entire introduction section (and manuscript as well). We first said why elevation data are important in a coastal flooding assessment (lines 23–46), stating that because of the limitation in the traditional methods, remote sensing techniques have been widely applied for topography and bathymetry data acquisition. Thus, we showed the main techniques regarding satellite-derived bathymetry (SDB) (lines 47–68) and topography (SDT) (lines 69–81). In the following lines (83–94), we said how cloud computation and satellite-derived techniques have allowed great progress in coastal science but there are just a few studies where numerical modelling and satellite-derived elevation have been used in combination (which is the main motivation of our work). To finish the section, we clearly state our main and specific goals (lines 96–102) and the content of each section (lines 104–113).

Please note also that we have added new references, some examples are at the end of the document.

2 - I mainly concern of the reasoning and the reading flow, which is quite confusing and the reader can easily miss the guidelines of the study. Section 1 does not clearly show the developments achieved by the scientific community, the relevance of the chosen methodology and, above all, the authors' motivations.

Please see the answer to comment 1.   In addition, we re-structured the entire paper, and we added a chart-flow to illustrate all the processes and content of the sections (Figure1).
The motivation is now clearly stated in lines 89-94:
"... Despite the vast and growing application of SDB and SDT methods to coastal science and engineering (Turner et al., 2021), it is not yet clear whether the accuracy of the resulting estimates is suitable for modelling extreme water levels in coastal areas (e.g., estuaries and bays). Only limited studies exist relating to SDB and SDT and numerical modelling —   generally aimed at using the model to assign the waterline height (Fitton et al., 2021; Khan et al., 2019; Salameh et al., 2020). For instance, Mason et al. (2010) used SDT to calibrate a morphodynamical model..."

3 - Section 2 is very long and presents too many technical concepts, and even results, that is hard to follow how they were implemented. The study area should be expanded with a description of the main processes describing water level dynamic, since the work's title mentions storm surge modelling.
We have described the physical particularities of the study areas in Sect. 2.1 (lines 119–126):
"...The studied sites have micro-tidal regimes — the spring tidal range varies between 1.4 m to 1.9 m within estuaries – and the equinoctial spring tides combined with severe storm surges drive the extreme sea levels (Rueda et al., 2019; Stephens et al., 2020). In New Zealand, the surges caused by storms usually add < 0.5 m to the water level, but the maximum surge ever registered was 2.29 m (Stephens et al., 2020). In Tauranga Harbour, the maximum storm-driven surge ever recorded is equal to 0.88 m (Stephens et al., 2020), and the tide can be attenuated by 10% to 17% (M2 component) when the tidal wave propagates through the estuary (Tay et al., 2013). The water level inside the study site estuaries is not considered to be substantially affected by the action of waves (i.e., wave set-up) because all of them are enclosed coastal lagoons

with restricted entrances."

The discussion section was increased to 7 pages, divided in 4 subsections :
4.1 Our proposed waterline method for deriving topography from space-borne images and its limitations.
4.2 Our proposed correction methods for waterline-SDTs.
4.3 Comparison between the waterline-method and ratio-log for intertidal zones.
4.4 Hydrodynamic modelling assessment

In the first three discussion subsections we addressed our first and second specific objectives, discussing our main findings and limitations when applying the waterline and ratio-log methods, including suggestion for improvements. In the fourth part, we addressed our third objective, discussing the challenges of using SDTs and SDBs in modelling extreme water levels within a complex-morphology estuary.

The Section 2 is now divided in 7 parts to better discretize each one of the methods used in the work. Also Figures 1 (pg. 5) and Figure 3 (pg. 8) bring a general guidance throughout the structure of the manuscript.

4- Results and figures in Section 3 present a lack of consistency of SI units, authors should homogenize them. I think criteria presented in Fig. 3 and 4 are unclear and need a deeper discussion. Errors should be accompanied by their percentage for better interpretation. Unfortunately, the color map chosen for Fig. 7 and 9 is not good for presenting such significant results. The conclusions are extremely short and not summarise the reasoning of the work.

We changed the figures to consistent SI units. We rewrote Section 2.2 with more details on the waterline method and added Figure 3 (pg. 8), a general chart flow of the method. We also start the section by stating all the stages of the method and use Figure 3 as a guide for the reading (lines 161–163):

"Our proposed framework to generate the SDT in intertidal zones using the waterline method (waterline-SDT) is composed of 4 stages, as illustrated in Figure 3: stage 1 is to query an image collection, stage 2 is to identify the intertidal zone, stage 3 is to determine the waterline position and height, and stage 4 is to post-process results. First, we acquired an image collection for

each estuary through the Google Earth Engine application (Gorelick et al., 2017) using the Google Colaboratory environment...."

We changed the colour maps of Figures 7 and 9 (now Figures 12 and 11, respectively). We added several more references and pages to the discussion, as mentioned in the previous question 3. We also made the Conclusions more clear (although avoiding repeating the discussion).

Information about the range of depth in the LiDAR data for the intertidal zone is shown with general results for each estuary in Table 3 (pg.15). Thus, it is possible to directly infer how the waterline-SDT is performing related to the LiDAR. We also showed the digital elevation model (DEM) errors in the same table.

NEW REFERENCES ADDED

Almeida, L. P., Efraim de Oliveira, I., Lyra, R., Scaranto Dazzi, R. L., Martins, V. G., and Henrique da Fontoura Klein, A.: Coastal Analyst System from Space Imagery Engine (CASSIE): Shoreline management module, Environ. Model. Softw., 140, 105033, https://doi.org/10.1016/j.envsoft.2021.105033, 2021.

Ashphaq, M., Srivastava, P. K., and Mitra, D.: Review of near-shore satellite-derived bathymetry : Classification and account of five decades of coastal bathymetry research, J. Ocean Eng. Sci., 6, 340–359, https://doi.org/10.1016/j.joes.2021.02.006, 2021.

Caballero, I. and Stumpf, R. P.: Towards routine mapping of shallow bathymetry in environments with variable turbidity: Contribution of sentinel-2A/B satellites mission, Remote Sens., 12, https://doi.org/10.3390/rs12030451, 2020.

Falcão, A. P., Mazzolari, A., Gonçalves, A. B., Araújo, M. A. V. C., and Trigo-Teixeira, A.: Influence of elevation modelling on hydrodynamic simulations of a tidally- dominated estuary, J. Hydrol., 497, 152–164, https://doi.org/10.1016/j.jhydrol.2013.05.045, 2013.

Fitton, J. M., Rennie, A. F., Hansom, J. D., and Muir, F. M. E.: Remotely sensed mapping of the intertidal zone: A Sentinel-2 and Google Earth Engine methodology, Remote Sens. Appl. Soc. Environ., 22, 100499, https://doi.org/10.1016/j.rsase.2021.100499, 2021.

Geyman, E. C. and Maloof, A. C.: A Simple Method for Extracting Water Depth From Multispectral Satellite Imagery in Regions of Variable Bottom Type, Earth Sp. Sci., 6, 527–537, https://doi.org/10.1029/2018EA000539, 2019.

IHO, I. H. O.: ORGANISATION HYDROGRAPHIQUE INTERNATIONALE ORGANIZACION HIDROGRAFICA INTERNACIONAL IHO / OHI Publication C-55, 2020.

Khojasteh, D., Hottinger, S., Felder, S., DeCesare, G., Heimhuber, V., Hanslow, D. J., andGlamore, W.: Estuarine tidal response to sea level rise: The significance of entrance restriction, Estuar. Coast. Shelf Sci., 244, 106941, https://doi.org/10.1016/j.ecss.2020.106941, 2020.

Khojasteh, D., Glamore, W., Heimhuber, V., andFelder, S.: Sea level rise impacts on estuarine dynamics: A review, Sci. Total Environ., 780, 146470, https://doi.org/10.1016/j.scitotenv.2021.146470, 2021.

Lee, Z., Carder, K. L., Mobley, C. D., Steward, R. G., andPatch, J. S.: Hyperspectral remote sensing for shallow waters I A semianalytical model, Appl. Opt., 37, 6329, https://doi.org/10.1364/ao.37.006329, 1998.

Liu, Y., Li, M., Zhou, M., Yang, K., andMao, L.: Quantitative analysis of the waterline method for topographical mapping of tidal flats: A case study in the dongsha sandbank, china, Remote Sens., 5, 6138–6158, https://doi.org/10.3390/rs5116138, 2013.

Salameh, E., Frappart, F., Marieu, V., Spodar, A., Parisot, J. P., Hanquiez, V., Turki, I., andLaignel, B.: Monitoring sea level and topography of coastal lagoons using satellite radar altimetry: The example of the Arcachon Bay in the Bay of Biscay, Remote Sens., 10, 1–22, https://doi.org/10.3390/rs10020297, 2018.

Salameh, E., Frappart, F., Almar, R., Baptista, P., Heygster, G., Lubac, B., Raucoules, D., Almeida, L. P., Bergsma, E. W. J., Capo, S., DeMichele, M. D., Idier, D., Li, Z., Marieu, V., Poupardin, A., Silva, P. A., Turki, I., andLaignel, B.: Monitoring Beach Topography and Nearshore Bathymetry Using Spaceborne Remote Sensing: A Review, Remote Sens., 11, https://doi.org/10.3390/rs11192212, 2019.

Salameh, E., Frappart, F., Turki, I., and Laignel, B.: Intertidal topography mapping using the waterline method from Sentinel-1 & -2 images: The examples of Arcachon and Veys Bays in France, ISPRS J. Photogramm. Remote Sens., 163, 98–120, https://doi.org/10.1016/j.isprsjprs.2020.03.003, 2020.

Turner, I. L., Harley, M. D., Almar, R., and Bergsma, E. W. J.: Satellite optical imagery in Coastal Engineering, Coast. Eng., 167, 103919, https://doi.org/10.1016/j.coastaleng.2021.103919, 2021.

**Responses to Review Anonymous Referee 3**

Dear Reviewer anonymous referee 3,

Thank you for your observations regarding our preprint. Your suggestions helped to improve our manuscript. See below our responses marked in blue for each of your main concerns (marked in black).

1. I have read the paper entitled "Modelling tides and storm surge using intertidal bathymetry derived from the waterline method applied to multispectral satellite images" by Costa et al. The study aims to determine whether satellite imagery can be used to extract accurate intertidal bathymetric data; and assess the use of the SDB for hydrodynamic modelling of estuaries. The paper is interesting, and one can see that quite a lot of effort, based on the complex methodology, was put into it. However, the manuscript is quite confusing making it difficult to understand. Part of the problem I had was regarding the use of the term SDB to represent extracted shorelines when the term is coined to deriving bathymetry. The study seems to have its merits but needs a complete re-structure. I found results difficult to understand.

We are pleased that the reviewer finds our manuscript interesting. We are aware (based on your review and others) that the structure of the paper is not ideal, and the structure makes the paper confusing to read. We re-structured the paper according to the reviewers' comments. With respect to the methods, we realized that the reviewer misunderstood us, and we have rewritten them to make ourselves more clear. The identification of the waterline position is done as part of the waterline method for estimating bathymetric data in the intertidal zone. We have now changed our definitions, and in the intertidal region we use "satellite-derived topography" and in shallow water we use "satellite-derived bathymetry". Following past work, we use the movement of shoreline over the intertidal as a way of detecting the topography. We made the methods more clear by also adding a chart-flow about the paper structure (Figure 1, pg.5) and our framework for the waterline method (Figure 3, pg.8).

2. Because it presents lots of technical concepts, I'd divide this paper in two manuscripts and focus on convincing the reader that waterline extraction can be useful to derive intertidal DEM in NZ and leave the SDB and modelling approach for another opportunity. A short discussion is also

presented in this submitted version for such complex topic. Therefore, a major revision or complete new submission is recommended

We understand that the novelty of our work is the part that discusses the assessment of hydrodynamic modelling. Thus, removing this part from our manuscript would affect the paper's novelty. The SDB techniques shown here are well discussed in the literature and our paper is not aiming to improve any of them in a significant way. We believe that now with the improvement of the discussion part (several new references and a new discussion section of 7 pages) and the clearer methodology will make the topic less complex, and the two parts of the study fit together more naturally.

3. It is not clear to me why one would embark on a shoreline extraction method to create an intertidal model, when one could use SDB (Stumpf and others) to obtain bathymetry, especially where white water/waves are absent.

The Stumpf method has the disadvantage of not working well in deep areas or when the turbidity of the water is great. The waterline technique has been proven to be efficient in intertidal zones, performing better than the Stumpf method in several studies, when compared. Also the Stumpf method requires in situ depth measurements, but the waterline method does not. We compared both methods in the discussion section (Sect. 4.3.).

4. I found the introduction a quite confusing as it mixed two different uses of satellite. One to derive bathymetry and the other to derive shoreline positions. This is carried out from the Abstract to Introduction to the other parts of the text, and therefore I suggest a complete rewrite of these sections.

Shoreline positions are used as part of the method for estimating bathymetric data through satellite images. We hope that this is more clear in the revised manuscript. We re-structured the paper, now we make it very clear that the satellite images are used to derive topographic data (SDT) (Sect. 2.2) and bathymetric data (SDB) (Sect. 2.3). We also added new references in the introduction (please see references at the end of the present document)

Some specific points below:

5. L9 - I'd suggest to modify text to make better use of the acronym – Satellite-derived bathymetry (SDB) which obviously differ from "use of satellite images to estimate bathymetry"

We changed "use of satellite images to estimate bathymetry" to "use of satellite images to derive bathymetry" (now line10).

6. L11 four instead of 4. Same in L16

Done.

7. L18 The use of the satellite derived bathymetry in hydrodynamic models does not result in significant differences in terms of water levels, when compared with the scenario modelled using surveyed bathymetry. This seems a big claim to me considering that the method was only used in microtidal settings and NDWI performance in macrotidal places can be more complicate due to the larger wet areas. Sea grass bed areas appears also to be an issue.

We agree. We modified the text to "For Tauranga Harbour, the use of SDT and SDB in hydrodynamic models does not result in significant differences in predicting high water levels when compared with the scenario modelled using surveyed bathymetry." (lines 20–21)

8. LN23- what about meteorological tides? They seem quite important for predicting floods

We agree that these are important. However, we decided to focus on the astronomical tides as many places around the world have extreme astronomical tides (spring and equinoctial tides) as the main driver for coastal flooding. In addition, storm surges in New Zealand are quite low (average of 0.5 m). We added a paragraph in Sect. 2.1, to better describe the physical processes involved in the flooding events and the particularities of New Zealand locations (lines 119–126):

" The studied sites have micro-tidal regimes — the spring tidal range varies between 1.4 m to 1.9 m within estuaries – and the equinoctial spring tides combined with severe storm surges drive the extreme sea levels (Rueda et al., 2019; Stephens et al., 2020). In New Zealand, the surges caused by storms usually add < 0.5 m to the water level, but the maximum surge ever registered

was 2.29 m (Stephens et al., 2020). In Tauranga Harbour, the maximum storm-driven surge ever recorded is equal to 0.88 m (Stephens et al., 2020), and the tide can be attenuated by 10% to 17% (M2 component) when the tidal wave propagates through the estuary (Tay et al., 2013). The water level inside the study site estuaries is not considered to be substantially affected by the action of waves (i.e., wave set-up) because all of them are enclosed coastal lagoons with restricted entrances."

Also, we discussed the effect of waves and storm surges in Sect. 4.4, last paragraph (lines 560–568):

"Although the differences in the resulting water level between the SDT, SDB, and surveyed bathymetry simulation scenarios show that satellite techniques compare well, our simulations were only conducted in one estuary, albeit a large and relatively complex estuary — where the astronomical spring tides are the main driver for estuarine flooding. Therefore, studies are required in sites with different physical conditions would be useful to validate the use of SDT and SDB more broadly. For instance, estuaries where the storm surge is the main driver for flooding; or/and exposed estuaries where the wave forces can increase the water level (i.e., wave set-up) (e.g., Bertin et al. (2019)). Furthermore, modelling studies focusing on understanding whether or not the use of SDT and SDB properly represent the tide-surge interactions within the estuary are encouraged, due to the importance of the topic in water level modelling (Spicer et al., 2019; Wankang et al., 2019; Zheng et al., 2020), especially in the context of sea level rise."

9. LN24 no hyphen in sea level

We have changed this throughout

10. LN25- to my knowledge atmospheric pressure is the one driving storm surges along the coast, fluvial discharge definitely adds to it, but it is the difference in pressure that elevates the water level

We agree and have changed the sentence. It is now written as follows: "In practice, predicting flooding depends on understanding the contribution from the astronomical tide, storm surge, wave run-up, changes in the sea level and, in some cases, the fluvial discharge and vertical land motion." (lines 25–27)

11. Ln34 the references following this sentence should focus on SDB and not shoreline "To overcome these issues, efforts have centred on using spaceborn remote sensing (RS) techniques (Bishop-Taylor et al., 2019; Bué et al., 2020; Caballero and Stumpf, 2019)," should be replaced

We re-wrote the entire manuscript dividing the techniques into satellite-derived topography (SDT) and satellite-derived Bathymetry (SDB). We used the waterline and ratio-log methods to derive topography data (as described in sections 2.2 and 2.3, respectively). We have also applied the ratio-log method for deriving bathymetry in shallow waters (Sect. 2.3). We explained in detail all the paper's structure in Figure 1 (pg.5) and the waterline framework in Figure 3 (pg.8). Details on the implementation of the ratio-log method are given in Supplement A.

12. LN41 Again I don't think the Bishop et al. ref is appropriate here, as their article addresses shoreline and not bathymetry

We agree and we have rephrased the reference to Bishop et al.: "...and extensive intertidal areas (Bishop-Taylor et al., 2019; Fitton et al., 2021)..." (lines 49–50)

13. LN 42 rewrite "use a radiometric approach, which uses the property that different wavelengths are attenuated to varying degrees in the water column".

We re-wrote all of the introduction. Now the equivalent sentence is located in lines 52–53: "Most of methods are developed around the process of light attenuation through the water column, and fall into two approaches".

14. LN55 detecting the land-water boundary has nothing to do with deriving bathymetry with satellites. The shoreline is the interface not the morphology of the seafloor. Bishop et al., didn't derive bathymetry. They derived intertidal DEM, linking terrestrial and bathy datasets

We have clarified the differences between techniques to derive topography and bathymetry data (SDT and SDB, respectively), and made it much clearer how the shoreline can be used to map topography. We are using both datasets to create DEM that can be use as the input in hydrodynamic modelling.

15. LN57-59 only here I start to get a feeling of why you are talking about SDB, but even after that I think that you are creating a DEM of a few mts- depending on the local amplitude- instead of lets say shallow water bathymetry

We agree, and we made modifications in all paper structure as answered in the previous questions in this document.

16. LN71 bathymetry. You are talking about creating a DEM based on shoreline positions. Some of these positions will be above tide datum. Does that make bathymetry or topography?

We separated in intertidal topography (Sect.2.2 and Sect. 2.3) and shallow water bathymetry (Sect. 2.3 and Supplement A).

17. LN77 2 main steps (Fig. 1).

Ok. Changes are done. (line104)

18. LN78 two instead of 2

Ok. Changes are done (line104).

19. LN88 the intertidal zone is easily distinguished by the colour of sand accentuating reflectance in the near infrared band – This sentence seems out of context or needs some further clarification as Fig 2 is not a false colour image.

Ok. We re-wrote the sentence to: "The extent of the tidal flats is evident in Tauranga Harbour by comparing low (e.g., Figure 2B) and high (e.g., Figure 2C) tide satellite images." (lines 127–129)

20. LN89 Associated with tidal flats, mangrove forest can be observed in all the studied estuaries. Where can I observe mangrove and seagrass banks? Modify text

Ok. The sentence was modified to: "Mangrove forests can be observed in all the estuaries and seagrass banks are visible in Maketū, Ōhiwa and Tauranga Harbours (the latter was studied in Ha et al. (2020)). Detailed images of the intertidal zones in Tauranga Harbour and its seagrass banks and mangroves can be seen in Figure S3." (lines 130–131).

21. Ln92 I get confused here." For the implementation of SDB techniques, only tidal levels and imagery are needed. We used additional in situ bathymetric data to validate the SDB." Do you need bathy or not for implementing SDB?

The entire section was rewritten, but the corresponding sentence is now: "Imagery, tidal levels and topography data (e.g., LiDAR) were acquired to implement and validate the SDT techniques." (line 133)

22. LN114 t seems to me that some of the derived shoreline elevations cannot be considered bathymetry and this is part why I don't like the use of the term for this shoreline extractions. Some of the elevations will be above MSL. Can you still call it bathymetry? Shouldn't be topography then?

As mentioned above, we changed the term to satellite-derived topography (SDT).

23. LN140 you are using NDWI to define the intertidal area. Please explain know the 9 images for Tauranga or the 7 for Ohiwa are capturing the full extent of the intertidal area. Where they acquired during the lowest-highest tidal range?

You can see the full extension of the range captured by looking at Figure 7 for Tauranga harbour and in the supplementary material for the other locations. Indeed, the number of images limits the method, this aspect is discussed in Section 4.1 A greater number of images would make for smaller errors (lines 413–417):

" …Furthermore, having enough images to characterise the morphology of the study site is also a limiting factor in the waterline method, as pointed out by several studies (Salameh et al., 2019; Liu et al., 2013). Our results are also clearly affected by the number of images in our collection. For instance, we observed gaps between different waterlines, where no topographic data could be derived, shown in Fig. 7 (Sect. 3.2). Although we have used Sentinel-2 images acquired every five days, they are often not useable due to cloud coverage. …"

24. LN156 Again- colloquialism

We removed this (lines 203–204).

25. LN157 (Fig. 4)

Change done. All figures are referred as Figure.

26. LN158 once the waterline for a given image is identified, a height value is assigned to it accordingly to the corresponding tide level observed at the closest tide gauge (Omokoroa for the Tauranga Harbour case study, Fig. 2D). I see 4 gauges in the estuary. What's the rationale for choosing Omokoa? Oraumatua seems way closer to let's say Rangataua Bay! How do you account for the tidal lag? The level at the entrance is different than at the head.

We choose to use just Omokoroa tide observations to show the difference that could occur when the tidal lag is not accounted for, particularly when the tide gauge is situated a long way from the region of interest. In section 2.4 and 3.3 the use of what we called "dynamical correction" is applied to correct the effects of tide lag (where a local tide is used).

27. Ln202 hyphen mean-sea level

Change done. (line 240)

28. LN259 Tauranga Harbour's waterline-derived SDB (primary SDB)- Sometimes I get really lost- What's primary SDB and how it differs from the other SDBs. Please explain

We are no longer using this term, which makes the manuscript easier to follow.

29. Fig 8 font size too small

Change done.

30. Fig 9 Why are the coordinates in this map in NZGD? This figure needs to be improved. The colour scheme does not allow differentiation btw gauges and lines. It has 2 contradicting legends showing water lines as points and lines. Where are the LiDAR and the dynamic waterlines? Are they only shown in profile? I bit confusing to understand

Changes are done. We change the legend and the colour scheme, please see Figure 11. Lidar data are represented in black, only in profile.

LN323 The simulation scenarios showed that it is possible to obtain similar, or even enhanced water level predictions, by using the SDB rather than the

surveyed bathymetry – I'm a bit lost here. My understanding is that we need bathymetry to do SDB! At least I had to use a few lines in the past.

We now are referring to SDT (satellite-derived topography) and SDB (satellite-derived bathymetry). Scenarios S1, S2, S3 and S4 explore different DEMs in hydrodynamic modelling. It is true that the SDB needs some calibration data, and we have noted this (lines 353–355).

31. LN 383 Bathymetric data are fundamental for solving the hydrodynamic equations in shallow water – This seems obvious, isn't?

We removed this sentence from the manuscript.

32. LN6that it? A 1.5 pg long discussion, for such a complex paper?

We expanded the discussion to 7 pages divided in 4 subsections. We hope the rewritten paper brings a richer discussion now.

**NEW ADDED REFERENCES**

Almeida, L. P., Efraim de Oliveira, I., Lyra, R., Scaranto Dazzi, R. L., Martins, V. G., and Henrique da Fontoura Klein, A.: Coastal Analyst System from Space Imagery Engine (CASSIE): Shoreline management module, Environ. Model. Softw., 140, 105033, https://doi.org/10.1016/j.envsoft.2021.105033, 2021.

Ashphaq, M., Srivastava, P. K., and Mitra, D.: Review of near-shore satellite-derived bathymetry : Classification and account of five decades of coastal bathymetry research, J. Ocean Eng. Sci., 6, 340–359, https://doi.org/10.1016/j.joes.2021.02.006, 2021.

Caballero, I. and Stumpf, R. P.: Towards routine mapping of shallow bathymetry in environments with variable turbidity: Contribution of sentinel-2A/B satellites mission, Remote Sens., 12, https://doi.org/10.3390/rs12030451, 2020.

Falcão, A. P., Mazzolari, A., Gonçalves, A. B., Araújo, M. A. V. C., and Trigo-Teixeira, A.: Influence of elevation modelling on hydrodynamic simulations of a tidally- dominated estuary, J. Hydrol., 497, 152–164, https://doi.org/10.1016/j.jhydrol.2013.05.045, 2013.

Fitton, J. M., Rennie, A. F., Hansom, J. D., and Muir, F. M. E.: Remotely sensed mapping of the intertidal zone: A Sentinel-2 and Google Earth Engine methodology, Remote Sens. Appl. Soc. Environ., 22, 100499, https://doi.org/10.1016/j.rsase.2021.100499, 2021.

Geyman, E. C. and Maloof, A. C.: A Simple Method for Extracting Water Depth From Multispectral Satellite Imagery in Regions of Variable Bottom Type, Earth Sp. Sci., 6, 527–537, https://doi.org/10.1029/2018EA000539, 2019.

IHO, I. H. O.: ORGANISATION HYDROGRAPHIQUE INTERNATIONALE ORGANIZACION HIDROGRAFICA INTERNACIONAL IHO / OHI Publication C-55, 2020.

Khojasteh, D., Hottinger, S., Felder, S., DeCesare, G., Heimhuber, V., Hanslow, D. J., andGlamore, W.: Estuarine tidal response to sea level rise: The significance of entrance restriction, Estuar. Coast. Shelf Sci., 244, 106941, https://doi.org/10.1016/j.ecss.2020.106941, 2020.

Khojasteh, D., Glamore, W., Heimhuber, V., andFelder, S.: Sea level rise impacts on estuarine dynamics: A review, Sci. Total Environ., 780, 146470, https://doi.org/10.1016/j.scitotenv.2021.146470, 2021.

Lee, Z., Carder, K. L., Mobley, C. D., Steward, R. G., andPatch, J. S.: Hyperspectral remote sensing for shallow waters I A semianalytical model, Appl. Opt., 37, 6329, https://doi.org/10.1364/ao.37.006329, 1998.

Liu, Y., Li, M., Zhou, M., Yang, K., andMao, L.: Quantitative analysis of the waterline method for topographical mapping of tidal flats: A case study in the dongsha sandbank, china, Remote Sens., 5, 6138–6158, https://doi.org/10.3390/rs5116138, 2013.

Salameh, E., Frappart, F., Marieu, V., Spodar, A., Parisot, J. P., Hanquiez, V., Turki, I., andLaignel, B.: Monitoring sea level and topography of coastal lagoons using satellite radar altimetry: The example of the Arcachon Bay in the Bay of Biscay, Remote Sens., 10, 1–22, https://doi.org/10.3390/rs10020297, 2018.

Salameh, E., Frappart, F., Almar, R., Baptista, P., Heygster, G., Lubac, B., Raucoules, D., Almeida, L. P., Bergsma, E. W. J., Capo, S., DeMichele, M. D., Idier, D., Li, Z., Marieu, V., Poupardin, A., Silva, P. A., Turki, I., andLaignel, B.: Monitoring Beach Topography and Nearshore Bathymetry Using Spaceborne Remote Sensing: A Review, Remote Sens., 11, https://doi.org/10.3390/rs11192212, 2019.

Salameh, E., Frappart, F., Turki, I., and Laignel, B.: Intertidal topography mapping using the waterline method from Sentinel-1 & -2 images: The examples of Arcachon and Veys Bays in France, ISPRS J. Photogramm. Remote Sens., 163, 98–120, https://doi.org/10.1016/j.isprsjprs.2020.03.003, 2020.

Turner, I. L., Harley, M. D., Almar, R., and Bergsma, E. W. J.: Satellite optical imagery in Coastal Engineering, Coast. Eng., 167, 103919, https://doi.org/10.1016/j.coastaleng.2021.103919, 2021.

---

## Referee Report (RR1)

**Second review of**
*Modelling extreme water levels using intertidal topography and bathymetry derived from multispectral satellite images*

All comments given in the previous review have been taking into account and they are reflected in this superior version of the work. I would like to highlight all the effort made in rewriting the paper and, in particular, the Introduction. The aims are clear and the reasoning is good. That said, I still have some concerns about the implementation and the analyses of the application. I think that the concept "one paper, one idea" should be take into account here, because the main results might vanish among the other relevant results. However, authors have proved that they are more than capable of addressing my considerations.

**Major Comments**

1) I understand the aim of Figure 1 and appreciate the effort done, but quite confusing. Your description in the paragraph starting at line 104 is clear. I suggest, in any case, more work on the text.

2) In line 16 it's said that the system is assessed during a storm surge event, but only tidal levels are presented. In fact, the utilized model, which validation is presented in the Supporting Material, considerates only the tide as external forcing. Why is it not forced with atmospheric fields? As you state in the Introduction, the storm surge peaks can reach values close to 30% of the tidal signal; hence, the incorporation of the atmospheric signal may be more than relevant. Indeed, I wonder if the incorporation could change the results presented in Section 3.4.

3) Based on the scale of the region, the tide presents wavelength of about hundreds of kilometers and the Rossby radius of deformation of about tens of kilometers. Therefore, the tide has co-oscillant dynamics rather wave propagation. Furthermore, due to the large scale of the process, it natural that the tide doesn't "feels" the improves achieved with the presented methodology. It's a restriction of the physics instead of the modeling. It would be interesting, in a future work, to perform a similar analysis with a relatively short wave such as wind waves.

4) In line 237 it says that astronomical tide analyses was undertaken, why do you apply the analysis? What information is utilized for this? Given the dimension of region, I expected that it to be nested or, at least, forced with tidal global models.

5) Some sections in Methods deserve more work to highlight the main ideas. For instance, Section 2.3, 2.5.1, 2.5.2, 2.6 and even Fig. 6 present brief descriptions and rely mainly on the Supporting Material. I recommend summarizing the main concepts needed to discuss the results and adding these few sentence before discussing the corresponding results.

6) For a better reading and to stress the main results, I suggest authors to merge Discussion and Conclusions and summarize the content.

**Minor comments**

1) There are still inconsistencies in unit system, for instance lines 14, 15 and 16.

2) In line 122 the term is "storm surge", please correct that expression throughout the manuscript.

3) In line 330 please be careful with the use of the correlation in this context, is it correlation or the goodness of fit? In any case, are they tested?

4) For a better reading, before the analysis, first introduce the figure and its aim.

5) In section 3.4 homogenize the statistical evaluation, please use the same scores for a better interpretation.

6) In line 386, what is the third part?

**Typo**

l8 Should be ",e.g.,"

l16 The sentence is too wordy, please rewrite to stress your idea.

L38 Should be "areas, respectively".

L58 SBD is not defined yet.

L87 SBD and SDT were defined above.

L125 Remove "study site"

L130, L231 Error in latex citing reference.

L135 Was or is?

L184 The equations should be separated according to the reading flow.

L207 Should be "2d"

L209 SDT or waterline-SDT?

L247 Please avoid expressions such as the last sentence.

L287 The statistical scores acronymous are not defined yet.

L327 MRE is not defined.

L534 IWD is defined but then it's not utilized.

L554 SLR stands for sea level rise?

---

## Referee Report (RR2)

I have finished my report on the revised version of Costa et al. 'Modelling extreme water levels using intertidal topography and bathymetry derived from multispectral satellite images'. I'm happy to inform that I'm more satisfied with the current version of this study which clarified several issues I had from the original submission. However, I still think the revised manuscript is quite complicated to understand and unbalanced (e.g. compare length of results vs methods). Part of the difficulties I have to comprehend relates to the use of the language and the length and complexity of methods. The latter extends from pg 5 to 15 excluding details allocated in supplementary material. The revised submission needs a good review by the authors emphasizing on the appropriate use of scientific language (conciseness, objectiveness, etc...) and attention to detail. As an overseas academic based in Australia whose first language is not English, I understand some of the language difficulties are not restricted to Mr. Costa only. A thorough revision is therefore needed. This should start on Ln3 (missing space between L.L.); full address for 1 and reduced address for 2 (Ln4 and 5) and follows down all the way to the reference list. The figures are difficult to understand and the captions need improving. Since the study was conducted in similar estuaries in NZ, a revised discussion would also need to contemplate the range of estuarine morphologies and forcing for the international audience. Below are some of the minor issues I detected.

Minor:

Ln 12- four instead of 4

Ln24 sea-level rise instead of sea level rise. Please revise this throughout paper (E.g. Ln34…). Note that we write 'sea level' and 'sea-level rise'.

Ln36 Not sure I agree with the fact that inundated areas in estuaries are generally shallow. What do you mean by this assertion? What do you consider shallow? 2, 5, 20, 50 m below msl. Some estuaries are deeper than that.

Ln 37 Consistency--Consider placing 'which are areas flooded and exposed…' into (). Alternatively take out the () of 'which are generally shallow'.

Ln 44 reword this sentence as the way it is written it seems that ~70% of the coast has not been surveyed, when in fact the whole world has been surveyed. What scale are you talking about? Make sure you are referring to large scale (e.g 1:100)

Ln 104-113 _ Is this explanation needed on a scientific paper? This seems to be a statement adapted from Mr. Costa's thesis. Please consider revising it.

Ln115 you previously indicated that you had three specific objectives (Ln98). Do you want to elaborate more on this Fig caption to indicate that (a) not only shows the steps taken to derive SDT/SDB but also to investigate stats relations/sources of errors?

Ln118 What's Aotearoa? Some part of text you say Aotearoa NZ, some others are only NZ. Is there a difference?  for the international reader?

Ln119 (Figure 2A)

Ln 120 the spring tide ranges from 1.4 to 1.9 m within estuaries

Ln121-123- Revise text. Is the use of BUT appropriate? I don't understand what's being said here. Storm surges add <0.5 m but the max surge was 2,29 m? I guess you mean the extreme sea level. Similar for the next sentence---  Tauranga = 0.88 m

Ln135 1 x 1 m – Add space

Fig 2  I have a major cartographic issue with this figure. Your use of different shapes and colours is completely inappropriate here. It makes the reader completely lost. See: You use coloured circles to represent the four locations (nothing wrong with it!). Then in B, you use different symbols with different colours and sizes to represent the different gauges. This is clearly a thematic cartography problem as we don't normally vary all three elements at once. In doing so, I tend to relate the Moturiki gauge with the Tauranga Harbour in A – they are both green circles! Then you complicate things with the D plots: Omokoroa is represented by black stars when legend in B shows red star, Hairini is also a black triangle and Oruamatua a black box. Finally, you mess it up one more time with the yellow and red symbols in legends! When I read the caption to try to understand this Fig, I don't find a reason why you have four gauges in B and only three plots in D. What's wrong with Moturiki (Green)? You need to spend more time on this figure. A lot of your issues come from the symbology in B. Try to make it easier for the reader.

Ln191 The threshold is set using the Otsu approach (Otsu, 1979) – Refine text to avoid repetition of the author's name

Table 1 – Surface area column – Round the area values to represent the approximation you indicated. Why are they in italic?

Fig 3 – This Fig summarises the framework for deriving topographic data. OK, but what is 4 Post-Processing? Why is this stage not explained as the other three? Why does it come before 3 in this framework? The caption also needs attention. It indicated that this is the framework and that NDWI is the index used, but stops there. The caption is extremely limited for such a complicated figure!

Fig 5 – Make this a 3 x 3 plot figure instead of a 4x4x1. Use your programming skills to make better use of the page's space.

Ln 225 What is a sub-estuary? Please clarify.

Ln247 Can you expand on " The model approximates the predicted data well (Sup B)"? How well? Please use a stat method to be more precise here. A single sentence will do it. Then refer to Sup B.

Ln255- Rewrite this sentence to indicate that only groundwater can be a potential source of error. If that's what you mean

Ln263 - 265– Rewrite paragraph to make your point clearer for the reader.

Ln330/331 I assume R2 is $R^2$  Please use superscript. Revise whole text

Ln332 – There's no Figure 6C

Ln335 – Is this a separate equation or part of plot 7?

Fig. 8 - Align the bottom two plots in relation to the second and third columns

Ln359/365 Use spaces RMSE~7cm, etc…

Fig 10 Why are the gauge symbols different from Fig2? Delete the ')' from y-axis in top left plot (after 60S)

Ln386-390 – Discuss with your co-authors the need of this paragraph. This is supposed to be a discussion of your results and not a summary of what was done!

Ln391 'Our' – Throughout text you write possessive forms. Stick to scientific language. Avoid Ours, We, etc… What about: '4.1 Waterline method for….limitations'; 4.2 Correction methods for… ?

Ln400 New Zealand

Ln 401 Our results also show.

Ln401-402 I'm not sure I understand this discussion. I was under the impression that all estuaries were quite similar and only groundwater had the potential to reduce the accuracy. Here you are indicating that that environmental conditions such as complex morphology can also reduce it. There's a weak argument here. Maybe you referring to the different estuarine types and stages of evolution and how this can interfere with results but this needs better arguments and references.

Given the range of estuaries and the international audience of this journal. Is there space for micro x macro tidal discussion in here? Is NDWI a good proxy in macrotidal settings or flat (low slope) shorelines?

Ln 455 Rewrite we are not eliminating horizontal errors. This is not scientific language at all!

Fig 11- I'm completely lost here. What am I looking at? I see the three profiles. The points are waterlines, but the map has lines, which I suppose are SDT WLH according to the legend below…Not sure what you mean here. Then on the right, you indicate p1 being the dyn. Corr, WLH 1; p2 being the dyn corr WLH2… Do you see what I am seeing?

Ln483 Capital letters- Check manuscript for proper names. We say Arcachon Bay, Maketu Estuary (Ln317)

Ln492 This is such a big claim considering that your method was only tested locally under limited conditions and restricted estuarine settings. Please reframe this to place your findings accordingly.

Ln492 – 508 You keep indicating that the waterline-SDT performs better than the ratio-log-SDT, but the text suggests that the latter is done on px-by-pixel basis. Is this the reason why the former outperforms the latter? Shouldn't you be comparing only the lines here?

Fig 12. Fix caption –missing ) after d1. Expand caption to explain why left is better than right.

Ln518 would be more uncertain than what?

Ln593 onwards: My editorial eye couldn't let this pass without a comment. A comprehensive revision of the reference list is needed. There are references in CAPITAL letters; typos; several words together; wrong author names; wrong abbreviations; lack of abbreviations, missing information; etc… These start in the very first ref (below) but extent throughout the list

Almeida, L. P., Efraim de Oliveira, I., Lyra, R., Scaranto Dazzi, R. L., Martins, V. G., andHenrique da Fontoura Klein, A.: Coastal Analyst System from Space Imagery Engine (CASSIE): Shoreline management module, Environ. Model. Softw., 140, 105033, https://doi.org/10.1016/j.envsoft.2021.105033, 2021.

Some of these issues are observed on a single ref.: 'Costa, W., Bryan, K. R., Coco, G., Zealand, N., andZealand, N.: ASSESSING THE USE OF SATELLITE DERIVED BATHYMETRY IN ESTUARINE STORM SURGE MODELS – STUDY CASE : TAURANGA, 2021.'

---

## Author Response (AR2)

Response to the second review of Modelling extreme water levels using intertidal topography and bathymetry derived from multispectral satellite images

All comments given in the previous review have been taking into account and they are reflected in this superior version of the work. I would like to highlight all the effort made in rewriting the paper and, in particular, the Introduction. The aims are clear and the reasoning is good. That said, I still have some concerns about the implementation and the analyses of the application. I think that the concept "one paper, one idea" should be take into account here, because the main results might vanish among the other relevant results. However, authors have proved that they are more than capable of addressing my considerations.

Thank you for your positive comments on the modifications made in the first round of revisions. We acknowledge your time and effort put in this second review. Below you can find your comments answered one by one in blue font. Your original comments are in black font.

**Major Comments**

1) I understand the aim of Figure 1 and appreciate the effort done, but quite confusing. Your description in the paragraph starting at line 104 is clear. I suggest, in any case, more work on the text.

We have made changes to the Figure 1 symbols and text in the manuscript. Now the text closely follows the figure. Lines 106–118.

2) In line 16 it's said that the system is assessed during a storm surge event, but only tidal levels are presented. In fact, the utilized model, which validation is presented in the Supporting Material, considerations only the tide as external forcing. Why is it not forced with atmospheric fields? As you state in the Introduction, the storm surge peaks can reach values close to 30% of the tidal signal; hence, the incorporation of the atmospheric signal may be more than relevant. Indeed, I wonder if the incorporation could change the results presented in Section 3.4.

We have included 4 additional simulation scenarios (explained in Section 2.6, lines 308–318). They include storm surge signal. We investigated two different extreme events recorded at the Moturiki tide gauge. They have similar magnitudes at the peak of water-level (~1.4 m) but one has less storm surge contribution (~22%) than the other (40%). Note that we forced the model with water level time series (which includes astronomical tide and storm surge), but not wind or atmospheric fields. We

agree that the use of wind/atmospheric fields in the model would have been interesting addition, because it can change the storm surge inside the estuary. However, given that the aim of our work is to assess SBD/SDT topo-bathymetry in modelling storm surge, and our paper is already quite long, we would like to leave this for future work.

3) Based on the scale of the region, the tide presents wavelength of about hundreds of kilometers and the Rossby radius of deformation of about tens of kilometers. Therefore, the tide has cooscillant dynamics rather wave propagation. Furthermore, due to the large scale of the process, it natural that the tide doesn't "feels" the improves achieved with the presented methodology. It's a restriction of the physics instead of the modeling. It would be interesting, in a future work, to perform a similar analysis with a relatively short wave such as wind waves.

We have included your observation on lines 531–532:

"Furthermore, the tidal wavelength is hundreds of kilometres, which means that the water level should not be affected significantly by smaller-scale bathymetric features."

4) In line 237 it says that astronomical tide analyses was undertaken, why do you apply the analysis? What information is utilized for this? Given the dimension of region, I expected that it to be nested or, at least, forced with tidal global models.

The model is forced with the local astronomical constituents of the tide record at Moturiki tide gauge. Due to the proximity of the tide gauge to the entrance of the estuary, we consider this a simple and reliable way to model the water level within the estuary. We are aware of the limitations of this assessment and highlighted them in the discussion section in numerous parts. For instance, Sect. 4.2 (lines 528–545) and 4.4 (lines 625–632).

5) Some sections in Methods deserve more work to highlight the main ideas. For instance, Section 2.3, 2.5.1, 2.5.2, 2.6 and even Fig. 6 present brief descriptions and rely mainly on the Supporting Material. I recommend summarizing the main concepts needed to discuss the results and adding these few sentence before discussing the corresponding results.

Change made. We have re-written part of the paragraphs of Section 2.3, 2.4, 2.5.1, 2.5.2, and 2.6 to make them clearer. We first write what we are doing, and then why we are doing it. Modifications can be seen in the tracked-changes document. We also improve the link between the first paragraphs of methods and discussion. For instance in Sect. 2.3 the first two paragraphs we make it clear that we are comparing ratio-log and waterline and why. In Section 4.3, we start the discussion by directly comparing

the methods, as follows:

*"The results show that, for Tauranga Harbour, the waterline and the ratio-log techniques performed similarly for the task of deriving topography over intertidal zones using satellite images. Thus, for estuaries with low water column turbidity, pre-existing surveyed topo-bathymetric data, and low numbers of available satellite images covering its area — as is the case of Tauranga Harbour — the ratio-log method could potentially replace the waterline method for deriving elevation data for intertidal zones. Although the waterline method shows better performance when considering the RMSE — either evaluated on a point-to-point basis (0.20 m) or evaluated using the DEM (0.23 m), see Table 3 — than the ratio-log method (0.25 m). Evaluating RMSE using the DEM provides more information for comparison. Figure 12 shows the density SDT points and distribution of the relative vertical error (RE) for Tauranga Harbour's waterline-SDT and ratio-log-SDT for intertidal zones, where the colour represents positive (red) or negative (blue) errors. Positive (negative) errors indicate that SDT estimates are deeper (shallower) or further landward (seaward) than the LiDAR data (see Sect. 2.7). The waterline-SDT (Figure 12: a1, b1, c1, d1) provides estimates that are generally shallower or further seaward than the LiDAR — as the negative RE indicates — with the worst estimates in the tidal flat's upper region (bluer colour dots). The positive RE values (redder colour dots) are concentrated in the estuary's wide flat centre region (Figure 12 b1) and indicate that the estimates are deeper or further landward than the LiDAR data. As discussed in Sect. 4.1, the waterline method is mainly limited by the number of images required to properly define the morphology of the study site. In the case of Tauranga Harbour, as consequence of the high complexity of its morphology, the SDT provided by the present framework could be substantially improved with more images, making the waterline method even more accurate than the ratio-log method. "*

Similarly, we repeat the rationale for Sections 2.5.1, 2.5.2, and 2.6. We also made the description of Figure 6 more clear (please see the markups).

6) For a better reading and to stress the main results, I suggest authors to merge Discussion and Conclusions and summarize the content.

We understand and appreciate your point; however, because we are discussing different points in the discussion (separated in subsections), and that initial reviews found our paper confusing (and also reviewer 2), we would like to retain a separate discussion so that we can clearly lay out our main findings using subsections.

**Minor comments**

1) There are still inconsistencies in unit system, for instance lines 14, 15 and 16.

Change made: all units are in metres through all the document.

2) In line 122 the term is "storm surge", please correct that expression throughout the manuscript.

Change made: the term was changed for storm surge as required throughout the manuscript.

3) In line 330 please be careful with the use of the correlation in this context, is it correlation or the goodness of fit? In any case, are they tested?

Change made. We changed the term for the goodness of fit (line 367). Goodness of fit is tested for Tauranga Harbour results, as shown in the results section.

4) For a better reading, before the analysis, first introduce the figure and its aim.

Change made. We have introduced first the figure and its aim before analysis for Figure 7 (lines 364–371). We also tried to follow this pattern throughout the manuscript, whenever practicable.

5) In section 3.4 homogenize the statistical evaluation, please use the same scores for a better interpretation.

Change made. We homogenized and clarified the text in Section 3.4.

6) In line 386, what is the third part?

We have removed this sentence from the manuscript.

**Typo**

Q1 - l8 Should be ",e.g.,"

Done. Re-arranged for "used in models.." (line 8).

Q2 - l16 The sentence is too wordy, please rewrite to stress your idea.

Done. The sentence was modified (lines 16–19) *"The use of SDT in numerical simulations of surge levels was assessed for Tauranga Harbour in eight different simulation scenarios. Each scenario explored different ways of incorporating the SDT to replace the topographic data collected using non-satellite survey methods. In addition, one of these scenarios combined SDT (for intertidal zones) and SDB (for subtidal bathymetry), so only satellite information is used in surge modelling."*

Q3 - L38 Should be "areas, respectively".

Done. Line 39

Q4 - L58 SBD is not defined yet.

It is defined in the previous paragraph (line 52).

Q5 - L87 SBD and SDT were defined above.

Yes, they were. SDB (line 52); SDT (line70)

Q6 - L125 Remove "study site"

Done, removed from line 125. It is study areas now.

Q7 - L130, L231 Error in latex citing reference.

Done. Reference removed to clarify the text. The reference is used in the discussion Section 4.1.

Q8 - L135 Was or is?

We re-arranged the sentence to: "the topography data consisted of the LiDAR survey, with a spatial resolution of $1 \times 1$ m, available on the Land Information New Zealand (LINZ) data portal (https://data.linz.govt.nz/)" (lines 142–143).

Q9 - L184 The equations should be separated according to the reading flow.

Done. Please see lines 190–200.

Q10 - L207 Should be "2d"

Done. Corrected at line 222.

Q11 - L209 SDT or waterline-SDT?

Done. We re-wrote the sentence to clarify. We use the term waterline-SDT hereafter. Line 222.

Q12 - L247 Please avoid expressions such as the last sentence.

Done. Sentence was replaced with "*For details on the model calibration and validation, see Supplement B.*" line 262.

Q13 - L287 The statistical scores acronymous are not defined yet.

Done. We have deleted this sentence. The full name of the statistical scores and acronymous are explained in Section 2.7.

Q14 - L327 MRE is not defined.

We have adjusted the term. We meant BIAS. We have adjusted Figure 7, equation 8, and the text (lines 364–373) according.

Q15 - L534 IWD is defined but then it's not utilized.

Done. "IWD" was removed from the text. (line 618)

Q16 - L554 SLR stands for sea level rise?

Done. We replaced "SLR" with "sea level rise". (line 618)

I have finished my report on the revised version of Costa et al. 'Modelling extreme water levels using intertidal topography and bathymetry derived from multispectral satellite images'. I'm happy to inform that I'm more satisfied with the current version of this study which clarified several issues I had from the original submission. However, I still think the revised manuscript is quite complicated to understand and unbalanced (e.g. compare length of results vs methods). Part of the difficulties I have to comprehend relates to the use of the language and the length and complexity of methods. The latter extends from pg 5 to 15 excluding details allocated in supplementary material. The revised submission needs a good review by the authors emphasizing on the appropriate use of scientific language (conciseness, objectiveness, etc…) and attention to detail. As an overseas academic based in Australia whose first language is not English, I understand some of the language difficulties are not restricted to Mr. Costa only. A thorough revision is therefore needed. This should start on Ln3 (missing space between L.L.); full address for 1 and reduced address for 2 (Ln4 and 5) and follows down all the way to the reference list. The figures are difficult to understand and the captions need improving. Since the study was conducted in similar estuaries in NZ, a revised discussion would also need to contemplate the range of estuarine morphologies and forcing for the international audience. Below are some of the minor issues I detected.

We are pleased that the reviewer appreciates the modifications made in the first round of revisions. We acknowledge your time and effort put into this second review. Below you can find your comments answered one by one in blue font. Your original comments are in black font.
We are aware of the complexity of the manuscript. We have made extensive modifications to improve clarity and conciseness, which we hope will satisfy your requirements. These are shown in the marked-up copy of the manuscript.

We have also addressed all your questions. We also have re-written most of the methods section to first express what we did and why we did it and linked the methods clearly to the corresponding discussion section. For instance, in Sect. 2.3, the first two paragraphs clarify that we are comparing ratio-log and waterline methods and why. In Section 4.3, the corresponding discussion section, we start the discussion directly comparing the methods, as follows (lines 547–563):

*"The results show that, for Tauranga Harbour, the waterline and the ratio-log techniques performed similarly for the task of deriving topography over intertidal zones using satellite images. Thus, for estuaries with low water column turbidity, pre-existing*

*surveyed topo-bathymetric data, and low numbers of available satellite images covering its area — as is the case of Tauranga Harbour — the ratio-log method could potentially replace the waterline method for deriving elevation data for intertidal zones. Although the waterline method shows better performance when considering the RMSE — either evaluated on a point-to-point basis (0.20 m) or evaluated using the DEM (0.23 m), see Table 3 — than the ratio-log method (0.25 m). Evaluating RMSE using the DEM provides more information for comparison. Figure 12 shows the density SDT points and distribution of the relative vertical error (RE) for Tauranga Harbour's waterline-SDT and ratio-log-SDT for intertidal zones, where the colour represents positive (red) or negative (blue) errors. Positive (negative) errors indicate that SDT estimates are deeper (shallower) or further landward (seaward) than the LiDAR data (see Sect. 2.7). The waterline-SDT (Figure 12: a1, b1, c1, d1) provides estimates that are generally shallower or further seaward than the LiDAR — as the negative RE indicates — with the worst estimates in the tidal flat's upper region (bluer colour dots). The positive RE values (redder colour dots) are concentrated in the estuary's wide flat centre region (Figure 12 b1) and indicate that the estimates are deeper or further landward than the LiDAR data. As discussed in Sect. 4.1, the waterline method is mainly limited by the number of images required to properly define the morphology of the study site. In the case of Tauranga Harbour, as consequence of the high complexity of its morphology, the SDT provided by the present framework could be substantially improved with more images, making the waterline method even more accurate than the ratio-log method."*

Given our paper is a test of different techniques, we think it is important to have a solid methods section, but we acknowledge that it was a little long, and have made an effort to make it more concise.

Minor:
Q1 - Ln 12- four instead of 4
Done. Line 12.

Q2 - Ln24 sea-level rise instead of sea level rise. Please revise this throughout paper (E.g. Ln34…). Note that we write 'sea level' and 'sea-level rise'.
Done. Revised throughout the paper.

Q3 - Ln36 Not sure I agree with the fact that inundated areas in estuaries are generally shallow. What do you mean by this assertion? What do you consider shallow? 2, 5, 20, 50 m below msl. Some estuaries are deeper than that.
Done. We removed the comment in parenthesis *"which are normally shallow"*. Now

the sentence is written as follows (lines 37–39):

*"In estuaries, there are permanently inundated areas and intertidal zones, which are flooded and exposed by the tide. Here we define the terms bathymetry and topography to reflect permanently-inundated and intertidal areas, respectively ".*

Q4 - Ln 37 Consistency--Consider placing 'which are areas flooded and exposed...' into (). Alternatively take out the () of 'which are generally shallow'.
Done. We have reworded this sentence (lines 37–38).

Q5 - Ln 44 reword this sentence as the way it is written it seems that ~70% of the coast has not been surveyed, when in fact the whole world has been surveyed. What scale are you talking about? Make sure you are referring to large scale (e.g 1:100)
Done. The sentence was re-worded as follows (lines 45–46): *"Consequently, according to IHO (2020), approximately 70% of the world's coastal areas have bathymetric surveys that need updating or are insufficiently detailed (e.g., are of large scale 1:100)."*

Q6 - Ln 104-113 _ Is this explanation needed on a scientific paper? This seems to be a statement adapted from Mr. Costa's thesis. Please consider revising it.
Done. A description of the sections was requested in the first round of reviews, and that is why we put so much detail in the previous version. We agree with the comment, and have modified the text to closely follow the Figure 1 (lines 106–119).

Q7 - Ln115 you previously indicated that you had three specific objectives (Ln98). Do you want to elaborate more on this Fig caption to indicate that (a) not only shows the steps taken to derive SDT/SDB but also to investigate stats relations/sources of errors?
Done. The legend of Figure 1 has been changed accordingly:
*"A flow chart showing the main structure of the manuscript. The panel (a) shows the steps taken to derive the SDT/SDB and how the statistical relationships and source of errors were investigated. The panel (b) summarizes the framework to test the utility of SDT/SDB in modelling high water levels."*

Q8 - Ln118 What's Aotearoa? Some part of text you say Aotearoa NZ, some others are only NZ. Is there a difference? for the international reader?
Done. We have changed the text throughout the manuscript. We now refer to the country "New Zealand" as "Aotearoa New Zealand". Aotearoa is the Māori name for New Zealand. It means "the land of the great white cloud". In NZ, there is a treaty between the crown and Māori that requires us to provide equal weight to Māori interests, and we have been advised that using "Aotearoa New Zealand" shows

support for respecting this Treaty (it has also become common practice in NZ). Our funding body also requires us to show the ways in which we are respecting the Treaty in our work.    .

Q9 - Ln119 (Figure 2A) Ln 120 the spring tide ranges from 1.4 to 1.9 m within estuaries
Done. We have modified "Figure 2A" for "Figure 2a" and remove the "within estuaries". Line 126–128.

Q10 - Ln121-123- Revise text. Is the use of BUT appropriate? I don't understand what's being said here. Storm surges add <0.5 m but the max surge was 2,29 m? I guess you mean the extreme sea level. Similar for the next sentence--- Tauranga = 0.88 m
Done. We have re-written the text as follows (lines 127–132):
*"The sites consist of barrier-enclosed sandy estuaries which are common in Aotearoa New Zealand (Hume et al., 2015) and all have micro-tidal regimes — the spring tidal range varies between 1.4 m to 1.9 m – and spring tides combined with severe storm surges drive the extreme sea levels (Rueda et al., 2019; Stephens et al., 2020). In Aotearoa New Zealand, the storm surges usually add ≤ 0.5 m to the water level; however larger storm surges can occur occasionally (Stephens et al., 2020). The extensive intertidal zones and vegetation (e.g., seagrass and mangrove) that are present in the majority of the estuaries in Aotearoa New Zealand can attenuate tides (Tay et al., 2013) and storm surges (Montgomery et al., 2019)."*

Q11 -Ln135 1 x 1 m – Add space
Done. Line 142.

Q12 - Fig 2 I have a major cartographic issue with this figure. Your use of different shapes and colours is completely inappropriate here. It makes the reader completely lost. See: You use coloured circles to represent the four locations (nothing wrong with it!). Then in B, you use different symbols with different colours and sizes to represent the different gauges. This is clearly a thematic cartography problem as we don't normally vary all three elements at once. In doing so, I tend to relate the Moturiki gauge with the Tauranga Harbour in A – they are both green circles! Then you complicate things with the D plots: Omokoroa is represented by black stars when legend in B shows red star, Hairini is also a black triangle and Oruamatua a black box. Finally, you mess it up one more time with the yellow and red symbols in legends! When I read the caption to try to understand this Fig, I don't find a reason why you have four gauges in B and only three plots in D. What's wrong with Moturiki (Green)? You need to spend more time on this figure. A lot of your issues come from the symbology in B. Try to make it easier for the reader.
We have modified Figure 2 to remove confusion. The symbols now follow a logical

colour and shape.

Q13 - Ln191 The threshold is set using the Otsu approach (Otsu, 1979) – Refine text to avoid repetition of the author's name

Done. We modified the text to *"…The threshold is set using the Otsu (1979) approach…"* (line 204).

Q14 - Table 1 – Surface area column – Round the area values to represent the approximation you indicated. Why are they in italic?

Done. The number were rounded and the font italic was removed.

Q15 - Fig 3 – This Fig summarises the framework for deriving topographic data. OK, but what is 4 PostProcessing? Why is this stage not explained as the other three? Why does it come before 3 in this framework? The caption also needs attention. It indicated that this is the framework and that NDWI is the index used, but stops there. The caption is extremely limited for such a complicated figure!

Done. Step 4 post processing was eliminated from the figure. We refer to post-processing as a manual quality control step where incorrectly identified waterlines were eliminated    (for example, as caused by a small cloud or shadow of the cloud). This is explained in the text (lines 168–176). The legend was re-written as:

*" Figure 1:   The framework for application of the waterline method to derive topographic data in intertidal zones.   First (1) an image collection was acquired. Second (2), the intertidal zone was identified by calculating the temporal of NDWI. Note that NDWI is the index used to detect the existence of water from satellite reflectance (see text). Third (3), the waterline position and height were determined. This was done by identifying the boundary between wet and dry cells within the intertidal zone (i.e., waterline) and assigning a height value for the waterline obtained from the local tide gauge observation at the time of the image acquisition."*

We hope now with the improvements to the legend, the main text, and the text embedded in the figure,provide a better understanding of the method.

Q16 - Fig 5 – Make this a 3 x 3 plot figure instead of a 4x4x1. Use your programming skills to make better use of the page's space.

Done.

Q17 - Ln 225 What is a sub-estuary? Please clarify.

Done. We modified the sentence to (lines 237—240) :

*"To compare, the ratio-log method was applied to an image acquired at high tide,*

*where the intertidal zone was completely flooded. The numerical assessment was built on a pilot study in Costa et al. (2021), where the method was trialled in small region within the Tauranga Harbour."*

Q18 - Ln247 Can you expand on "The model approximates the predicted data well (Sup B)"? How well? Please use a stat method to be more precise here. A single sentence will do it. Then refer to Sup B.

Done. To improve the conciseness of the text, we modified the sentence to: "For details on the model calibration and validation, see Supplement B." (line 262)

Q19 - Ln255- Rewrite this sentence to indicate that only groundwater can be a potential source of error. If that's what you mean
Done. Note that there are several factors that can affect the waterline method, groundwater is only one of them. We re-wrote the paragraph (lines 266–270) as follows:

*"The first method to correct the waterline-SDT trialled was to remove the statistical bias—potentially caused by conditions that can interfere in the pixel reflectance and as consequence, the waterline position at different tide levels within the tidal flats. Conditions that can interfere with detection include the complexity of the intertidal zone morphology, water turbidity, variation of the benthic substrates (sand, seagrass), and groundwater seepage. Specifically, groundwater seepage leaves a film of moisture on the exposed intertidal detectable in images (Huisman et al., 2011). "*

Q20 - Ln263 - 265– Rewrite paragraph to make your point clearer for the reader.
Done. See comment above.

Q21 - Ln330/331 I assume R2 is $R^2$ Please use superscript. Revise whole text
Done. We put superscript in all text.

Q22 - Ln332 – There's no Figure 6C
Done. Figure 7c (line 368).

Q23 - Ln335 – Is this a separate equation or part of plot 7?
It is contained in plot 7c. We made this more clear in the text. Line(369).

Q24 - Fig. 8 - Align the bottom two plots in relation to the second and third columns

Done.

Q25 - Ln359/365 Use spaces RMSE~7cm, etc...
Done. We added space in these cases in all manuscript.

Q26 - Fig 10 Why are the gauge symbols different from Fig2? Delete the ')' from y-axis in top left plot (after 60S)
The figure was modified address the issues raised by the reviewer (symbols and typos). Note that as part of the reviewing process, we added new simulation scenarios and the figure setting has changed.

Q27 - Ln386-390 – Discuss with your co-authors the need of this paragraph. This is supposed to be a discussion of your results and not a summary of what was done!
Done. The paragraph was deleted.

Q28 - Ln391 'Our' – Throughout text you write possessive forms. Stick to scientific language. Avoid Ours, We, etc... What about: '4.1 Waterline method for....limitations'; 4.2 Correction methods for... ?
Done. Possessive forms removed from the section's titles and text.

Q29 - Ln400 New Zealand
Done. We changed to "Aotearoa New Zealand's" (line 445)

Q30 - Ln 401 Our results also show.
Done. Text modified to (line 446): *"Environmental conditions such as the complex morphology, varied bed substrates, and groundwater seepage could reduce the accuracy of the waterline position."*.

Q31 - Ln401-402 I'm not sure I understand this discussion. I was under the impression that all estuaries were quite similar and only groundwater had the potential to reduce the accuracy. Here you are indicating that that environmental conditions such as complex morphology can also reduce it. There's a weak argument here. Maybe you referring to the different estuarine types and stages of evolution and how this can interfere with results but this needs better arguments and references. Given the range of estuaries and the international audience of this journal. Is there space for micro x macro tidal discussion in here? Is NDWI a good proxy in macrotidal settings or flat (low slope) shorelines?
Ground water seepage is only one of the factors that could cause errors in the waterline method. We explain this better now in Section 2.5.1. NDWI is widely applied

in other studies, however, other techniques can be used to determine the intertidal zones and waterline (we discuss these in this same section, lines 482–490).    We understand that international readers will not be familiar with the estuaries that we have studied. We have add text that they are barrier-enclosed micro tidal estuaries and a reference a paper that classifies our estuary types (Hume et al. 2007, 2016). We now made clear that the tidal regime is important in the waterline method (lines 466— 468).

Q32 - Ln 455 Rewrite we are not eliminating horizontal errors. This is not scientific language at all!

Done. We have re-written the sentence as follows (lines 509–510): *"Note that in the dynamical correction, just the waterline height is corrected, and the observed waterline position remains unchanged."* .

Q33 - Fig 11- I'm completely lost here. What am I looking at? I see the three profiles. The points are waterlines, but the map has lines, which I suppose are SDT WLH according to the legend below…Not sure what you mean here. Then on the right, you indicate p1 being the dyn. Corr, WLH 1; p2 being the dyn corr WLH2… Do you see what I am seeing?

We have rewritten the paragraph in the manuscript that refers to the figure (lines 501– 517). The figure clearly shows three different waterlines throughout the estuary (green, red, blue lines) plotted on the map and profiles intersecting these waterlines (m1). The profiles and the intersections are represented by points in panels p1, p2, p3 (the right panels). We simplified Figure 11 and made it more concise.

Q34 - Ln483 Capital letters- Check manuscript for proper names. We say Arcachon Bay, Maketu Estuary (Ln317)

Done. These have been changed throughout the manuscript.

Q35 - Ln492 This is such a big claim considering that your method was only tested locally under limited conditions and restricted estuarine settings. Please reframe this to place your findings accordingly.

Done. We have changed the text as follows (lines 547–551):

*"The results show that, for Tauranga Harbour, the waterline and the ratio-log techniques performed similarly for the task of deriving topography over intertidal zones using satellite images. Thus, for estuaries with low water column turbidity, pre-existing surveyed topo-bathymetric data, and low numbers of available satellite images covering its area — as is the case of Tauranga Harbour — the ratio-log method could*

*potentially replace the waterline method for deriving elevation data for intertidal zones.*"

Q36 - Ln492 – 508 You keep indicating that the waterline-SDT performs better than the ratio-log-SDT, but the text suggests that the latter is done on px-by-pixel basis. Is this the reason why the former outperforms the latter? Shouldn't you be comparing only the lines here?

Done. We have rewritten several paragraphs of Section 4.3. We clarify the comparison between methods and results. In addition to the changes in response to Q35, the first paragraph now reads (lines 551–563):

*" Although the waterline method shows better performance when considering the RMSE — either evaluated on a point-to-point basis (0.20 m) or evaluated using the DEM (0.23 m), see Table 3 — than the ratio-log method (0.25 m). Evaluating RMSE using the DEM provides more information for comparison. Figure 12 shows the density SDT points and distribution of the relative vertical error (RE) for Tauranga Harbour's waterline-SDT and ratio-log-SDT for intertidal zones, where the colour represents positive (red) or negative (blue) errors. Positive (negative) errors indicate that SDT estimates are deeper (shallower) or further landward (seaward) than the LiDAR data (see Sect. 2.7). The waterline-SDT (Figure 12: a1, b1, c1, d1) provides estimates that are generally shallower or further seaward than the LiDAR — as the negative RE indicates — with the worst estimates in the tidal flat's upper region (bluer colour dots). The positive RE values (redder colour dots) are concentrated in the estuary's wide flat centre region (Figure 12 b1) and indicate that the estimates are deeper or further landward than the LiDAR data. As discussed in Sect. 4.1, the waterline method is mainly limited by the number of images required to properly define the morphology of the study site. In the case of Tauranga Harbour, as consequence of the high complexity of its morphology, the SDT provided by the present framework could be substantially improved with more images, making the waterline method even more accurate than the ratio-log method. "*

Q37 - Fig 12. Fix caption –missing ) after d1. Expand caption to explain why left is better than right.

Done. The caption of Figure 12 was modified according to the requirements as follows:
*"Figure 12: Estimated SDT and corresponding relative vertical error (LiDAR-SDT) for intertidal zone in Tauranga Harbour using waterline-derived (a1, b1, c1, and d1) and ratio-log (a2, b2, c2, and d2) methods. The root mean-squared error for waterline method is 0.20 m and for the ratio-log method is 0.25 m (not shown in the figure). However, the waterline method results in less density of estimates (due to imagery*

*constraints), while ratio-log method results in a pixel-by-pixel estimate density. Background image: ESA Sentinel 2A. Date and time of the background image acquisition: 18/12/2018 10:15 h."*

Q38 - Ln518 would be more uncertain than what?

Done. We change the sentence to (line 586): *"Consequently, determining the intertidal areas would be less accurate in environments with high concentration of suspend material."*

Q39 - Ln593 onwards: My editorial eye couldn't let this pass without a comment. A comprehensive revision of the reference list is needed. There are references in CAPITAL letters; typos; several words together; wrong author names; wrong abbreviations; lack of abbreviations, missing information; etc… These start in the very first ref (below) but extent throughout the list Almeida, L. P., Efraim de Oliveira, I., Lyra, R., Scaranto Dazzi, R. L., Martins, V. G., andHenrique da Fontoura Klein, A.: Coastal Analyst System from Space Imagery Engine (CASSIE): Shoreline management module, Environ. Model. Softw., 140, 105033, https://doi.org/10.1016/j.envsoft.2021.105033, 2021. Some of these issues are observed on a single ref.: 'Costa, W., Bryan, K. R., Coco, G., Zealand, N., andZealand, N.: ASSESSING THE USE OF SATELLITE DERIVED BATHYMETRY IN ESTUARINE STORM SURGE MODELS – STUDY CASE : TAURANGA, 2021.

Done. A thorough revision of the references formatting has been done.